

# A comparison between Envisat and ICESat sea ice thickness in the Antarctic

Jinfei Wang[1,2], Chao Min[1,2], Robert Ricker[3], Qinghua Yang[1,2], Qian Shi[1,2], Bo Han[1,2], Stefan Hendricks[3]

[1]School of Atmospheric Sciences and Guangdong Province Key Laboratory for Climate Change and Natural Disaster Studies,
Sun Yat-sen University, Zhuhai 519082, China
[2]Southern Marine Science and Engineering Guangdong Laboratory (Zhuhai), Zhuhai 519082, China
[3]Alfred Wegener Institute Helmholtz Centre for Polar and Marine Research, Bremerhaven 27570, Germany

*Correspondence to*: Qinghua Yang (yangqh25@mail.sysu.edu.cn) and Qian Shi (shiq9@mail.sysu.edu.cn)

**Abstract.** The crucial role that Antarctic sea ice plays in the global climate system is strongly linked to its thickness. While in situ observations are too sparse in the Antarctic to determine long-term trends of the Antarctic sea ice thickness on a global scale, satellite radar altimetry data can be applied with a promising prospect. A newly released Envisat-derived product from the European Space Agency Sea Ice Climate Change Initiative (ESA SICCI), including sea ice freeboard and sea ice thickness, covers the entire Antarctic year-round from 2002 to 2012. In this study, the SICCI Envisat sea ice thickness in the Antarctic is firstly compared with a conceptually new proposed ICESat ice thickness that has been derived from an algorithm employing modified ice density. Both data sets have been validated with the Weddell Sea upward looking sonar measurements (ULS), indicating that ICESat agrees better with field observations. The inter-comparisons are conducted for three seasons except winter based on the ICESat operating periods. According to the results, the deviations between Envisat and ICESat sea ice thickness are different considering different seasons, years and regions. More specifically, the smallest average deviation between Envisat and ICESat sea ice thickness exists in spring by -0.03 m while larger deviations exist in summer and autumn by 0.86 m and 0.62 m, respectively. Although the smallest absolute deviation occurs in spring 2005 by 0.02 m, the largest correlation coefficient appears in autumn 2004 by 0.77. The largest positive deviation occurs in the western Weddell Sea by 1.03 m in summer while the largest negative deviation occurs in the Eastern Antarctic by -0.25 m in spring. Potential reasons for those deviations mainly deduce from the limitations of Envisat radar altimeter affected by the weather conditions and the surface roughness as well as the different retrieval algorithms. The better performance in spring of Envisat has a potential relation with relative humidity.

## 1 Introduction

Antarctic sea ice plays an important role in the global climate system by reflecting the solar energy and modulating the surface water salinity (Goosse and Zunz, 2014; Massom et al., 2018; Maksym, 2019). In the context of global warming and the significant declines of Arctic sea ice cover, Antarctic sea ice extent has unexpectedly increased over recent decades (Zhang, 2007; Parkinson and Cavalieri, 2012; Comiso et al., 2017), but dropped to a historic low in 2017 (Turner and



Comiso, 2017). However, it is still unclear if the recent increase in Antarctic sea ice extent is also associated with a similar change in sea ice thickness (SIT). Compared to the Arctic, knowledge about Antarctic sea ice thickness is still sparse. As sea ice thickness is an equally critical component as sea ice extent, more accurate estimations are needed, in that the total sea ice volume could be quantified and monitored more precisely (Connor et al., 2009), and the sea ice dynamics and

thermodynamic processes could be guided to better simulations in the models (e.g., McLaren et al., 2006).

However, Antarctic sea ice thickness information is difficult to obtain due to the harsh weather conditions, thick snow and sometimes complex snow metamorphism, which can interfere with the satellite radar measurements. In situ measurements like drilling data (e.g., Meiners et al., 2012; Ozsoy-Cicek et al., 2013) are accurate but extremely limited to a short time and a narrow space, and hence they cannot be used to obtain an understanding of the large-scale Antarctic sea ice thickness.

Ship-based observations collected from the Antarctic Sea Ice Processes and Climate (ASPeCt) (Worby et al., 2008a) can provide more spatial information than drilling but they tend to underestimate the actual thickness because of visual interpretation errors and biases due to ship routing preferably through thinner ice (Giles et al., 2008; Williams et al., 2015). In addition, airborne electromagnetic (AEM) data were collected during expeditions like ISPOL (2004/05) (El Naggar et al., 2007) and WWOS (2006) (Lemke, 2009). Yet the Antarctic AEM data is still sparse and only exists in the Weddell Sea. The

airborne remote sensing program NASA Operation IceBridge provides time series of freeboard observations (Koenig et al., 2010) but also limited to several trajectories in the Weddell Sea. Upward looking sonars (ULS), located at 13 different sites in the Weddell Sea, provides valuable temporal evolution of sea ice thickness (Harms et al., 2001; Behrendt et al., 2013a; Behrendt et al., 2013b), but the basin-wide spatial distribution cannot be derived. More recently, satellite remote sensing has been widely applied to investigate the spatial coverage and long-term trend of sea ice thickness in the whole Antarctic (e.g.,

Kurtz and Markus, 2012; Bernstein et al., 2015). Passive microwave sensors are used to obtain thin ice thickness (basically below 1 m) by retrieving the brightness temperature, and are effectively applied in coastal polynyas (Nihashi and Ohshima, 2015). Active microwave sensors such as SAR (Nakamura et al., 2009) and satellite altimetry including radar and laser altimetry also have been used in the Antarctic to retrieve sea ice thickness (e.g., Giles et al., 2008; Zwally et al., 2008).

Within the framework of the Sea Ice Climate Change Initiative (SICCI) project, radar altimeter data collected by European

Space Agency (ESA) satellites over the past three decades has been reprocessed and assessed. Based on these data, a new SICCI sea ice thickness data was released in 2018, including the two radar altimetry satellites Envisat and CyroSat-2 (Hendricks et al., 2018a; Hendricks et al., 2018b). The SICCI product covers the entire Antarctic sea ice for the complete annual cycle from 2002 to 2017, and therefore, it is one of the latest sea ice thickness data sets publicly available. Thickness retrieval from radar altimetry is based on the assumption that the dominant source of radar backscatter is the snow/ice

interface (Beaven et al., 1995) and sea ice freeboard is measured by differential ranging over sea ice and ocean surfaces, illustrated in Fig. 1. However, for Antarctic sea ice, multiple backscatter interface might exist due to the complex snow stratigraphy and frequent snow flooding associated with the formation of snow ice and superimposed ice (Willatt et al., 2010). The uncertainties of the radar altimeter range retrieval also originate from the surface roughness (Hendricks et al., 2010). Due to the large footprint, Ku band radar altimeters are prone to errors from surface type mixing, which introduces



additional range biases for imperfect surface type classification. Meanwhile, the snow depth estimations that are needed for
retrieving ice thickness, are quite unreliable in the Antarctic (Worby et al., 2008b). Therefore, sea-ice thickness retrieval
algorithms must use climatologies for snow depth on sea ice in the southern hemisphere that are less developed and quality-
controlled than their counterparts in the northern hemisphere. The SICCI Antarctic sea-ice thickness data record has
therefore been categorized as experimental by the data producers compared to a more mature climate data record in the

Arctic.

The Geoscience Laser Altimeter System (GLAS) aboard the Ice, Cloud and land Elevation Satellite (ICESat) estimates total
freeboard (sea ice freeboard + snow depth) from 2003 to 2009, illustrated in Fig. 1. This data set has been investigated for
many years (e.g., Markus et al., 2011; Yi et al., 2011; Kurtz and Markus, 2012; Xie et al., 2013; Kern and Spreen, 2015).
Several freeboard-to-thickness retrieval algorithms have been developed (Kern et al., 2016). Compared to radar altimetry,

laser altimetry has the advantage of a well-defined reflective horizon, which is the air/snow interface. The main deficiencies
of ICESat data are data gaps due to cloud coverage, and more generally the discontinuous and short observation period.
Therefore, ICESat data cannot reflect the current characteristics of the fast-changing Antarctic sea ice.

The Envisat sea ice thickness data in the Antarctic has already been evaluated with the drilling data, airborne electromagnetic
(EM) sounding, moored upward looking sonar (ULS) and some ship-based data (SICCI-PVIR-SIT, 2018). These evaluations

are comprehensive but still have their limitations due to small spatial coverage or short temporal coverage, thus we cannot
achieve an overall understanding of the data quality in the Antarctic.

To get a better understanding of the characteristics of the newly released SICCI product, we aim to investigate how the
SICCI Envisat retrieval compares to the ICESat sea ice thickness data record, also how the different retrieval methods are
represented in the ice thickness distribution. Based on the former inter-comparison study, we choose the ICESat sea ice

thickness data derived from the modified ice density approach suggested by Worby (Kern et al., 2016) as our reference,
which seems to agree with independent observations best and has a reasonable winter-to-spring growth (Kern et al., 2016).
Furthermore, in order to evaluate the ICESat and Envisat data records in the Weddell Sea, we also compare both sea ice
thickness records with the Weddell Sea upward looking sonar (ULS) sea ice draft data first.

The study is organized as follows. In section 2, we describe the data used in this study in detail. Section 3 presents the results

of both the Weddell Sea ULS data validations and the inter-comparisons between the two data sets. Potential reasons for the
spatial and temporal deviations are discussed in section 4. The main conclusions are summarized in section 5 with further
discussions.

## 2 Data and methods

### 2.1 Sea ice thickness from Envisat / RA-2

The ESA CCI Sea Ice Climate Change Initiative (SICCI) provides a set of Antarctic sea ice freeboard and thickness data
([https://dx.doi.org/10.5285/b1f1ac03077b4aa784c5a413a2210bf5;](https://dx.doi.org/10.5285/b1f1ac03077b4aa784c5a413a2210bf5) Hendricks et al., 2018b) obtained from the satellite



missions Envisat (2002–2012) and CryoSat-2 (2010–2017). With 50-km gridded resolution and monthly temporal resolution, there is a consistent record for Antarctic sea ice freeboard and thickness for all month of the year. The freeboard measured by radar altimeter is defined as the sea ice freeboard shown in Fig. 1, which is the sea ice surface elevation relative to the sea

surface elevation. The sea ice thickness data is retrieved from ice freeboard based on the hydrostatic equilibrium approach (Kern et al., 2016):

$$I = \frac{F\rho_{water} + S\rho_{snow}}{\rho_{water} - \rho_{ice}},$$ (1)

where F represents Envisat ice freeboard, S represents snow depth, I represents ice thickness. AMSR-E snow depth climatology is employed to retrieve sea ice thickness from sea ice freeboard here.

Since only Envisat shares overlapping period with ICESat, we focus on the measurement characteristics of the Envisat radar altimeter (RA-2). Envisat RA-2 operates at the main frequency of 13.575 GHz (Ku band), with a secondary frequency of 3.2 GHz (S band) compensating the ionospheric error (Zelli and Aerospazio, 1999).

In addition, to mitigate the influence of open water, we only compare girds with sea ice concentration over 70% during the evaluations and inter-comparisons. Since the sea ice thickness values do not consider the fraction of open water, we also

multiply the sea ice concentration which is included in the data for each grid.

**2.2 Sea ice thickness from ICESat / GLAS**

The Ice, Cloud, and land Elevation Satellite (ICESat), operated as part of NASA's Earth Observing System, provides a set of Antarctic freeboard data from 2003 to 2009. However, the measurements are not continuous due to cloud coverage and each measurement campaign lasts for about 35 days. Different from the sea ice freeboard measured by radar altimeters, laser

altimeters detect the distance between the snow surface and sea surface, which is called total freeboard, as shown in Fig. 1. There are five main ICESat sea ice thickness data sets derived from different retrieval algorithms. Qualitative inter-comparisons have been done among several ICESat freeboard-to-thickness retrieval approaches (Kern et al., 2016). According to the conclusion by Kern et al. (2016), we choose the product derived with the modified ice density approach suggested by Worby as the reference in this study, because of its reasonable winter-to-spring increase and better agreements

with independent data. This is part of the SICCI phase 1 products and provided by Integrated Climate Data Center (ICDC, http://icdc.cen.uni-hamburg.de). The approach considers the snow–ice layer as one system with a modified density, in order to avoid using a potentially biased snow depth product. According to Kern et al. (2016), the modified density can be derived as follows:

$$\rho_{ice}^{*} = \frac{R\rho_{ice} + \rho_{snow}}{R + 1},$$ (2)

where R is the ratio of sea ice thickness over snow depth, which is a seasonally dependent factor and calculated from ASPeCt observations. And sea ice thickness can be determined from it:



$$I = F \frac{\rho_{water}}{\rho_{water} - \rho_{ice}^*},$$ (3)

where F represents ICESat ice freeboard.

Antarctic mean gridded sea ice freeboard and sea ice thickness with grid resolution of 100 km from 2004 to 2008 are
provided in this product. Table 1 presents the available time periods of the data. Only grid cells with a sea ice concentration
above 60% are used in the evaluations and comparisons.

### 2.3 Sea ice thickness from Weddell Sea ULS

The upward looking sonars (ULS) located in the Weddell Sea provides successive and high-frequency sea ice draft at each
site (https://doi.pangaea.de/10.1594/PANGAEA.785565; Behrendt et al., 2013a; Behrendt et al., 2013b). The mooring
locations are shown in Fig. 2. Among all the available in situ data in the Antarctic, the ULS data is the most suitable criterion
to evaluate the satellite data due to its accuracy and year-round continuity. We choose the sea ice draft measured during 2004
to 2008 to cover the ICESat periods in this study. Sea ice thickness (z) is converted from the sea ice draft (d) through an
empirical formula established from drilling data in the Weddell Sea (Harms et al., 2001):

$$z = 0.028 + 1.012d.$$ (4)

### 2.4 Meteorological data

As mentioned above, potential biases of Envisat sea ice thickness primarily result from the uncertainties of the snow
backscatter and the AMSR-E snow depth. Both of them are significantly affected by the weather conditions over the
Southern Ocean. Wet conditions can affect the dielectric properties of snow for that the dielectric constant of water is much
larger than that of dry snow (Hallikainen et al., 1986), and then weaken the penetration of radar altimeter signals into the
snow–ice interface (Willatt et al., 2010). Therefore, we assume that the relative humidity for the three seasons may be linked
to the deviations. In this study, we analyze monthly 1000hPa relative humidity data derived from the ERA-interim reanalysis
for each ICESat operating period during 2004–2008. We examine the data on the 2.5° × 2.5° latitude-longitude grid focused
on the whole Antarctic which is bounded by 60° S and 90° S and 180° E and 180° W.

### 2.5 Spatial and temporal divisions

The comparisons are realized by considering the differences between the two sea ice thickness data sets in different seasons
and different sectors. The seasonal classification is based on the ICESat operating periods presented in Table 1. For each
period, we choose the corresponding time periods during which Envisat monthly data are used, also given in Table 1. If
ICESat data has overlapping time over ten days with respective months, we average Envisat data over the two months.





Antarctic sea ice characteristics differ remarkably in different areas in the Southern Ocean. Therefore, we divide it into six
sectors following Worby et al. (2008a) and discuss the differences for each of them. The standard of the classification is
presented in Fig. 3.

# 3 Results

## 3.1 Comparisons with Weddell Sea ULS

Before the inter-comparison between Envisat and ICESat, the absolute accuracy of Envisat and ICESat sea ice thickness is
examined by a comparison with in situ observations at first. The ULS sea ice draft has been converted into monthly sea ice
thickness data with Eq. (4) in Sect. 2.3. As both Envisat and ICESat sea ice thickness have been processed into mean gridded
data including open water by multiplying them with sea ice concentration, we consider the open water fraction here and zero
thickness measured by ULS is included in the average. During ICESat operating period, there are only three sites with five
records suitable for the evaluation: 207-6, 229-5, 229-6, 231-6 and 231-7 (see also Fig. 2). Basically, the sites can be divided
into two regions. Site 207 is near the coast of the Antarctic Peninsula, characterized by deformed ice, while the others belong
to the eastern Weddell Sea, characterized by thin ice.

Figure 4 presents the time series of sea ice thickness for Weddell Sea ULS, Envisat and ICESat for each site available to
compare. Due to the discontinuity and lack of data along the southern coast, ICESat only has limited measurements for
evaluation. The Envisat gaps indicate that there is open water on the site, or thin ice with a concentration lower than 70%.
We find that both Envisat and ICESat are not consistent with the sea ice thickness observed from ULS. In the western
Weddell Sea, along the coast of the Antarctic Peninsula (at site 207), the ULS thickness ranges between 0 and 1.5 m, not
showing a clear seasonal cycle. Envisat thickness exceeds ULS remarkably, with the maximum value of about 4.2 m. The
relatively large error bars only cover part of the observations. In comparison, ICESat thickness still varies within a moderate
range, with a smaller root mean square deviation (RMSD) of 0.63 m. The error bars of ICESat also cannot cover the
observations. In the eastern Weddell Sea (at site 229 and 231), ICESat indicates open water where the ULS ice thickness is
smaller than 0.5 m, while having a few overestimations on thicker ice. In comparison, Envisat has larger overestimations, but
the error bars can cover almost all the observations. Except for site 231-7, ICESat thickness has smaller RMSD than Envisat.
In summary, ICESat thickness suggests better agreements with ULS measurements in the Weddell Sea while Envisat
thickness performs poorly. Therefore, during the inter-comparison in the following, we can know that Envisat has a larger
bias than ICESat and we will focus mainly on the potential errors in the Envisat data set, since ICESat uncertainties have
been discussed already in previous studies (Kern and Spreen, 2015; Kern et al., 2016).



### 3.2 Inter-comparisons between Envisat and ICESat

We first conduct an overall comparison between Envisat and ICESat for each ICESat operating period in each season, as shown in Fig. 5–7. The figures suggest that there are substantial inter-seasonal and interannual differences between the two sea ice thickness data sets.

In spring (ON), both positive and negative differences exist, shown in Fig. 5. And the average offset in spring seems smallest among the three seasons. They are both able to capture the thick ice lying in the western Weddell Sea and the Bellingshausen and Amundsen Sea. Deformed sea ice near the coast of western Pacific Ocean are also detected by both sensors, but seems thicker than the ship-based observations (Worby et al., 2008a). However, Envisat does not show the young ice in the Ross Sea while the Ross Ice Shelf polynya in ICESat maps are obvious. The same is found for the Ronne Ice Shelf polynya in the Weddell Sea. An anomalous phenomenon is that there is a thick ice tongue occurring in the Ross Sea in 2007 in both maps, which may come from the Amundsen Sea thick ice with a clockwise ice motion. In terms of the interannual variations, positive deviations between Envisat and ICESat SIT occur when thicker ice appears in 2004 and 2007, while negative deviations occur when ice thickness is thinner in 2005 and 2006. The sea ice thickness in latter years are smaller because the comparison periods are in November while former years in October. Since Envisat ice thickness is larger in October but smaller than ICESat in November, Envisat indicates a stronger seasonal cycle of sea ice thickness than ICESat. Table 2 provides the respective thickness, differences, root mean square deviations (RMSDs), the correlation coefficients (CCs) between Envisat and ICESat and the numbers of comparison pairs for each ICESat operating period. In general, Envisat has an overall negative deviation by -0.03 m compared to ICESat, mainly caused by the large negative deviation in 2006 by -0.24 m. Absolute differences in each year are relatively small among the three seasons and the RMSD for seasonal average comparison is the second smallest by 0.59 m. However, the correlation coefficients are not that high and range from 0.5 to 0.6, with a significance larger than 95%.

During summer (FM), the melting season, the ice coverage is rather small, limited to the western Weddell Sea, Bellingshausen and Amundsen Sea along the coast and southern Ross Sea, so that there are not many measurements to compare as presented in Fig. 6. In the western Weddell Sea, Envisat reveals that thick ice still exists and remains at least 3 m through all the years, while ICESat reveals thinner ice. Although both data sets present a reduction of thickness between 2005 and 2006, ICESat indicates a more distinct variation, which can also be reflected from the larger positive deviations from 2005 to 2006. As for the Ross Ice Shelf polynya, ICESat displays obvious thin ice there in 2004, 2007 and 2008. The lack of data in 2005 and 2006 may result from the low ice concentration which has been removed. Envisat also detects the Ross sea ice in 2006 and 2007, but the value is up to 1.5 m. According to Table 2, the numbers of comparison pairs are small. Generally, there is a mean positive deviation of 0.86 m in summer, the largest among the three seasons. The agreement between Envisat and ICESat seasonal average ice thickness in summer is bad, as the RMSD is the largest by 0.81 m and the correlation coefficient is the lowest by 0.34.



In autumn (MJ), regardless of the value range, the thickness patterns of the two data sets agree with each other well in each

215     year, shown in Fig. 7. Therefore, the errors are consistently positive over all regions except some regions in the Indian Ocean sector and the western Pacific Ocean sector. Comparing the values in the eastern Antarctic, ICESat shows some deformed ice up to 3 m while Envisat shows smaller thickness by about 1.5 m. Compared to summer, we can also see that the deviations in the western Weddell Sea spread to the whole Weddell Sea sector and slightly decrease from west to east. The ice growth in autumn could be recognized from this change. In addition, the positive errors in the Ross Ice Shelf polynya still

exist, mostly due to Envisat's disability to capture the thin ice there. As for the ice edge, ICESat presents some anomalous thick ice while Envisat presents a natural transition from to thinner ice. According to Table 2, the mean deviation for the entire Antarctic sea ice domain is 0.62 m, with a smallest RMSD of 0.56 m and a highest CC of 0.66. Although the deviations are large by 0.69 m and 0.64 m in 2004 and 2005, the correlation coefficients are high among all periods by 0.77 and 0.74, respectively.

From the probability distribution shown in Fig. 8, we can see that in spring Envisat agrees with ICESat well with a similar mean ice thickness by 1.57 m and 1.60 m, respectively. However, the agreement in summer turns bad, with an obvious malposition between two histograms. Envisat has the largest probability between 2.5 m and 3.0 m while ICESat has the largest probability between 1.5 m and 2.0 m. Therefore, Envisat has a positive deviation with respect to ICESat with mean ice thickness by 2.78 m and 1.92 m, respectively. In autumn Envisat presents a smaller difference with ICESat than in

summer. Envisat shows a large probability during 1.5 m to 2.0 m, leading to a mean thickness of 1.98 m, while ICESat during 1.0 m to 1.5 m and a mean thickness of 1.36 m. Both of the two data sets cannot detect the thin ice smaller than 0.5 m, while Envisat even overlooks all the thin ice smaller than 1.0 m in summer. We find that the mean sea ice thickness of both Envisat and ICESat grow larger from spring to summer with different extent of growth. This anomalous thickness growth can be partly explained by the limited comparison pairs located in the western Weddell Sea, Bellingshausen and Amundsen

Sea, and characterized by thick ice. The rest can be attributed to the uncertainties of both data sets. Envisat's RA-2 altimeter will be affected by weaker penetration into snow depth due to the warm and wet weather condition and flooding sometimes, measuring a larger sea ice freeboard and thus a larger sea ice thickness. Since ICESat has a spatial resolution of approximately 70 m, it is likely to miss most of the thin ice in each grid cell.

Regional differences are necessary to take into account in the Antarctic. Here we make another comparison for each region

and each season as shown in Fig. 9. Due to the limited measurements in the Indian Ocean and western Pacific Ocean, we combine them into the whole Eastern Antarctic. In the western Weddell Sea, the regression lines have large positive intercepts in all three seasons. Table 3 reveals that the largest deviation through all seasons and all regions occurs in the summer western Weddell Sea by 1.03 m. Despite the smallest deviation of 0.02 m happens in spring, the smallest RMSD by 0.69 m and the highest CC by 0.63 happens in autumn. The comparison results perform better in eastern Weddell Sea in

general, with smaller intercepts and increased correlation coefficients compared to western Weddell Sea. In the Eastern Antarctic, the result performs worse in autumn. The largest negative deviation by -0.25 m occurs in spring in this region. Positive deviations take place in autumn Ross Sea with a low correlation by 0.22. As for the summer Bellingshausen and



Amundsen Sea, we cannot derive any correlation between the two data sets due to little comparison pairs. The reasons for these regional differences may deduce from the different surface roughness and different synoptic events. Deformed sea ice and storms can cause biases on both sensors. However, quantity assessments of these impacts are difficult to realize due to lack of enough field observations.

## 4 Potential reasons for the deviations

Although there are uncertainties existing in both Envisat and ICESat data sets, we can still figure out some potential reasons for their sea ice thickness differences. There are two main differences between the two data sets. One is that the different implementation of altimeter sensors to determine surface elevation and freeboard. Envisat is equipped with a radar altimeter (RA-2), whose backscatter is assumed to originate from snow/ice interface, though it is known that this assumption is flawed for snow that is not dry and cold. ICESat instead is equipped with a laser altimeter (GLAS) whose signals are reflected from the air/snow interface with a reasonable level of certainty. In addition, considering the large footprint of radar pulses by about 2–10 km and smaller footprint of laser beams by about 70 m, they are likely to respond differently to different surface roughness. Another one is that they apply different retrieval algorithms to convert freeboard into thickness. Envisat uses the hydrostatic equilibrium with an extra AMSR-E snow depth product while ICESat uses the modified snow–ice density to get rid of the biased snow depth. We will discuss those factors in detail in the following.

### 4.1 Deviations due to different sensors

Theoretically, the dominant backscatter horizon for a Ku-Band radar altimeter is the snow/ice interface for cold and dry snow (Beaven et al., 1995). However, this would not always be the case in the Antarctic according to the field investigations conducted by Willatt et al. (2010). They demonstrate that the dominant scattering surface of the Ku band radar lies within the snowpack, usually half of the mean snow depth, when the snow cover is not cold and dry. Wet conditions can affect the dielectric properties of snow and then weaken the penetration of radar altimeter signals into the snow–ice interface. In other words, RA-2 measurements could consist of the sea ice freeboard and an uncertain part of the snow depth when morphological processes occur. Consequently, it will lead to a larger sea ice freeboard and larger sea ice thickness.

Figure 10 presents the seasonal average differences between Envisat and ICESat sea ice thickness for the three seasons. Apparently, the overall difference between Envisat and ICESat is the smallest in spring, with smaller positive deviations in the Weddell Sea and Ross Sea compared to summer and autumn. We speculate that this may result from the drier weather conditions. Therefore, we examine the average relative humidity of each season as Fig. 11 shows. Generally, in the regions that the relative humidity is high, Envisat tends to present positive deviations. Especially in autumn, the two maps seem to match with each other well. Notable negative deviations in the western Pacific sector still exist in autumn, but the degree is lighter than in spring, which can also be explained with the snow wetness change. Therefore, it is necessary to consider the humidity impacts during the freeboard retrieval. However, model experiments still need to be done to quantify such impacts.



In addition to the different penetration depth, the two sensors have different footprints. Since the pulse-limited radar
altimeter footprint is rather rough, basically larger than 2 km, the reflection from the leads can affect the detection of shorter
and thinner ice of Envisat (Tilling et al., 2019). Therefore, Envisat tends to detect wider and thicker sea ice leading to higher
freeboard in the first-year ice (Schwegmann et al., 2016; Tilling et al., 2019). The preference for wider sea ice floes may
cause overestimation of ice thickness compared with the laser altimeter loaded on ICESat. Consequently, we assume that
part of the positive deviations comes from the biases of radar altimeters large footprint.

## 4.2 Deviations due to different retrieval algorithms

According to the retrieval algorithms applied by the two data sets described in Sect. 2.2, the largest uncertainties may come
from the biased AMSR-E snow depth. AMSR-E snow depth used in the retrieval is retrieved from brightness temperature
based on the linear relation between brightness temperatures and in situ observations (Comiso et al., 2003). Based on this
method, AMSR-E has a consistent underestimation of the deep snow over perennial ice and is limited to the maximal
retrieval value being around 50 cm because of the similar radiometric signature of deep snow and multiyear ice (Comiso et
al., 2003). Previous study reveals that AMSR-E snow depth tends to considerably underestimates the actual value over
deformed sea ice, which usually occurs in the eastern Antarctic (Worby et al., 2008b; Ozsoy-Cicek et al., 2011).
Environmental conditions have great effects on the snow physical properties such as density, wetness and salinity. Previous
research indicates that the satellite passive microwave snow depth is sensitive to ice concentration errors, weather effects,
grain size, thaw and refreezing (Markus and Cavalieri, 1998). Especially, wet snow caused by melt or flooding could lead to
underestimations while refreezing of molten snow could lead to overestimations. All of the above biases can also cause the
differences between Envisat and ICESat.

In spring, large negative deviations are observed in the western Weddell Sea, the Bellingshausen Sea, the western Pacific
Ocean and the Indian Ocean, mainly along the coast, as shown in Fig. 10. Perennial ice with significant deformation exists in
the western Weddell Sea and the Bellingshausen Sea, which could result in underestimations of snow depth. Therefore, the
sea ice thickness deviation presents negative. For the Pacific and the Indian sector, we also consider the deviations coming
from the snow depth uncertainty over the deformed sea ice.

## 5 Conclusions and discussions

A new set of Antarctic sea ice thickness derived from Envisat RA-2 is compared based on ICESat sea ice thickness retrieved
from modified ice density algorithm. The absolute accuracy of Envisat and ICESat sea ice thickness has been assessed with
ULS in Weddell Sea at first and the results show that ICESat measurements are in better agreement with the ULS sea ice
thickness. Then a systematic comparison between the two data sets has been conducted. The comparison is carried for three
seasons except winter based on the ICESat operating periods. According to the results, the differences between Envisat and
ICESat sea ice thickness are different considering different seasons, years and regions. More specifically, the smallest



average difference in spring by -0.03 m while having larger difference in summer and autumn by 0.86 m and 0.62 m, respectively. Although the smallest absolute difference occurs in spring 2005 by 0.02 m, the largest correlation coefficient appears in autumn 2004 by 0.77. Probability distribution shows a mean ice thickness growth for both Envisat and ICESat, potentially due to the limited comparison pairs most of which are thick ice and the uncertainties of both data sets. The largest positive deviation occurs in the western Weddell Sea by 1.03 m in summer while the largest negative deviation occurs in the Eastern Antarctic by -0.25 m in spring. Potential reasons for those deviations mainly deduce from the limitations of Envisat radar altimeter affected by the weather conditions and the surface roughness as well as the different retrieval algorithms. The better performance in spring of Envisat has a potential relation with relative humidity.

Although we discuss the possible biases of Envisat and AMSR-E chiefly in this study, we should keep in mind that results from ICESat have their uncertainties. ICESat freeboard can be biased at the locations where the geoid or the sea surface height are inaccurate or where the elevation measurements are biased affected by ocean swell (Kern et al., 2016). In addition, the modified density used in the Worby retrieval algorithm does not consider the small-scale or regional variability of the snow depth, instead, only a seasonal constant density derived from the ASPeCt observations is given. Therefore, the largest uncertainty of ICESat comes from the potentially underestimations of the sea ice and snow observations for the computation of density (Kern et al., 2016). However, without sufficient data and further analysis, it is difficult to quantify these uncertainties.

Through the study, we know that there are a lot of deviations between Envisat and ICESat sea ice thickness, and those deviations potentially result from the uncertainties of both data sets. However, without enough observation data and numerical model experiments we cannot quantify the impacts of the uncertainties over the sea ice thickness. There is still more work to be done in the future to make better use of it such as assimilating the Antarctic sea ice thickness observations and analyzing the sea ice volume variations.

*Data availability.* The Envisat sea ice thickness are available at https://dx.doi.org/10.5285/b1f1ac03077b4aa784c5a413a2210bf5 (Hendricks et al., 2018b). The ICESat-1 sea ice thickness data are available at http://icdc.cen.uni-hamburg.de/1/projekte/esa-cci-sea-ice-ecv0/esa-cci-data-access-form-antarctic-sea-ice-thickness.html (Kern et al., 2016). The Weddell Sea upward looking sonar data are available at http://doi.pangaea.de/10.1594/PANGAEA.785565 (Behrendt et al., 2013a; Behrendt et al., 2013b). The relative humidity data are available at https://www.ecmwf.int/en/forecasts/datasets/reanalysis-datasets/era-interim.

*Author contributions.* JW, QY and QS developed the concept of the paper. JW analyzed the data and wrote the manuscript. CM, RR, QY, QS, BH and SH assisted during the writing process.

*Competing interests.* The authors declare that they have no conflict of interest.



*Acknowledgments.* This study is supported by the National Natural Science Foundation of China (No. 41941009, 41922044), the Fundamental Research Funds for the Central Universities (No. 19lgzd07). The authors would like to thank Alfred Wegener Institute Helmholtz Centre for Polar and Marine Research for providing the Weddell Sea upward looking sonar data, the Integrated Climate Data Center at the University of Hamburg for providing the ICESat-1 sea ice thickness data and European Centre for Medium-Range Weather Forecasts for providing ERA-interim relative humidity data.

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



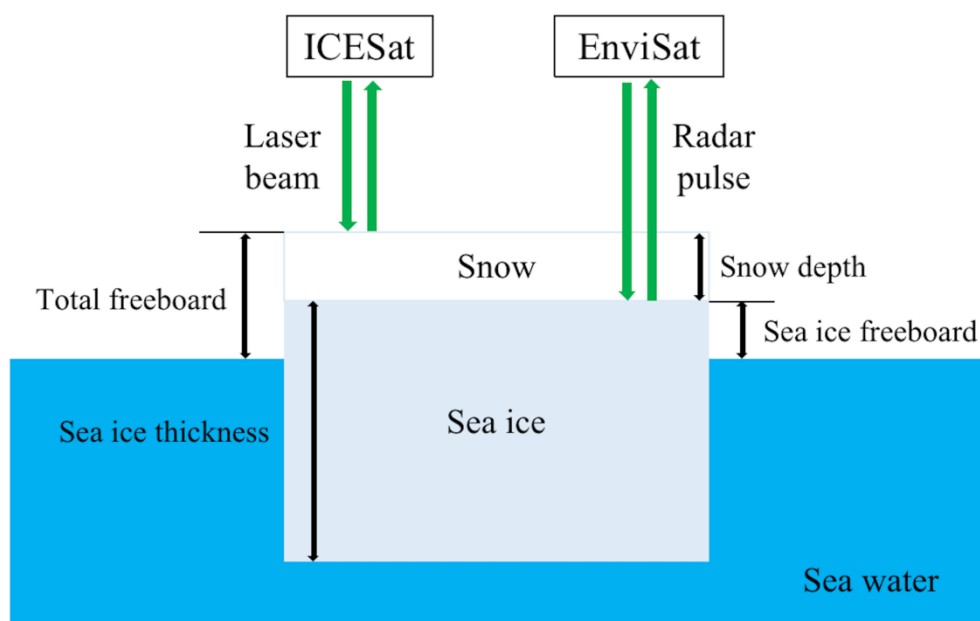

**Figure 1:** An illustration of measuring sea ice freeboard using ICESat and Envisat.

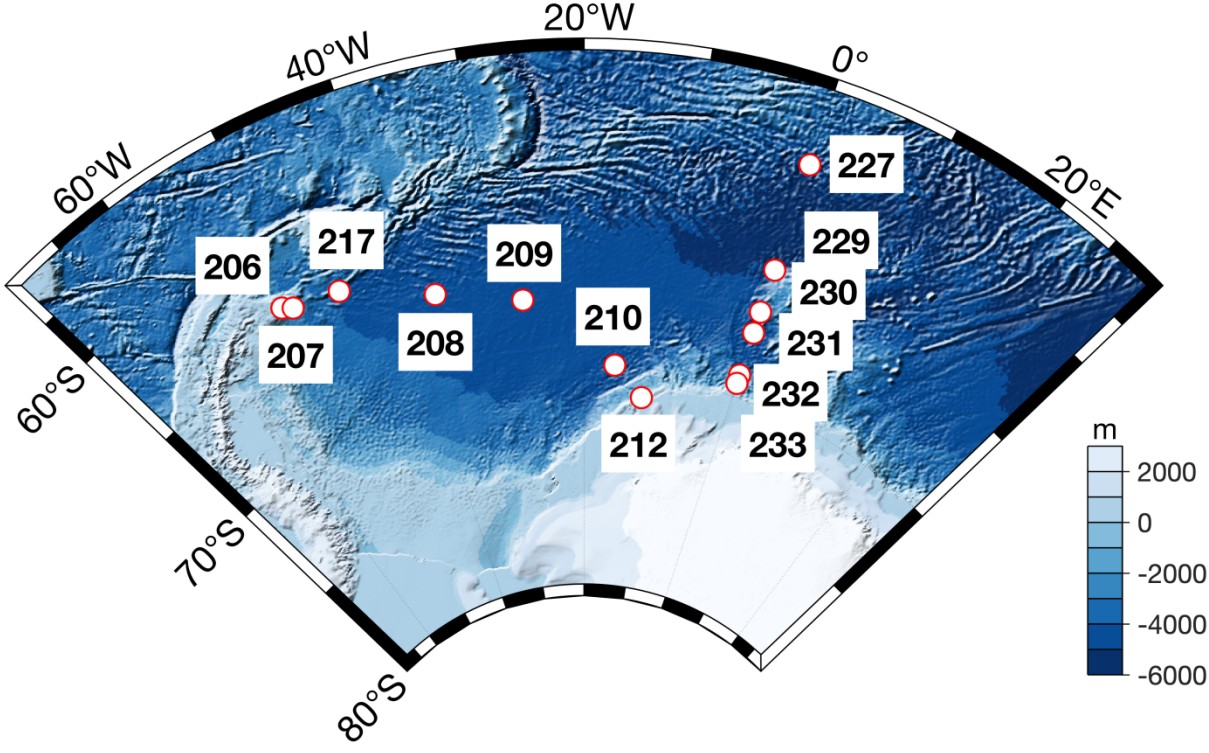



**Figure 2:** Map of the AWI ULS mooring locations.

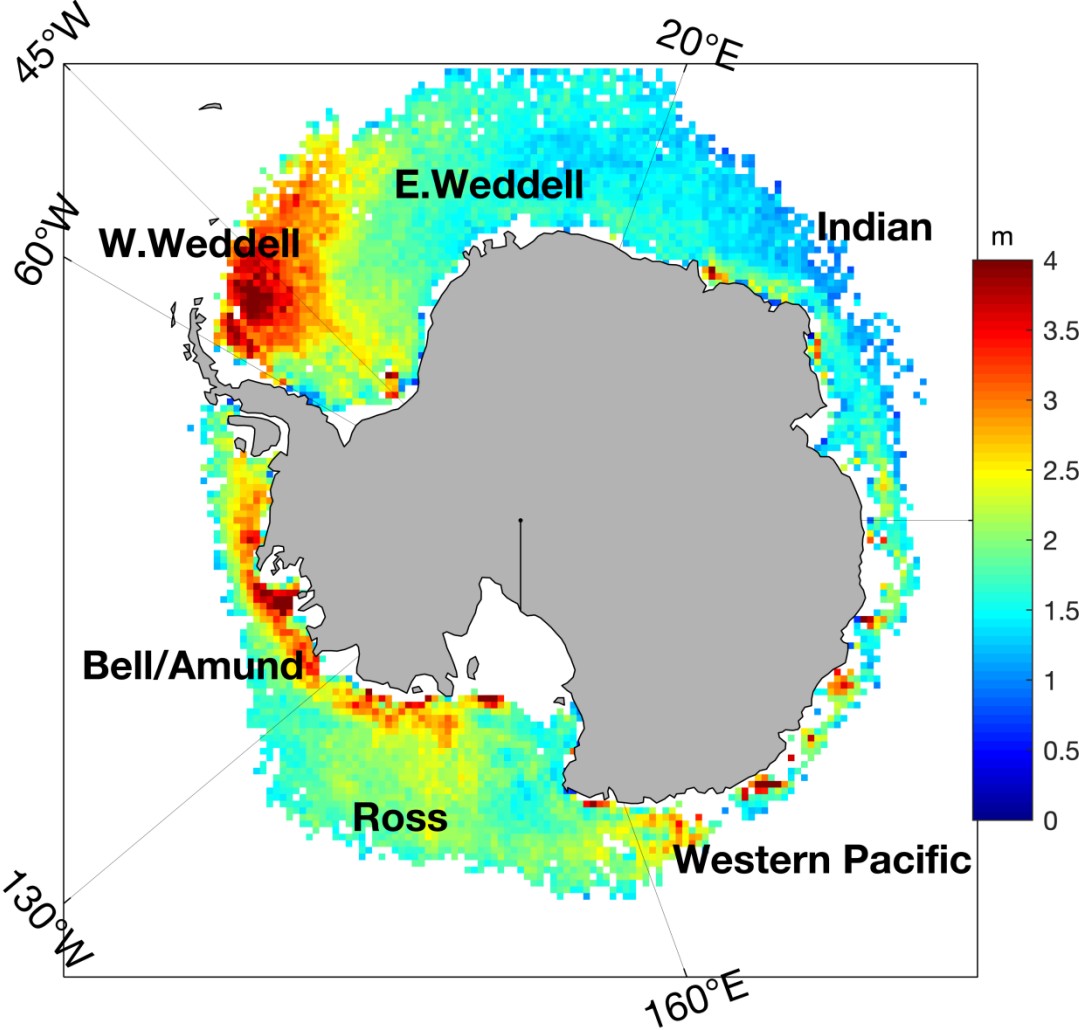

**Figure 3:** Map of the different sectors referred to in the study. The background is the average of the September sea ice thickness from Envisat during 2003-2011.







**Figure 4:** Time series of sea ice thickness for the Weddell Sea ULS data, Envisat and ICESat. The numbers on the top represent the location and period of each site for the comparisons. The site locations can be searched in Fig.2.

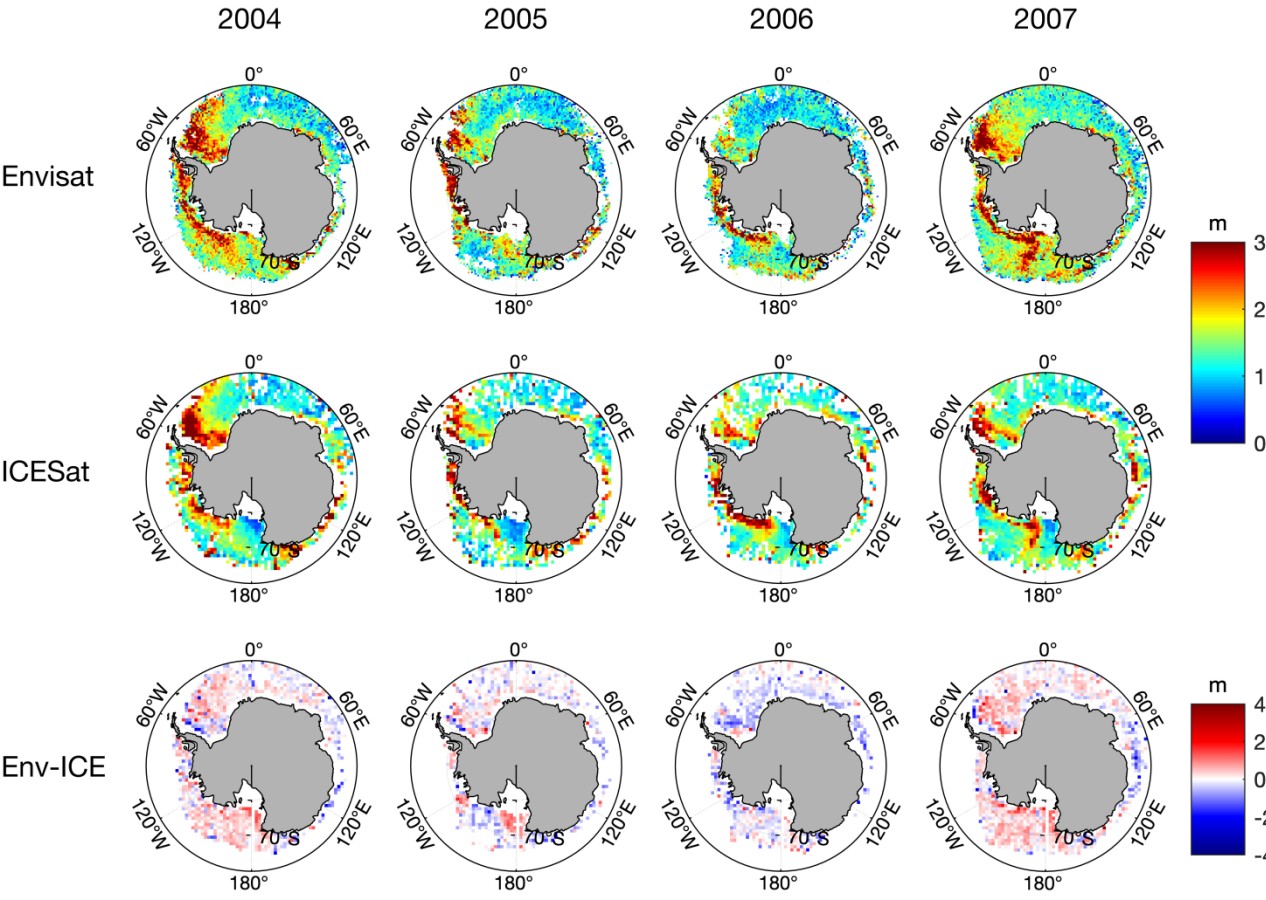

**Figure 5:** Comparisons of Envisat versus ICESat sea ice thickness for each ICESat operating period in spring (October & November). The first and second row show the sea ice thickness distribution of Envisat and ICESat respectively, and the last row show the deviation map (Envisat minus ICESat) of sea ice thickness. Each column represents a year from 2004 to 2007.





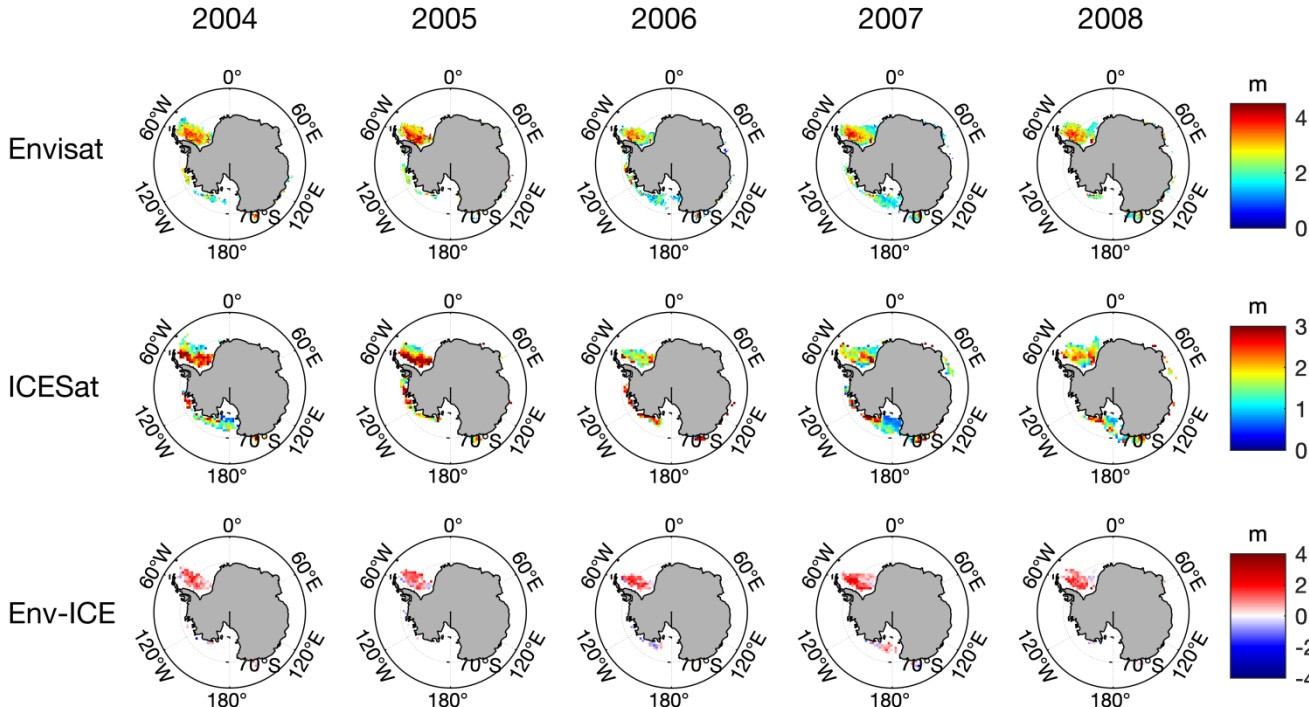

**Figure 6:** Same as Fig. 5 but for summer (February & March).





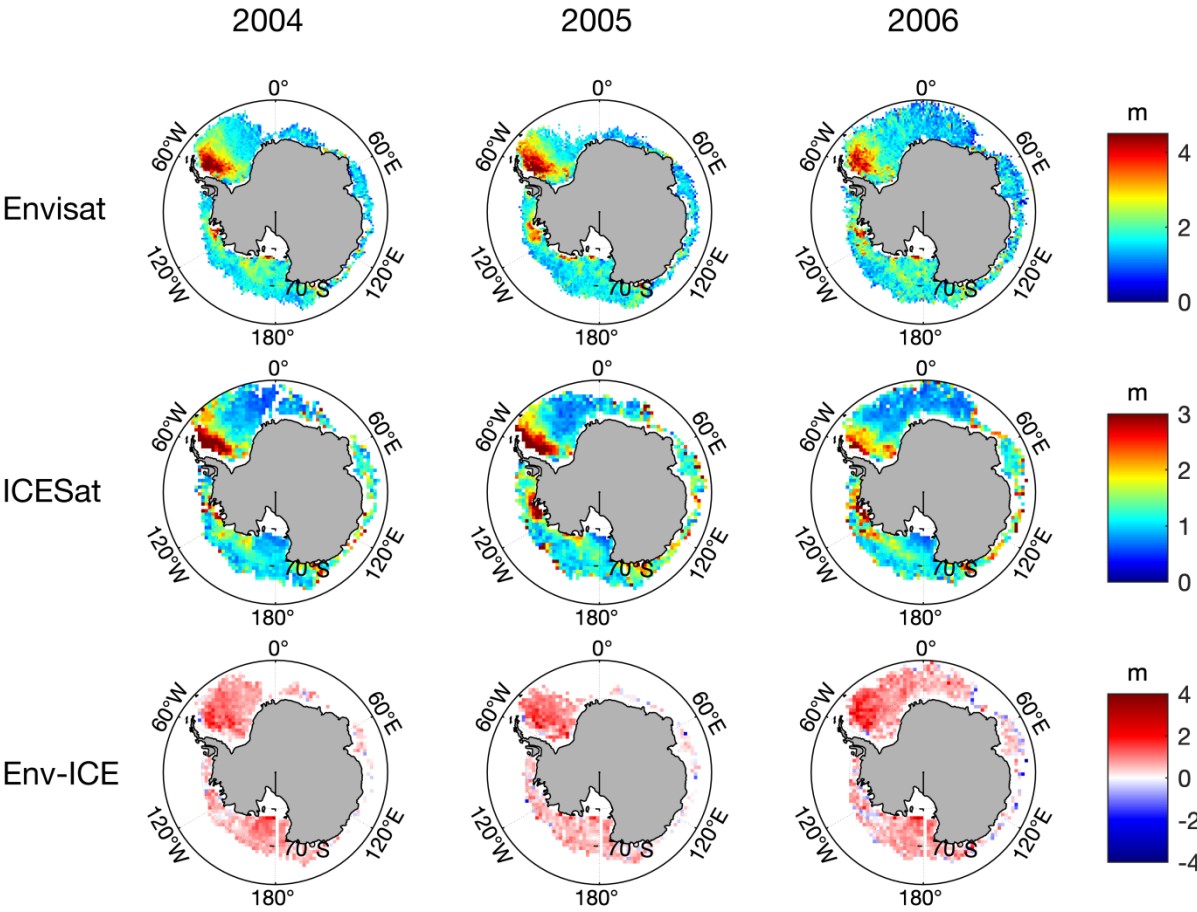

**Figure 7:** Same as Fig. 5 but for autumn (May & June).





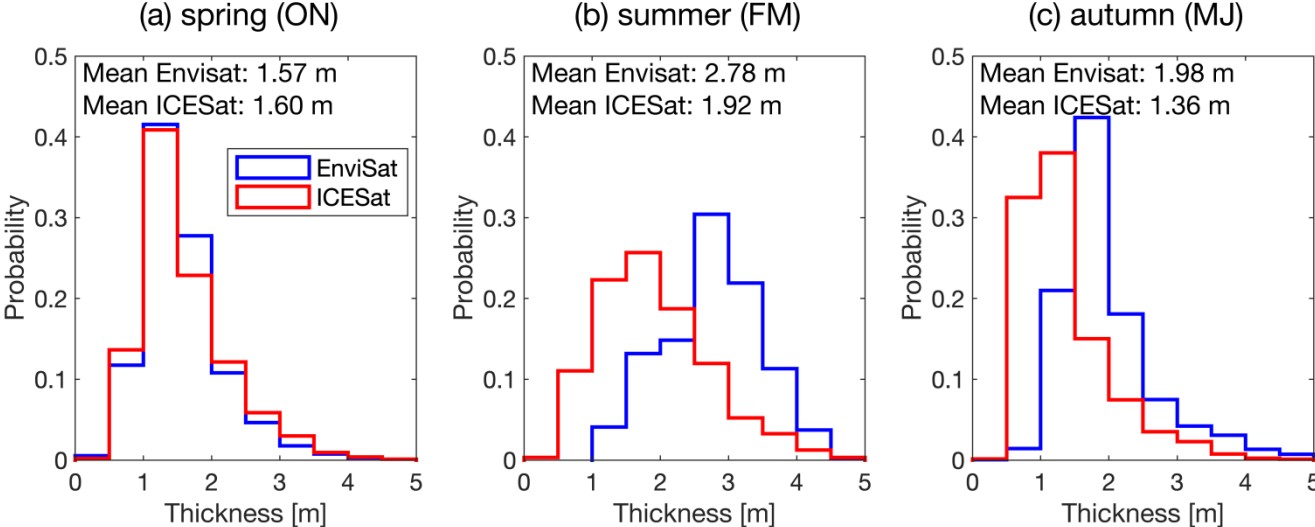

**Figure 8:** Probability of the Envisat SIT and the ICESat SIT for all the individual comparison pairs. The blue lines represent Envisat ice thickness and the red line represent ICESat ice thickness.

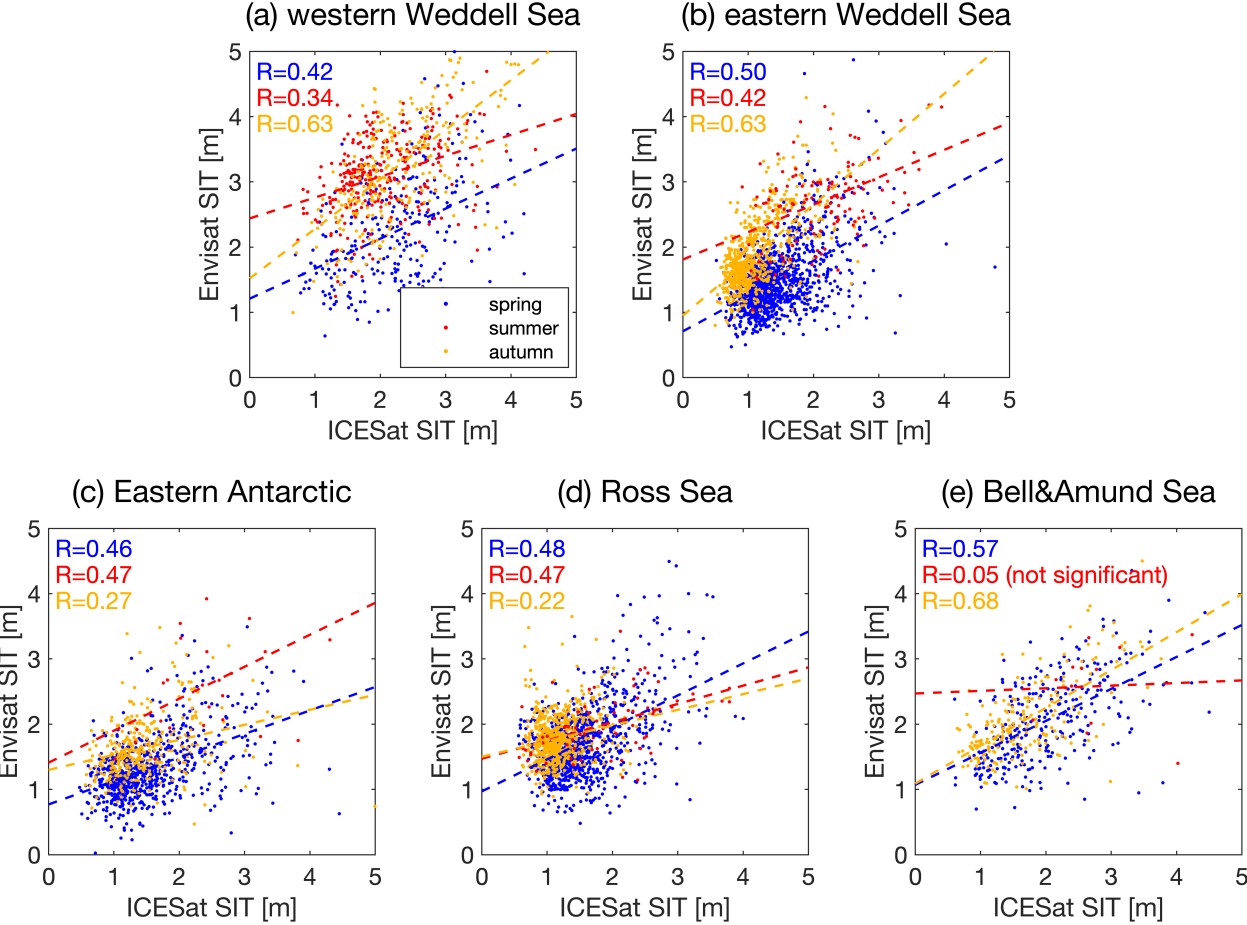

**Figure 9:** Scatterplots of the individual data pairs between Envisat SIT and ICESat SIT for each region and each season. Since the comparison pairs are too few in Indian Ocean and western Pacific Ocean, we combine these two regions into Eastern Antarctic.



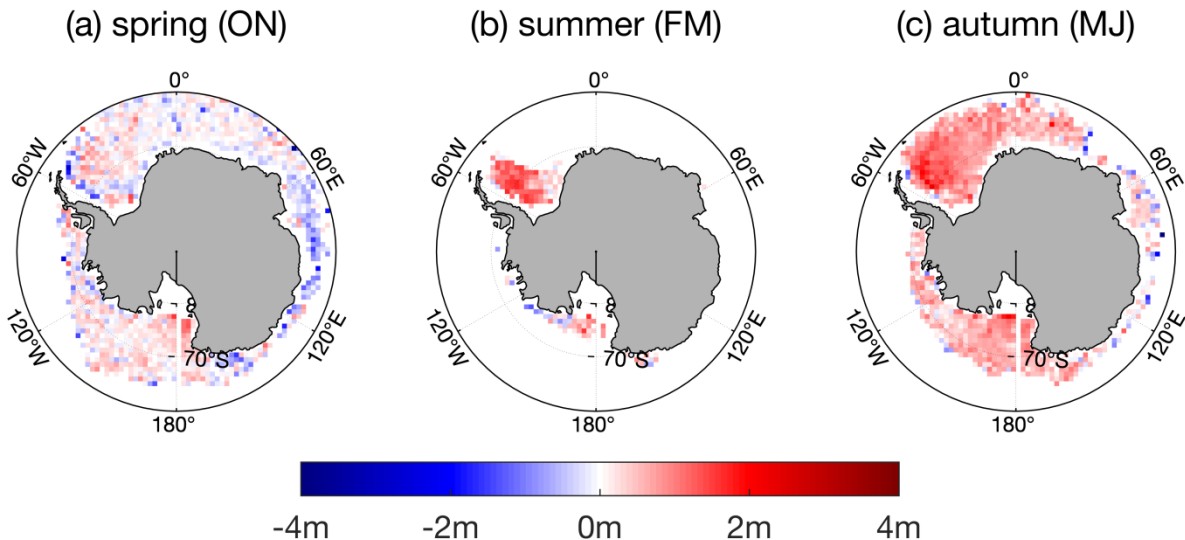

**Figure 10:** Seasonal average differences between Envisat and ICESat sea ice thickness for the three seasons.

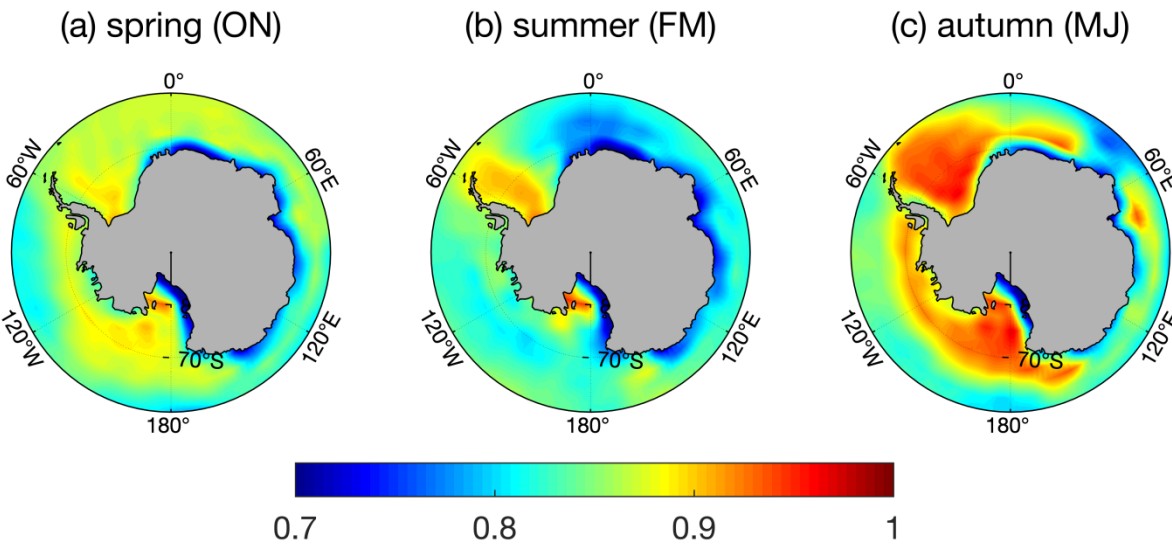

**Figure 11:** Seasonal average relative humidity distributions for the three seasons.





**Table 1:** ICESat operating periods and Envisat periods used for the comparisons. The three seasons are divided according to the ICESat operating periods.

| Years | Dates | | | | | |
|---|---|---|---|---|---|---|
| | Spring (ON) | | Summer (FM) | | Autumn (MJ) | |
| | ICESat | Envisat | ICESat | Envisat | ICESat | Envisat |
| 2004 | Oct 3 to Nov 8 | Oct 1 to Oct 31 | Feb 17 to Mar 20 | Feb 1 to Mar 31 | May 18 to Jun 20 | May 1 to Jun 30 |
| 2005 | Oct 21 to Nov 23 | Nov 1 to Nov 30 | Feb 17 to Mar 22 | Feb 1 to Mar 31 | May 20 to Jun 22 | May 1 to Jun 30 |
| 2006 | Oct 25 to Nov 26 | Nov 1 to Nov 30 | Feb 22 to Mar 26 | Mar 1 to Mar 31 | May 24 to Jun 25 | Jun 1 to Jun 30 |
| 2007 | Oct 2 to Nov 4 | Oct 1 to Oct 31 | Mar 12 to Apr 14 | Mar 1 to Apr 30 | – | – |
| 2008 | – | – | Feb 17 to Mar 20 | Feb 1 to Mar 31 | – | – |

**Table 2:** Statistical results of the comparisons between Envisat SIT and ICESat SIT for each ICESat operating period. The correlation coefficients in italic type have not passed the 95% significance test.

| | | ENV (m) | ICE (m) | Difference (m) | RMSD (m) | CC | N |
|---|---|---|---|---|---|---|---|
| Spring (ON) | average | 1.57 | 1.60 | -0.03 | 0.59 | 0.54 | 3275 |
| | 2004 | 1.66 | 1.64 | 0.02 | 0.58 | 0.59 | 938 |
| | 2005 | 1.48 | 1.54 | -0.06 | 0.61 | 0.50 | 643 |
| | 2006 | 1.39 | 1.63 | -0.24 | 0.54 | 0.60 | 631 |
| | 2007 | 1.67 | 1.57 | 0.10 | 0.57 | 0.55 | 1063 |
| Summer (FM) | average | 2.78 | 1.92 | 0.86 | 0.81 | 0.34 | 500 |
| | 2004 | 2.89 | 1.91 | 0.98 | 0.74 | 0.32 | 102 |
| | 2005 | 3.20 | 2.31 | 0.89 | 0.77 | 0.33 | 84 |
| | 2006 | 2.55 | 2.10 | 0.45 | 0.90 | *0.09* | 85 |
| | 2007 | 2.57 | 1.69 | 0.88 | 0.80 | 0.34 | 134 |
| | 2008 | 2.80 | 1.82 | 0.98 | 0.80 | *0.08* | 95 |
| Autumn (MJ) | average | 1.98 | 1.36 | 0.62 | 0.56 | 0.66 | 2021 |
| | 2004 | 2.02 | 1.33 | 0.69 | 0.46 | 0.77 | 609 |
| | 2005 | 2.07 | 1.43 | 0.64 | 0.55 | 0.74 | 571 |
| | 2006 | 1.88 | 1.32 | 0.56 | 0.62 | 0.46 | 841 |



**Table 3:** Statistical results of the comparisons between Envisat and ICESat SIT for each region divided as Fig. 9. The correlation coefficients in italic type have not passed the 95% significance test.

|  |  | ENV (m) | ICE (m) | Difference (m) | RMSD | CC | N |
|---|---|---|---|---|---|---|---|
| W. Weddell | spring (ON) | 2.24 | 2.22 | 0.02 | 0.87 | 0.42 | 267 |
|  | summer (FM) | 3.06 | 2.03 | 1.03 | 0.70 | 0.34 | 293 |
|  | autumn (MJ) | 3.21 | 2.28 | 0.93 | 0.69 | 0.63 | 247 |
| E. Weddell | spring (ON) | 1.44 | 1.47 | -0.03 | 0.49 | 0.50 | 1123 |
|  | summer (FM) | 2.70 | 1.84 | 0.86 | 0.70 | 0.42 | 129 |
|  | autumn (MJ) | 1.87 | 1.08 | 0.79 | 0.42 | 0.63 | 652 |
| Eastern Antarctic | spring (ON) | 1.30 | 1.55 | -0.25 | 0.60 | 0.46 | 701 |
|  | summer (FM) | 2.66 | 2.35 | 0.31 | 0.86 | 0.47 | 25 |
|  | autumn (MJ) | 1.64 | 1.50 | 0.14 | 0.62 | 0.27 | 321 |
| Ross Sea | spring (ON) | 1.70 | 1.51 | 0.19 | 0.54 | 0.48 | 940 |
|  | summer (FM) | 1.94 | 1.59 | 0.35 | 0.68 | 0.47 | 72 |
|  | autumn (MJ) | 1.78 | 1.19 | 0.59 | 0.42 | 0.22 | 636 |
| Bell/Amund | spring (ON) | 2.06 | 1.99 | 0.07 | 0.65 | 0.57 | 282 |
|  | summer (FM) | 2.54 | 2.22 | 0.32 | 0.83 | *0.05* | 14 |
|  | autumn (MJ) | 2.10 | 1.64 | 0.46 | 0.52 | 0.68 | 199 |