# Peer review of "A comparison between Envisat and ICESat sea ice thickness in the Antarctic"

_The Cryosphere, 2020_

## Referee Comment (RC1) · Anonymous Referee #1 · 31 Mar 2020

Review of

A comparison between Envisat and ICESat sea ice thickness in the Antarctic

by Wang, Jinfei, et al.

Summary: This paper is about a study inter-comparing two Antarctic sea-ice thickness products derived from satellite altimetry. One is using Envisat radar altimetry; the other one is based on ICESat laser altimetry. The comparison focusses on years 2004 through 2008, being limited to the periods during which ICESat's laser was operational. The two products are first compared to sea-ice thickness values derived from sea-ice draft observation of a few upward looking sonars (ULS) in the Weddell Sea. This comparison suggests a better agreement between ICESat and ULS sea-ice thickness.

The core of the paper deals with the inter-comparison between the two satellite-based sea-ice thickness products using standard statistical metrics such as mean, difference, RMSD and correlation. The study shows maps, histograms and scatterplots and summarizes results in two tables. The comparison is carried out for Antarctica as a whole but also broken down to the commonly used sub-regions. The results suggest as season-dependent over-estimation of ICESat sea ice thickness by the Envisat product, an over-estimation which is particularly pronounced during February/March and May/June. The observed differences between the products are discussed in the light of potential causes.

General Comments: Continuing to inter-compare the ESA SICCI-2 sea-ice thickness data sets with independent sea-ice thickness observations to better evaluate these products certainly is very useful, contributing to our understanding of the uncertainties and limitations of these. Since a more focus has been on inter-comparison studies for the CryoSat-2 part of these data sets, this study is particularly welcome because it focusses on the Envisat period. Sea-ice thickness observations from that period (before March 2011) are notoriously sparse. The authors solved this by using upward looking sonar draft observations and a sea-ice thickness data set based satellite-based laser altimetry - which appears close to the best one can do and nicely expands on existing work carried out in the framework of the ESA SICCI-2 project. The manuscript would benefit from a number of substantial amendments to figures and text, potentially making the results more credible. The most important points I laid out in my general comments GC2 and GC3 below, accompanied by a suite of specific comments detailing the potential measures that could lead to an improved credibility of the results and to a mitigation of their potential misinterpretation. It is recommended that all co-authors adequately contribute with their expert knowledge to the manuscript.

GC1 (editoral): Please be consistent with usage of total freeboard and sea-ice freeboard. Try to be as clear as possible in the way you assign the respective wordings to the two different types of instruments used.

GC2: The comparison with the ULS data requires more details and a slight change in the way the comparison is presented. Currently the message to be taken is based on very few data points with limited discussion of the reliability of such an inter-comparison. This bears the risk to over-interpret the results which to my opinion should be avoided. Please see my specific comments.

GC3: The discussion of the results of the comparison between Envisat and ICESat SIT data sets (i.e. Section 4) would benefit from some more considerations and re-writing. This applies to the quality of the ICESat data considered as "the truth", to the way the statistical results are interpreted, and to some details in the description how the comparison was carried out. Please see the respective specific comments.

Specific Comments: Abstract: - I recommend to include number of the comparison results between the two altimeter sea-ice thickness data sets and the ULS draft data set, i.e. something like the mean difference and its standard deviation. It would also be helpful to learn from the abstract rightaway whether you transferred the ULS draft measurements into sea-ice thickness or whether you converted the satellite data into sea ice draft by simply saying whether you compared draft or sea-ice thickness. Why? Because any conversion (draft to thickness or thickness to draft) has inherent errors. - I recommend to focus on the one parameter, e.g. the mean difference and its standard deviation, and report this for the entire Antarctic and all sub-regions. This gives a more complete picture of your intercomparison than the bits and pieces of smallest deviations, largest correlations, largest deviations, etc. you do provide currently.

Lines 63-64: - "Hendricks et al 2010" –> I guess there is more recent literature in peer-reviewed journals targeting this issue and even using for improvement of current retrievals? Please check. - "Ku-Band radar altimeters" –> Is this a special feature of Ku-Band or perhaps generally valid for radar altimeters? Please change accordingly if needed.

Lines 65-67: - There is more recent literature about this issue - perhaps Kern and

Ozsoy-Cicek 2016: Satellite Remote Sensing of Snow Depth on Antarctic Sea Ice: An Inter-Comparison of Two Empirical Approaches; or Kern and Ozsoy 2019: An attempt to improve snow depth retrieval using satellite microwave radiometry for rough antarctic sea ice; or Price et al. 2019: Snow-driven uncertainty in CryoSat-2-derived Antarctic sea ice thickness - insights from McMurdo Sound - "must use" –> no, not necessarily. There are other alternatives to deal with snow, e.g. using snow accumulation from re-analyses.

Line 77: Please already here add one sentence about ICESat-2 which is in orbit since September 2018 and provides excellent freeboard measurements which are not discontinuous in time anymore.

Line 85: This approach was not suggested by Worby. It was suggested by Kern et al. 2016. This approach has been called 1-layer Worby et al approach in Kern et al. (2016) because the ratio between snow depth and sea-ice thickness used in Eq. 2 is based on the ASPeCt observations published by Worby et al. (2008); Please correct accordingly. Note that this approach has been further refined by Li et al. 2018 (see comment under "typos/editoral issues" Lines 50-52), who derived first guess values of snow depth and sea-ice thickness directly from ICESat data using empricial approaches instead of the climatological values used by Kern et al. (2016); that way the sea-ice thickness retrieval using this 1-layer method is certainly considerably more accurate than the one introduced for the first time in Kern et al. (2016) because it takes the actual situation into account better. Of course, it needs to be noted that the empirical approaches used by Li et al. (2018) were in turn developed from a (more or less comprehensive) suite of historic in-situ observations of freeboard, snow depth and sea-ice thickness which in a way have the character of a climatology as well.

Lines 98-107: - The last paragraph of this block about Envisat should come first and it should be expanded. Please provide a bit more information about the Envisat RA2 (time of operation, orbit details (observation hole at pole?), nominal resolution, altimeter type etc.). Then you could come up with something along the lines how sea-ice

freeboard is obtained and then how from the sea-ice freeboard sea-ice thickness is derived. I suggest to avoid this "is defined" in Line 99; I don't think there is a definition. What the RA2 measures is a runtime which is converted into a surface elevation which in turn is converted into a radar freeboard in case of sea ice. For this freeboard it is not precisely known whether this is representative of the location of the ice-snow interface. It is assumed that this is the case based on the experience from laboratory work (you cited it, Beaven et al.) and because of this snow-depth dependent radar signal delay is applied to convert the radar freeboard into what is assumed (not defined) the sea-ice freeboard. - Line 101: Kern et al. (2016) is certainly not the correct reference here. That paper is about ICESat and has nothing to do with radar altimetry. You hence need to find another paper. Since you have at least two radar altimeter experts, which were involved in the derivation of the Envisat data set you are using, among your co-authors it would be easy to find an appropriate reference. Just ask them! - Equation 1: Please explain in the text what the rhos are. - Lines 103/104: I am sure the information about how the sea-ice thickness is retrieved is given in the respective ATBD of the data set; again I recommend to ask the two people who were actively involved in the processing of the Envisat data set. You will need this information also for answering which sea-ice concentration product is included in the data set (Line 110) and for figuring out what the sea-ice concentration threshold is (anyways) which was used to derive freeboard and sea-ice thickness (Line 108).

Line 131: 60%: Why do you use a different sea-ice concentration treshold for the ICESat data than you use for the Envisat data?

Lines 133-139: - I suggest to add a bit more information about i) the field of view of such a sensor; ii) the temporal resolution with which the data are provided and which you use for your comparison; iii) the accuracy. - I would not group ULS data into "in-situ" (Line 135) because also ULS technique is kind of remote sensing and not an in-situ measurement such as a drilling of snow stake measurement would be. - The sentence containing "the most suitable criterion to evaluate" should possibly be rewritten along

the lines that such data are particularly well suited to evaluate the temporal evolution of the sea-ice thickness at a fixed location. How well their accuracy is (suited) we don't know (yet). And such data are not overly well suited to evaluate the spatial distribution of the sea ice thickness. Also, when using ULS data sea-ice motion as to be taken into account. - What are the assumption behind the empirical equation 4 - or in other words: Where lie its limitations of applicability / inherent uncertainties?

Lines 141-148: "humidity" The amount of humidity in the atmosphere is certainly a impacting the quality of the Envisat RA2 measurements but is to my knowledge taken into account by the on-board microwave radiometer being sensitive to the atmospheric water vapor load. While I would not exclude that the humidity of the atmosphere could also have an impact on snow wetness or the available of free (liquid) water within the snow I an inclined to say that its effect is much smaller than the direct influences of both the air temperature / surface temperature and the net surface radiation flux. These parameters directly determine snow melt - be at the surface or internally, i.e. within the snow. Above a surface or snow temperature of -5degC the liquid water fraction in the snow increases. I would therefore suggest to look for air or surface temperature rather than humidity and also look at the net surface radiation. The relative humidity is a derived, secondary quantity which is possibly also more prone to errors than the temperature itself.

Line 151-153: - "For each period ... If ICESat data has overlapping time ..." –> Perhaps better: "We use Envisat data of every month which overlaps for at least ten days with a particular ICESat measurement period. For some ICESat measurement period this requires to average over Envisat sea-ice thickness data of two months (see Table 1)." - I note that period ON2005 starts October 21 and hence has 11 days in October with data while Envisat data are only taken from November. Please fix.

Figure 1: I suggest to replace "sea ice freeboard" with "freeboard" in the caption of this figure to underline that you are targeting to illustrate both, total freeboard and sea-ice freeboard.

Figure 2: - Did you create this map on your own or is this taken from the publication / data website? If the latter you either need to get the permission or at least state that you modified if from ... - What are the numbers in the white boxes referring to? - Please provide a table with the respective operation times of the ULS which overlapped with your combined ICESat - Envisat sea-ice thickness data set. If I recall correctly not all of these were in operation all the time. - Please write what the background information is ... it is bathymetry.

Figure 3: - The lines denoting the borders between the different regions is hard to see and should be made thicker or plotted using a different color. - Specify what the white grid cells stand for.

Lines 159-166: - How were ULS data co-located with the satellite data products? Also, I note that Envisat is on EASE grid but ICESat is on polar-stereographic grid. Which one did you choose? - What do you mean by "absolute" in Line 159? - "in situ" –> ULS is not an in situ measurements. - What does "-6" "-5" appended behind the ULS station numbers mean? - I don't think "deformed ice" and "thin ice" are particularly well chosen because also thin ice can be deformed. It might be better to refer to old or perennial ice in case of the location close to the peninsula - albeit local polynyas might produce an applicable amount of seasonal ice as well - and predominantly seasonal ice in case of the Eastern Weddell Sea.

Lines 167-181 / Figure 4: - It appears to me that the number of ICESat values used in this comparison are so small that it is almost impossible to provide an adequate statistics. In order to make the comparison look at bit more reasonable I suggest to combine the two successive records for 229 and 231, i.e. plot time series for the combined data set 229-5 and 229-6 as well as 231-6 and 231-7. That way you would not be forced to draw conclusions from just two data pairs. In addition, it appears to me that you computed RMSD values based on this completely different statistics and I suggest that you add a comparison between ULS data and Envisat data at the same sampling in time provided by ICESat. In that case you would need to specify

whether you'd average over May and June in the Envisat product to obtain a value representing the ICESat May/June measurement periods. - Please adequately note in the text that you take uncertainty information from the data sets, i.e. that both data sets, the ICESat and the Envisat one provide uncertainty information based on the retrieval. In this context you need to take into account that the uncertainty of the Envisat is possibly reduced by a factor of 1/sqrt(n) with n being the number of grid cells used to compute the 100km mean gridded version of the Envisat sea-ice thickness. - I suggest to rewrite the full paragraph once you have this third set of more sparsely sampled Envisat data. Please when re-writing it provide systematically the information about the range of differences and the RMSD for all data sets, i.e. ICESat - ULS, Envisat - ULS, Envisat_sparse - ULS for 207, 229 5+6, 231 6+7 - ideally in a table. I would not over-interprete grid cells with "open water" in the satellite products because values might simply be missing due to other factors. In your discussion of these results you should remind the reader that ULS is a measurement device obtaining very local data of sea ice which passes / develops over it - in your case - over the course of a month. My question is: how accurately did you take the exact measurement period of ICESat into account here? Did you average over the, e.g. 35 days of a period or did you average over the calendar months? In addition: you compare one sea-ice thickness value from a 100 km x 100 km large grid cell with a value representative of a footprint of several 10 meters (I assume). This has an influence on both the sea-ice thickness as well as your estimate of the open water fraction obtained to carry out the comparison using the grid-cell mean sea-ice thickness.

Figures 5 to 7: - Please consider extending the latitudinal range further to the north so that you do not truncate the ice-covered area. - Please consider to show all maps with the same size. Currently, the maps for autumn are largest while those for summer are smallest. Having three maps in a row (as in Figure 7) seems to be a viable solution; hence, if you transpose rows to columns and show the years in rows rather than columns you can realize my suggestion. - I note that you show the two sea-ice thickness data sets at their native grid resolution while the difference map (of course)

is in the grid resolution of the ICESat product. Please make a note in the caption of Figure 5 to clarify that this is done on purpose. - I assume that white denotes missing data? This is fine (as long as stated once in the caption) for the sea-ice thickness maps but interferes with grid cells having a valid sea-ice thickness value in both data sets but a difference close to zero; in the difference maps the color white is not used uniquely - which I suggest to change. - I note a vertical line of grid cells without valid values just west of the date line, i.e. 180°E. Please consider removing this / filling it with valid difference values. - Cosmetics: The color bar would be sufficient with half its horizontal width. Am I correct in assuming that the quantity displayed is sea-ice thickness or sea-ice thickness difference? Consider you want to uses these figures in a presentation - it cannot go without the caption the way you solved it now. If you would state along with the color bar which physical quantity is shown the figures would be self-explainable. You could combine this information with the information about the product which currently is provided left of the maps.

Table 2: - Please consider adding the standard deviation of the mean for the columns denoted "ENV (m)" and "ICE (m)". Ideally, you do this after the interpolation of the Envisat data to the ICESat resolution.

Lines 186-202: - Line 186: Delete "And" - Line 187: Who is "They"? - Line 189-191: I don't find the spring maps clear enough to make the statement given about the Ronne Ice Shelf polynya. The other issue, that Envisat kind of fails to see the young in in the southern Ross Sea is something which is noted in the above-mentioned (by you) ESA-SICCI-2 project PVIR report; you might want to cite this here. - Line 191: I don't think that this recirculation of thicker ice from the Amundsen Sea into the Ross Sea is an anonamous phenomenon. You might consider removing this statement. - Line 193/194: What are "posivite" or "negative deviation" between Envisat and ICESat SIT? Perhaps it would be better to speak about "Envisat SIT exceeds ICESat SIT" ... and "Envisat SIT is below ICESat SIT" ... or similar. - Line 194-195: Isn't this statement a bit hypothetic? In the Arctic, for example, maximum SIT values are observed in April

or even May with little change between April and May. In addition, since you compare grid-cell mean SIT values you might have a substantial influence of a potentially lower sea-ice concentration. It would be interesting to look at the pure SIT values, wouldn't it? - Line 195/196: Where is this statement substantiated from?

Lines 203-213: - Line 203: I note that FM-periods are at the end of summer and mark the begin of the freezing season; in 2007 this period is even shifted by one month towards winter (March-April). It might therefore be better to refer to this period as autumn or summer-autumn transition and denote the May/June period as winter. Or simply change the beginning to "At the end of summer melt (FM), ..." - Lines 208/209: "may result": Don't you have sea-ice concentration data provided along with the Envisat SIT data? You can simply check this out and make a more clear statement here. - Lines 209/210: "Envisat also detects the Ross sea ice ..." –> I guess you wanted to refer to the Ross Ice shelf polynya and the sea-ice thickness? - Line 211: See my previous comment about this formulation. - "is bad" –> please change to something without valueing, e.g. "We find larger RMSD values ... by XY and smaller correlation values ... by XY than for spring.

Lines 214-224: - Line 215: Which "errors" are you referring to here? I guess you still refer to the differences between Envisat and ICESat SIT, right? So there is no error involved. - Line 218: Which "deviations"? To which panel of Figure 7 are you referring? I only find differences. - Line 219: "The ice growth ..." –> I suggest to delete this sentence as it is not relevant in this context. - Line 221: "from to thinner" –> "from thicker to thinner" - Line 223: "... are high among all periods ..." –> I assume you wanted to write "highest among"? - Line 224: I suggest to add 1-2 sentences pointing out that correlations are substantially better for 2004 and 2005 than 2006 while, at the same time, the coverage with valid Envisat SIT data (Fig. 7, top row) is considerably smaller than for ICESat (Fig. 7, middle row). This needs to be discussed: two months for 2004 and 2005, including May, only 1 month (June) for 2006; since MJ periods are located in the seasonal upswing of sea-ice coverage and thickness it can make a big

difference to include or exclude one month of data. I also suggest to point out the inter-annual variability in data pairs with valid data for each season. There a considerable variations in the number of valid data pairs within one season - mostly because of data gaps in the ICESat SIT data set.

Figure 8: - I suggest to re-order the panels to [end-of-]summer (left), winter (middle) and spring (right) to have the correct time line of development. This would ease discussion about whether mean (and modal?) SIT increase from winter to spring - which is what should be the case. - I am wondering about the coarse bin-size of 0.5 meters; did you experiment with smaller bin-sizes of, e.g. 0.25 m or 0.2 m. If not, what is your motivation to use such a large bin size. Could that be the accuracy? - I suggest to use blue and red also as font colors for the annotation within the panels, make the rectangles in a) simple lines, and add the modal value, ideally both as a (dashed) vertical line and as a numner. Note that the modal SIT value is a better measure for thermodynamic sea ice thickness growth than the mean SIT. - Are the normalized to have sum 1?

Lines 225-238: - Line 226: "... agreement ... turns bad ..." –> please re-phrase and omit "bad". - Line 231: "Both of the two data sets cannot detect the thin ice smaller than 0.5 m ..." –> Looking at Fig. 8 a) and b) reveals that there are very small but non-zero fractions of SIT values y 0.5 m. I suggest to rewrite: "We find that thin ice with thickness values < 0.5 m is practically absent in both investigated products." In order to avoid the conclusion that ICESat or Envisat are not capable to resolve such thin ice you might want to add a statement into that direction. - Lines 232-235: This passage should be re-written. It is not clear what you are referring to here. Of course the mean (!) SIT is larger in summer than winter or spring because any young ice or thin ice is missing and the only grid cells left with sea-ice coverage contain - at that time of the year which is actually end-of-melt - thick ice; this is comparable to the situation in the Arctic during the sea ice minimum in September. One should not refer to "growth" or "grow" here. What is more interesting is that the mean SIT derived with ICESat increases from MJ to ON periods while the Envisat one does not. As commented in the context of Figure

8 I also encourage to include modal SIT values into the discussion. I recommend to delete the "The rest can be ..." sentence. - Line 236: "weaker" –> "smaller"; in addition you mention warm and wet weather condition and flooding sometimes. Does this refer to summer only or to the entire year? You might want to specify the key issue for penetration depth here which is wet snow as a consequence of, for example, above-freezing air temperatures - Line 237/238: "Since ICESat ..." –> This sentence should be deleted because it is wrong. ICESat is much better suited to detect thin ice than Envisat thanks to its small footprint size.

Lines 239-251: - I suggest to delete the very first sentence. It is fine to simply also show results broken down by region as is commonly done in many publications about sea-ice concentation and thickness products. - I find this paragraph not particularly well structured. Perhaps you can solve this by avoiding to jump between the different statistical parameters but stick with one and describe it in a comparative manner across the regions / seasons - ideally highlighting the most interesting issues. Clarification what you mean by "deviations" would also help - see respective previous comments. - Lines 248-251: I suggest to delete these sentences as they appear to be too hypothetical. Also, you start the discussion in the next section anyways.

Figure 9: - I suggest to add the identity line in each scatterplot.

Table 3: - I suggest to re-order seasons such that one can clearly see the change in SIT from autumn (or winter as I'd call it) to spring. - Please add a unit for RMSD. - I possible please provide standard deviations for the means of ENV and ICE (like I suggested already in the context of Table 2).

Lines 253-262: Title of this section: "deviation" –> I'd say "deviation" is used if one wants to describe how well a data set agrees to a known standard or calibration data set. This is not the case in your study. You inter-compare different data sets of which none is the standard. Therefore the term "difference" would match considerably better. - I suggest to delete the first sentence of this paragraph. You don't need this kind of

introduction here. - Line 259/260: "respond differently to different surface roughness." –> This is true ... sure ... but how about resolving leads / open water required for an adequate representation of the local sea-surface height required to derive the freeboard? Isn't this a much more important difference in the obervation capabilities of the two sensors? - Line 261: "... while ICESat uses ..." –> This reads as if the ICESats sea-ice thickness values are not retrieved using the hydrostatic equilibrium assumption ... but for both sensors this assumption is applied; ONE of the differences is in fact the treatment of the snow depth - here you are correct.

Figure 10: I have the same comments as I voiced in the context of Figures 5 to 7 in terms of latitudinal extent, the vertical line of missing data just west of 180E and the fact that the Figure cannot be used on its own because the annotation of the color bar is not complete.

Figure 11: - The maps in this figure would need to be changed according to changes noted for Figure 10. - Relative humidity is typically given in percent, i.e. your scale would go from 70% to 100% and would be entitled "Relative humidity [percent]". - Do you think these values are credible? I can agree with the fringe of low values around the continent, indicating the impact of the catabatic air flow ... but why is the relative humidity higher over sea-ice covered areas than over open water areas? Isn't open water a much more efficient source of evaporation and hence input of water vapor into the lower troposphere? The only reason I could think of why the relative humidity is higher over sea ice than open water is because of the colder temperatures. I note that the difference between open water and sea ice is most pronounced for the cold May/June period compared to the milder October/November period. This is kind of in line with my argumentation. - It does make a difference whether one uses saturation with respect to a water or to an ice surface when computing the vapor pressure and hence the relative humidity. Which of these is used for the relative humidity you plot here?

Lines 264-278: - Line 268: "weaken" –> "decrease"; "into the snow-ice interface" –>

"into the snow" - In the context of what is written here and in the paragraph Line 279++, I recommend to read » Kwok and Kacimi, 2019: Three years of sea ice freeboard, snow depth, and ice thickness of the Weddell Sea from Operation Icebridge and CryoSat-2 and the paper by » Paul et al., 2018: Empirical parametrization of Envisat freeboard retrieval of Arctic and Antarctic sea ice based on CS-2: progress in the ESA CCI . These papers are good preparations for a thorough discussion of the issues touched upon. - Lines 271++: I don't want to rule out that the atmospheric humidity might have had an impact on Envisat RA2 measurements that differs from the impact on ICESat measurements. However, I would expect that this would involve the entire troposphere and not the near surface relative humidity. I am sceptical, however, whether the near surface humidity, which is second order parameter, is a good measure to judge whether a snow surface / cover is wet or dry. I would have thought that the air-temperature is the more direct driver. Please check.

Lines 279-284: - Line 280 "footprint is rather rough" –> I don't understand. How can a footprint be rough? The ice or snow surface can be rough. - That the detection of leads can affect the detection of thinner ice sounds trivial. If a sensor cannot resolve leads and/or resolve leads only indirectly and potentially also at the wrong position, then it is more than likely that thin ice thickness is not retrieved overly well. But what has this to do with "shorter ice"? What is this? What is "wider sea ice". I suggest to completely rewrite this paragraph after having studies the two paper I just recommended to read. Also, and this is what I cannot understand: You have two experts for radar altimetry based sea-ice thickness retrieval among your co-authors - so two people you could ask directly.

Lines 286-302: - I suggest to repeat one more time a) Envisat SIT is based on an AMSR-E snow-depth climatology and b) ICESat SIT does not use snow depth in the retrieval but uses climatological ratios between snow depth and sea-ice thickness in form of the factor R to compute the modified density. This is a lot of "meat" for discussion. –> pitfalls of using a climatology instead of actual snow depth values (Bunzel and

Notz, 2018, Retrievals of Arctic sea-ice volume and its trends significantly affected by interannual snow variability); –> reduction of the relative uncertainties by using a climatology rather than actual values; –> limitations of the R-factor usage in the ICESat SIT data set (also based on climatology, and in addition based on ASPeCt observations with their known difficulties); –> discussion how the new approach by Li et al. (2018, mentioned above) would compare here; discussion of when which snow physical properties would cause which bias for which of the periods considered (i.e. winter compared to spring compared to summer). - Line 299/300: Kwok and Maksym (2014, see above) looked into this and figured out what typical snow depths appear to look like and Kern and Ozsoy-Cicek (2016, see above) figured out that indeed there is a substantial under-estimation. I recommend to refer to numbers from the respective publications to underline your assumptions and hypotheses. - Line 200/301: The sentence "Therefore, the sea ice ... negative." needs to be rewritten. - Line 302: "snow depth uncertainty over the deformed sea ice" –> Please be more specific. You know which sign this uncertainty would have (a positive bias!); hence, cases with actually a lot of deformed ice would have too little snow retrieved while cases with just a bit deformation would have fairly good retrievals. But: The ENV data set uses a climatology. Therefore, the largest effect might not come from deformed / level ice's impact on the snow-depth retrieval but the largest effect might be due an actual snow depth which differs completely from that represented by the climatology - no matter whether the sea ice is particularly deformed or not.

Lines 318-325: This paragraph would read better in Section 4 I suggest.

Typos / editoral issues: Line 11: "global" –> "hemispheric" because you consider the Antarctic not the entire Earth.

Line 16: Suggest to add "these" between "with" and "field" to underline that the results of your comparison are only based on the ULS data.

Line 42: "errors" –> I'd prefer to see "limitations"

Line 43: "AEM data" –> Please tell the reader which parameter is derived from the AEM data.

Line 44: "only exists in the Weddell Sea" –> this is not correct. Please change to: "has mostly been obtained in the Weddell Sea"

Line 45: - Why "time series"? These observations are not carried out at a specific point. - "freeboard observation" –> Yes, but there is an additional snow radar which allows additional snow depth estimates. Also, you might want to include the information whether the freeboard measured is the total or the sea-ice freeboard.

Line 46: "limited to several trajectories in the Weddell Sea" –> This is not entirely true since there are also trajectories in the Bellingshausen Sea; more importantly, however, is that substantial work has been done with these data, e.g. Kwok and Maksym, 2014: Snow depth of the Weddell and Bellingshausen sea ice covers from IceBridge surveys in 2010 and 2011: An examination; and Kwok and Kacimi, 2019: Three years of sea ice freeboard, snow depth, and ice thickness of the Weddell Sea from Operation Ice-bridge and CryoSat-2; this work the the importance of this data for our understanding of sea-ice thickness retrieval of Antarctic sea ice using altimeter data should not be undervalued.

Line 47: ULS data provide primary the sea-ice draft; sea-ice thickness is a derived variable. I suggest to change this to draft.

Line 48: "the basin ..." –> "a basin ..."

Line 50-52: - Bernstein et al. (2015) –> I suggest to add Li et al. 2018: Spatio-temporal variability of Antarctic sea-ice thickness and volume obtained from ICESat data using an innovative algorithm

- I suggest to remove passive microwave and SAR here to remain concise and then stress that altimetry has proven to currently be the best source for Antarctic wide sea-ice thickness retrieval over the full thickness range.

Line 55: "three"? –> I guess it is two decades because ERS1/2 is in progress but not yet ready to be released.

Line 58: "one of the latest ..." –> Perhaps latest is not the predominant issue here but the fact that it is finally a combination of CryoSat-2 AND Envisat.

Line 71: "estimates ... depth)" –> "allowed to estimate the total freeboard (sea ice freeboard + snow depth) via determination of the surface elevation"

Line 74: I suggest to write "compared" instead of "developed".

Line 76: "period" –> "periods"

Line 78-81 / 88 - Since you defined ULS already there is not need to again write it in full length. - The same applies to airborne electromagnetic EM sounding which should be "AEM sounding" in Lines 79. - I am sure the PVIR has an author team so you can refer to it as "author et al., 2018". - I am relatively sure that the evaluation this report refers to also included CryoSat-2 data. Please check and amend accordingly.

Line 96/97: The URL given only points to the Envisat sea-ice thickness data set; hence you need to remove the CryoSat-2 part of this sentence.

Line 97: "gridded" –> "grid"

Line 98: "consistent" –> why consistent? What do you mean by this?

Line 108: "girds" –> "grids"

Line 112: You have defined ICESat already, so no need to still use the full name.

Line 113: This is total freeboard, right? See GC1.

Line 116: "five main" –> "several"

Line 119: "suggested by Worby" –> see comment I had under "specific comments" already.
Line 126: "ASPeCt observations" –> cite the respective Worby et al. (2008) paper here.

Line 128: This is total freeboard, right? See GC1.

---

## Referee Comment (RC2) · Anonymous Referee #2 · 17 Apr 2020

Three Antarctic sea-ice thickness data sets are analyzed: Envisat radar altimeter, ICE-Sat laser altimeter, and Upward-Looking Sonar (ULS). The satellite data sets are compared in five regions and three seasons over several years in the early 2000s. In spring, the mean difference between Envisat and ICESat sea-ice thickness is small, but in summer and fall, the Envisat values are much larger than ICESat. Differences are attributed to lack of radar penetration through wet snow, different footprint sizes, and snow depth assumptions. Relative humidity may explain the seasonal differences that arise from wet snow.

In general, the data comparisons in this paper are done adequately, but the analysis of differences is speculative and weak. I think major revisions will be needed.

.

[Figure]

Main Comments

The first paragraph of the Introduction discusses the extent of Antarctic sea ice, but it fails to mention the fact that almost all of that sea ice is seasonal – it completely melts away every year, except in a small portion of the western Weddell Sea. Therefore, it's not clear to me why "sea ice thickness is an equally critical component as sea ice extent" (line 33). I understand the importance of sea-ice extent as a barrier between the ocean and the atmosphere that affects albedo and the exchange of heat and moisture, but I think the authors need to explain better why Antarctic sea-ice thickness is so critical, given that almost all the ice melts away every year.

In the comparisons of ULS data with Envisat and ICESat (Section 3.1) there is no mention of the fact that ULS measurements are made at a single point, whereas Envisat and ICESat measurements are made over large footprints or areas. How might that affect the comparisons?

Lines 232-238. The mean sea-ice thickness in both the Envisat and ICESat data increases from spring to summer. The authors call this an "anomalous thickness growth" as if it can't possibly be true, and they attribute it to "limited comparison pairs" and "uncertainties of both data sets". However, isn't it possible that the mean ice thickness could actually be greater in summer, because the thinnest ice melts away, leaving only thicker ice? Antarctic sea-ice extent is about 18 million sq km in spring and 3 million sq km in summer. Suppose the spring ice extent consists of 15 million sq km of 1.3 m ice and 3 million sq km of 3.2 m ice, for a mean thickness of about 1.6 m (matching the actual spring ICESat mean thickness). And suppose that all the ice loses 1.3 m of thickness in the summer melt. Then only 3 million sq km are left, with a mean thickness of 3.2 - 1.3 = 1.9 m (matching the actual summer ICESat mean thickness). I'm not saying that these are the correct numbers, I'm just pointing out the plausibility of the argument that the mean thickness could be greater in summer than in spring. Of course it requires a more careful analysis.

Section 4 discusses the reasons for the differences between Envisat and ICESat sea-ice thickness: (i) if the snow is wet, the Envisat radar does not penetrate all the way to the snow/ice interface; (ii) the footprints of Envisat and ICESat are different; (iii) snow depth is treated differently in the retrieval algorithms. These are all legitimate potential reasons for the differences between Envisat and ICESat ice thickness, but as presented here, they are speculative and qualitative arguments, not quantitative. For example, consider equation (1) for the ice thickness from Envisat, in which F is the measured freeboard and S is the assumed snow depth. If the radar backscatter is from wet snow within the snow layer, rather than the snow/ice interface, then the measured freeboard F is partially ice and partially wet snow. This would lead to a modification of equation (1) and a new ice thickness I' instead of I. Does the difference I-I' account for the bias in Envisat relative to ICESat? Another example: regarding snow depth, how much of a change in snow depth in the Envisat retrieval algorithm would be needed to account for the bias in ice thickness relative to ICESat? Is this change in snow depth within the uncertainty of the snow depth measurements? Another example: Figures 10 and 11 suggest a connection between ice thickness differences and relative humidity. What is the correlation? Have other researchers considered this connection? Another example: a very simple model of ice thickness is based on cumulative freezing-degree-days (FDD), e.g. Lebedev 1938. Using temperature fields from (say) a reanalysis product, how does a simple FDD ice thickness model compare to Envisat and ICESat ice thickness? Would this provide any insight into biases? My overall point is that the analysis in this paper (Section 4) needs to be more quantitative. The authors claim that "without enough observation data and numerical model experiments we cannot quantify the impacts of the uncertainties over the sea ice thickness." (lines 327-328). But my suggestions above for further quantitative analysis do not require any additional ice thickness data or numerical model runs that are not already publicly available. This is not a question of lack of data, it's a question of digging into the comparisons of Section 3 more quantitatively.

.

[Figure]

Minor Comments

Lines 19-23. It's not clear whether the deviations are Envisat minus ICESat, or ICESat minus Envisat.

Line 21. "the large correlation coefficient" – Is this a spatial correlation or a temporal correlation? Please add the word "spatial" or "temporal" as appropriate.

Line 103. Following equation (1), say that rho is density.

Line 104. Give a reference for AMSR-E snow depth climatology.

Line 108. A threshold of 70% is given here, but on line 131 it says 60%.

Lines 125-126. "R is the ratio of sea ice thickness over snow depth, which is a seasonally dependent factor and calculated from ASPeCt observations." I understand that R changes seasonally, but does it also change from year to year based on ASPeCt observations?

Line 139, equation (4). Please provide UNITS for the quantities in this equation.

Line 146. "the deviations." – please say explicitly "the deviations between Envisat and ICESat sea-ice thickness" or whatever deviations are being referenced here.

Lines 151-153. "The seasonal classification is based on the ICESat operating periods... If ICESat data has overlapping time over ten days with respective months, we average Envisat data over the two months." OK, but wouldn't it be better to do a time-weighted average of the monthly Envisat data to match the ICESat period? For example, consider the ICESat period from Feb 17 to Mar 20. Instead of averaging Envisat over all of February and March, consider this: the ICESat period is 32 days long – 12 days in February and 20 days in March. So calculate: (Env. avg.) = (12/32)*(Env. Feb.) + (20/32)*(Env. Mar.) Wouldn't that provide a more accurate Envisat average with which to compare ICESat?

Line 164. For the ULS data, I understand that "207" refers to location #207 in Figure

2, but I don't understand "207-6" – what is the "6"? Please explain your numbering system.

Line 168. "Due to the discontinuity..." – what discontinuity?

Lines 173-174. In Figure 4, where do the error bars come from? What do they represent? One standard deviation? 95th percentile?

Line 190. It would be helpful to indicate on one of the maps the location of the Ross Ice Shelf polynya and the Ronne Ice Shelf polynya.

Line 192. I think "clockwise" should be "counter-clockwise". Please check.

Lines 216-217. "Comparing the values in the eastern Antarctic, ICESat shows some deformed ice up to 3 m while Envisat shows smaller thickness by about 1.5 m." I don't see this in Figure 7. Please give approximate longitudes or otherwise indicate in Fig 7.

Lines 241-242. In reference to Figure 9, "In the western Weddell Sea, the regression lines have large positive intercepts in all three seasons." Yes, this is true for all five regions, not just the western Weddell Sea.

Lines 261-262. "ICESat uses the modified snow-ice density to get rid of the biased snow depth." I don't see how equation (2) would get rid of a biased snow depth. The snow depth S is part of the factor R = I/S.

Lines 295-298. "wet snow caused by melt or flooding could lead to underestimations while refreezing of molten snow could lead to overestimations [of snow depth]. All of the above biases can also cause the differences between Envisat and ICESat." I don't see how underestimates AND overestimates of snow depth can BOTH lead to a positive bias in Envisat ice thickness relative to ICESat. Please clarify.

Lines 300-301. "in underestimations of snow depth... the sea ice thickness deviation presents negative." It's not clear to me whether "deviation" refers to the error in the Envisat ice thickness, or the difference Envisat-ICESat ice thickness. Consider equation

(1). If the snow depth S is an underestimate, then the true snow depth S' > S. If the measured freeboard F remains constant, then the true ice thickness I' > I. The error I-I' < 0 (i.e. negative error in the original estimate I). On the other hand, I'-ICEsat > I-ICESat (i.e. increased bias of Envisat). Please clarify the use of "deviation" and the effect of snow depth on the calculated ice thickness.

Figure 2. It would be helpful to outline the zero contour (the coastline) to make it easier to distinguish land from ocean. Also, the caption should say that the background is bathymetry, and give the source of the bathymetry data.

Figure 3. Please add at the end of the caption: "with 50 km grid size."

Figure 4. See comment above for line 164: I understand that 207 refers to a location in Figure 2, but what does "207-6" mean? Also, see comment above for lines 173-174: in the caption, say what the error bars represent. Also, the dates along the horizontal axes should be in a more readable format such as 2008/03 instead of 200803. Perhaps the journal has a standard format for such dates.

Figure 6. Consider rotating the whole figure into landscape mode, which would allow the panels to be larger.

New table. This is just a suggestion, but I found it helpful to create a table for myself of the different data sources, their spatial and temporal resolutions, and their treatment of snow. For example:

Source | Spatial res | Temporal res | Snow
* * *
Envisat | 50 km grid | monthly avg | AMSR-E climatology

ICESat | 100 km grid | see Table 1 | ASPeCt observations

ULS | single point | monthly avg | built into eq (4)

.

Typographical Corrections

Line 14. Change "firstly" to "first"

Line 14. After "ICESat ice thickness" add "product"

Line 23. Change "deduce from" to "come from"

Line 47. Change "provides" to "provide"

Line 56. After "ice thickness data" add "set"

Line 96. Delete the URL. It is given in the "Data availability" section on page 11.

Line 98. Change "month" to "months"

Line 100. Delete the word "data"

Line 108. Change "girds" to "grids"

Line 133. Change "provides" to "provide"

Line 135. Delete the word "criterion"

Line 143. Perhaps "for that" should be "because"?

Line 160. Delete "at first"

Line 164. Delete the word "Basically,"

Line 172. Change "Envisat thickness exceeds ULS remarkably" to "Envisat thickness greatly exceeds ULS"

Lines 186 and 188. Change "seems" to "is"

Line 199 and following. "by -0.03" should be "of -0.03". In most of the paper, "by X.XX" should be either "of X.XX" or "at X.XX" depending on context.

[Figure]

Lines 205 and 206. Change "reveals" to "shows". The word "reveals" means that some-thing previously hidden or unknown is now exposed, implying that the revealed value is the true value. But in this case, the "revealed" value may be biased or uncertain, so it's better to say "shows".

Line 207. Change "which can also be reflected from the..." to "which is also reflected in the..."

Line 220. Change "disability" to "inability"

Line 221. Delete the word "from"

Line 227. Change "malposition between two histograms" to "offset between the two histograms"

Lines 230-231. Change "while ICESat during 1.0 m" to "while ICESat peaks at 1.0 m"

Line 231. Change "and a mean thickness" to "with a mean thickness"

Line 231. Delete the word "two"

Line 231. Change "the thin ice smaller than" to "the ice thinner than"

Line 232. Change "smaller" to "thinner"

Line 233. Change "grow" to "grows"

Line 244. Change "The comparison results perform better" to "The results are better"

Line 248. Change "little" to "few"

Line 249. Change "deduce" to "arise"

Line 250. Change "quantity" to "quantitative"

Line 254. Delete the word "that"

Line 255. Delete the words "implementation of"

Line 260. Change "Another one" to "Another difference"

Line 280. What is "shorter" ice? Does this mean "less extensive" in area?

Line 281. Does "wider" mean "more extensive"?

Line 283. Delete the word "loaded"

Line 287. Delete the words "used in the retrieval"

Line 296. What is "molten snow"? Is it "wet snow"?

Line 315. Change "deduce" to "come"

Line 320. Delete the word "affected"

Lines 324-325 and lines 327-328. These sentences say the same thing. It's not necessary to repeat.

Line 331. After "sea ice thickness" add "data"

Figure 5 caption. "first and second rows" (rows – plural) and "the last row shows" (shows – singular). Yes, English is strange.

Figure 8 caption. "red line" should be "red lines"

Table 1 caption. Say that ON is Oct-Nov, FM is Feb-Mar, and MJ is May-June.

Table 2 and Table 3 captions. Spell out "sea ice thickness" instead of SIT. Also say that CC stands for correlation coefficient.

---

## Author Comment (AC1) · 4 Jun 2020

**General Comments:**

GC1 (editoral): Please be consistent with usage of total freeboard and sea-ice freeboard. Try to be as clear as possible in the way you assign the respective wordings to the two different types of instruments used.

GC2: The comparison with the ULS data requires more details and a slight change in the way the comparison is presented. Currently the message to be taken is based on very few data points with limited discussion of the reliability of such an inter-comparison. This bears the risk to over-interpret the results which to my opinion should be avoided. Please see my specific comments.

GC3: The discussion of the results of the comparison between Envisat and ICESat SIT data sets (i.e. Section 4) would benefit from some more considerations and re-writing. This applies to the quality of the ICESat data considered as "the truth", to the way the statistical results are interpreted, and to some details in the description how the comparison was carried out. Please see the respective specific comments.

Dear Reviewer:

We would like to thank you for the helpful comments to improve this manuscript. For GC1, we changed all the "freeboard" to "total freeboard" for ICESat or "sea-ice freeboard" for Envisat to make them distinguished clearly. For GC2, we combined the two successive records of 229 and 231 and calculated the mean differences and root mean square deviations for Envisat-ULS and ICESat-ULS sea ice thickness following your comments, and modified the analysis based on the more detailed statistics. For GC3, we amended the descriptions and regarded ICESat data as a reliable reference rather than the truth value. Moreover, we carefully considered and modified the interpretations of these statistical results, and added more details about how the comparison was carried out.

The specific responses and revisions are shown below. They are in blue font for clarity.

Corresponding Authors:

Qinghua Yang (yangqh25@mail.sysu.edu.cn) and Qian Shi (shiq9@mail.sysu.edu.cn)

**Specific comments:**

**Point 1:** Abstract: - I recommend to include number of the comparison results between the two altimeter sea-ice thickness data sets and the ULS draft data set, i.e. something like the mean difference and its standard deviation. It would also be helpful to learn from the abstract rightaway whether you transferred the ULS draft measurements into sea-ice thickness or whether you converted the satellite data into sea ice draft by simply saying whether you compared draft or sea-ice thickness. Why? Because any conversion (draft to thickness or thickness to draft) has inherent errors. - I recommend to focus on the one parameter, e.g. the mean difference and its standard deviation, and report this for the entire Antarctic and all sub-regions. This gives a more complete picture of your intercomparison than the bits and pieces of smallest deviations, largest correlations, largest deviations, etc. you do provide currently.

*Response:* Following your constructive recommendations we have conducted more analyses:

(1) We amended the sentence describing the comparison with ULS: "Both SIT data sets are compared with that from upward looking sonar (ULS) in the Weddell Sea which are converted from the ice draft at first, showing the mean differences and the standard deviations (SD) of 0.78 m and 0.72 m for Envisat-ULS while 0.32 m and 0.52 m for ICESat-ULS, respectively." *(please see P1 line 15-17 in the revised manuscript)*

(2) We amended the sentences describing the results of inter-comparison: "More specifically, the smallest seasonal mean difference (with SD shown in brackets) that Envisat SIT minus ICESat SIT for the entire Antarctic exists in spring by -0.01 m (0.44 m) while larger differences exist in summer and autumn by 0.51 m (0.70 m) and 0.61 m (0.49 m), respectively. With respect to different sea sectors, the regional mean differences (with SD shown in brackets) are 0.62 m (0.89 m) in the Western Weddell Sea, 0.31 m (0.53 m) in the Ross Sea, 0.28 m (0.57 m) in the Eastern Weddell Sea, 0.18 m (0.65 m) in the Bellingshausen and Amundsen Sea, and -0.10 m (0.63 m) in the Eastern Antarctic." *(please see P1 line 22-27 in the revised manuscript)*

**Point 2:** Lines 63-64: - "Hendricks et al 2010" –> I guess there is more recent literature in peer-reviewed journals targeting this issue and even using for improvement of current retrievals? Please check. - "Ku-Band radar altimeters" –> Is this a special feature of Ku-Band or perhaps generally valid for radar altimeters? Please change accordingly if needed.

*Response:*

(1) We added two more citations here about the uncertainties resulting from sea ice surface roughness (Ricker et al., 2014; Landy et al., 2020). Ricker et al. (2014) revealed that freeboard bias caused by a combination of the choice of retracker threshold, the unknown penetration and surface roughness effects were roughly 0.06–0.12 m. Landy et al. (2020) revealed that overlooking the roughness introduced a freeboard uncertainty of up to 30% and developed a new algorithm for

retracking CryoSat-2 radar echoes, but haven't been employed by current Envisat SIT. *(please see P3 line 81-82 in the revised manuscript)*

(2) For the "Ku-Band radar altimeters" that you pointed out, we were meant to point out that compared with ICESat laser altimeter, Envisat and CryoSat-2 radar altimeter has a larger footprint. Therefore, we deleted "Ku-Band" and added "Due to the larger footprint compared to laser altimeters" in the beginning of the sentence. *(please see P3 line 82 in the revised manuscript)*

**Point 3:** Lines 65-67: - There is more recent literature about this issue - perhaps Kern and Ozsoy-Cicek 2016: Satellite Remote Sensing of Snow Depth on Antarctic Sea Ice: An Inter-Comparison of Two Empirical Approaches; or Kern and Ozsoy 2019: An attempt to improve snow depth retrieval using satellite microwave radiometry for rough antarctic sea ice; or Price et al. 2019: Snow-driven uncertainty in CryoSat-2-derived Antarctic sea ice thickness - insights from McMurdo Sound - "must use" –> no, not necessarily. There are other alternatives to deal with snow, e.g. using snow accumulation from re-analyses.

*Response:* We realized that many new approaches have been applied to generate better snow depth estimations in the Antarctic in the past few years. However, we still need to point out that the snow depth climatology used in the retrieval of Envisat and CryoSat-2 introduces bias to their sea ice thickness. Therefore, we modified the two sentences in line 65-68 as: "In addition, snow depth climatology used in the retrieval of Envisat and CryoSat-2 sea ice thickness can cause additional uncertainties due to neglecting inter-annual variability in snow depth (Bunzel et al., 2018)". *(please see P3 line 87-89 in the revised manuscript)*

**Point 4:** Line 77: Please already here add one sentence about ICESat-2 which is in orbit since September 2018 and provides excellent freeboard measurements which are not discontinuous in time anymore.

*Response:* We added a sentence here as suggested: "However, the ICESat-2 satellite which is in orbit since 2018 provides a new source of year-around observations of total freeboard." *(please see P4 line 98-99 in the revised manuscript)*

**Point 5:** Line 85: This approach was not suggested by Worby. It was suggested by Kern et al. 2016. This approach has been called 1-layer Worby et al approach in Kern et al. (2016) because the ratio between snow depth and sea-ice thickness used in Eq. 2 is based on the ASPeCt observations published by Worby et al. (2008); Please correct accordingly. Note that this approach has been further refined by Li et al. 2018 (see comment under "typos/editoral issues" Lines 50-52), who derived first guess values of snow depth and sea-ice thickness directly from ICESat data using empricial approaches instead of the climatological values used by Kern et al. (2016); that way the sea-ice thickness retrieval using this 1-layer method is certainly considerably more accurate than

the one introduced for the first time in Kern et al. (2016) because it takes the actual situation into account better. Of course, it needs to be noted that the empirical approaches used by Li et al. (2018) were in turn developed from a (more or less comprehensive) suite of historic in-situ observations of freeboard, snow depth and sea-ice thickness which in a way have the character of a climatology as well.

*Response:*

(1) We realized the mistake and corrected it as: "... suggested by Kern et al. (2016) as referenced data". *(please see P4 line 107 in the revised manuscript)*

(2) We investigated the ICESat product that Li et al. (2018) produces by comparing with ICESat from Kern et al. (2016) and the ULS SIT used in this study. From Fig. 1 we can see that the differences between two ICESat products are small in general, with some larger differences in the West Weddell Sea and Amundsen Sea. Table 1 shows that compared with ULS SIT, ICESat SIT from Kern et al. (2016) performs even better. Based on these analyses, we think that the inter-comparison results with Envisat SIT in this study are not affected by the choice of ICESat product. Therefore, our work is still based on the data produced by Kern et al. (2016). However, we will discuss the difference caused by this new retrieval method in Section 4 as your comments.

[Figure]

Figure 1. Maps of differences that ICESat SIT from Kern et al. (2016) minus ICESat SIT from Li et al. (2018) at each operating period.

Table 1. The differences and RMSD between ULS SIT and the two ICESat SIT at each site. ICE(K)

refers to ICESat SIT from Kern et al. (2016) and ICE(L) refers to ICESat SIT from Li et al. (2018).

| | ICE(K)-ULS | | ICE(L)-ULS | |
| --- | --- | --- | --- | --- |
| | D (m) | RMSD (m) | D (m) | RMSD (m) |
| Site 207 | 0.60 | 0.66 | 0.68 | 0.34 |
| Site 229 | -0.04 | 0.07 | 0.17 | 0.15 |
| Site 231 | 0.27 | 0.34 | 0.33 | 0.44 |

**Point 6:** Lines 98-107: - The last paragraph of this block about Envisat should come first and it should be expanded. Please provide a bit more information about the Envisat RA2 (time of operation, orbit details (observation hole at pole?), nominal resolution, altimeter type etc.). Then you could come up with something along the lines how sea-ice freeboard is obtained and then how from the sea-ice freeboard sea-ice thickness is derived. I suggest to avoid this "is defined" in Line 99; I don't think there is a definition. What the RA2 measures is a runtime which is converted into a surface elevation which in turn is converted into a radar freeboard in case of sea ice. For this freeboard it is not precisely known whether this is representative of the location of the ice-snow interface. It is assumed that this is the case based on the experience from laboratory work (you cited it, Beaven et al.) and because of this snow-depth dependent radar signal delay is applied to convert the radar freeboard into what is assumed (not defined) the sea-ice freeboard. - Line 101: Kern et al. (2016) is certainly not the correct reference here. That paper is about ICESat and has nothing to do with radar altimetry. You hence need to find another paper. Since you have at least two radar altimeter experts, which were involved in the derivation of the Envisat data set you are using, among your co-authors it would be easy to find an appropriate reference. Just ask them! - Equation 1: Please explain in the text what the rhos are. - Lines 103/104: I am sure the information about how the sea-ice thickness is retrieved is given in the respective ATBD of the data set; again I recommend to ask the two people who were actively involved in the processing of the Envisat data set. You will need this information also for answering which sea-ice concentration product is included in the data set (Line 110) and for figuring out what the sea-ice concentration threshold is (anyways) which was used to derive freeboard and sea-ice thickness (Line 108).

*Response:*

(1) The last paragraph was moved to the first part and we added more information about the Envisat RA-2: "Envisat was launched on 01 March 2002 and the mission ended on 08 April 2012. The Radar Altimeter 2 (RA-2) aboard on Envisat is a nadir-looking pulse limited sensor operating at the main frequency of 13.575 GHz (Ku band), with a secondary frequency of 3.2 GHz (S band) compensating the ionospheric error (Zelli and Aerospazio, 1999). It has an orbit inclination of 98.55° (Paul et al., 2017) covering the full Southern Ocean and nominal circular footprint of 2–10 km in diameter (Connor et al., 2009). Because the RA-2 is the only altimeter carried by Envisat, we call it Envisat hereafter." *(please see P4-5 line 121-126 in the revised manuscript)*

(2) We changed the way to describe the retrieval of Envisat sea ice freeboard: "The Envisat radar freeboard is retrieved based on the radar range obtained from RA-2 Level-1 waveform data over ice surface and leads between ice floes. Under ideal circumstance, the signal will return at the interface between snow and ice based on the experience from laboratory work (Beaven et al., 1995). Then, snow-depth dependent radar signal delay is applied to convert the radar freeboard into the sea-ice freeboard. The illustration of sea ice freeboard is shown in Fig. 1, which is the sea ice surface elevation relative to the sea surface elevation." *(please see P5 line 128-132 in the revised manuscript)*

(3) We find that the first usage of hydrostatic equilibrium method during sea ice retrieval is Laxon et al. (2003). Therefore, we substitute Kern et al. (2016) as Laxon et al. (2003). *(please see P5 line 132-133 in the revised manuscript)*

(4) We add the descriptions: "$\rho_{water}$, $\rho_{snow}$, $\rho_{ice}$ refer to the density of the sea water, snow cover and sea ice respectively." *(please see P5 line 136-137 in the revised manuscript)*

(5) We modified the description of the retrieval of Envisat sea ice thickness: "Sea ice thickness is retrieved from sea ice freeboard based on the hydrostatic equilibrium approach as first used by Laxon et al. (2003):

$$I = \frac{F\rho_{water}+S\rho_{snow}}{\rho_{water}-\rho_{ice}},$$ (1)

where $F$ represents Envisat sea ice freeboard, $S$ represents snow depth, $I$ represents ice thickness, $\rho_{water}$, $\rho_{snow}$, $\rho_{ice}$ refer to the density of the sea water, snow cover and sea ice, respectively. A snow depth climatology is employed to retrieve sea ice thickness from sea ice freeboard here. This snow depth climatology is derived from Advanced Microwave Scanning Radiometer-EOS (AMSR-E) and AMSR-2 data for the Antarctic and is based on a revised version of the approach described by Cavalieri et al. (2014) and provided by the Integrated Climate Data Center (ICDC, http://icdc.cen.uni-hamburg.de)." *(please see P5 line 132-141 in the revised manuscript)*

(6) We added the source of the sea ice concentration product included in the data set: "... we also multiply the sea ice concentration contained in the data for each grid which come from OSI-SAF Global Sea Ice Concentration (OSI-409) until April 16, 2015 and the OSI-SAF Global Sea Ice Concentration continuous reprocessing offline product (OSI-430) afterwards." *(please see P5 line 147-150 in the revised manuscript)*

**Point 7:** Line 131: 60%: Why do you use a different sea-ice concentration threshold for the ICESat data than you use for the Envisat data?

*Response:* The usage of different SIC thresholds is because of the different thresholds used in the retrieval of the two data sets. Envisat SIT employs a SIC threshold of 70% during the retrieval while the ICESat SIT uses 60%. Only areas with sea ice concentrations greater than the threshold are considered a valid area for detection of leads and sea ice. Therefore, we are meant to point out this information. In order to express more clearly, we changed the two sentences to:

"In addition, it is noted that values with sea ice concentration less than 70 % have been removed during Envisat SIT retrieval." *(please see P5 line 145-146 in the revised manuscript)*

"It is noted that grid cells with sea ice concentration less than 60% have been remove during ICESat SIT retrieval." *(please see P6 line 170-171 in the revised manuscript)*

We also tested the difference between using 60% and 70% SIC threshold for ICESat during the comparison with Envisat SIT. According to Table 2, this different threshold does not play an important role in the results of this paper. D(60) refers to Envisat minus ICESat (ENV-ICE) applying 60% SIC threshold for ICESat, while D(70) refers to ENV-ICE when SIC threshold for ICESat is 70%. Since the ice concentration gradients are usually quite steep, there will not be a lot of area with values 60% < SIC < 70%.

Table 2. Statistical results of the comparison between Envisat SIT and ICESat SIT using 60% and 70% SIC threshold at each operating period.

| | ON04 | ON05 | ON06 | ON07 | FM04 | FM05 | FM06 | MA07 | FM08 | MJ04 | MJ05 | MJ06 |
|---|---|---|---|---|---|---|---|---|---|---|---|---|
| D(60) (m) | 0.00 | 0.05 | -0.19 | 0.14 | 0.89 | 0.74 | 0.47 | 0.61 | 0.92 | 0.61 | 0.55 | 0.60 |
| D(70) (m) | 0.00 | 0.06 | -0.21 | 0.15 | 0.79 | 0.66 | 0.43 | 0.61 | 0.89 | 0.60 | 0.55 | 0.61 |

**Point 8:** Lines 133-139: - I suggest to add a bit more information about i) the field of view of such a sensor; ii) the temporal resolution with which the data are provided and which you use for your comparison; iii) the accuracy. - I would not group ULS data into "in-situ" (Line 135) because also ULS technique is kind of remote sensing and not an in-situ measurement such as a drilling of snow stake measurement would be. - The sentence containing "the most suitable criterion to evaluate" should possibly be rewritten along the lines that such data are particularly well suited to evaluate the temporal evolution of the sea-ice thickness at a fixed location. How well their accuracy is (suited) we don't know (yet). And such data are not overly well suited to evaluate the spatial distribution of the sea ice thickness. Also, when using ULS data sea-ice motion as to be taken into account. - What are the assumption behind the empirical equation 4 - or in other words: Where lie its limitations of applicability / inherent uncertainties?

*Response:*

(1) We added more information about the ULS: "The moorings are deployed at more than 900 m deep water and they transmit sound pulses upwards with a footprint of 6–8 m in diameter. The signals are reflected either by the sea ice bottom or the sea surface, yielding two distances based on the travel time. The sea ice draft, which is the depth of the sea ice underwater, can consequently be derived from the difference of the two distances. The measurements of sea ice draft were collected once several minutes from November 1990 to March 2008. In this study, we use the monthly average

sea ice draft at three sites corresponding to Envisat and ICESat operation time. According to Behrendt (2013), the uncertainty of sea ice draft varies from 5 to 12 cm, depending on the seasons. The uncertainty in summer is smaller than in other seasons because the frequent leads or open water in summer provides the benchmark for calibration." *(please see P6 line 174-182 in the revised manuscript)*

(2) We realized the mistake and changed "in situ data" to "observed data".

(3) We deleted the sentence and rewrote it in Section 3.1: "Among all the available observed data in the Antarctic, the ULS data has the advantage of high accuracy and year-round continuity to evaluate satellite SIT." *(please see P8 line 221-222 in the revised manuscript)* We conducted the comparison between ULS and satellite SIT by linearly interpolating both Envisat and ICESat SIT onto each ULS location. And we also discussed the uncertainties caused by such comparison at the end of Section 3.1: "However, it is noted that the ULS measurements are recorded at fixed locations with approximately 6–8 m footprint in diameter, while Envisat has a footprint of 2–10 km and the SIT data used in the comparison represents mean values over 50 km grid cells, and ICESat has a footprint of 70 m and the SIT data represents mean values over 100 km grid cells. This large scale difference can increase the selection biases. When the ULS measures a single point like a ridge or an edge of thin ice, satellites will detect a large area including the single point and other sea ice, and their SIT are averaged through the area. In addition, although the ULS SIT and satellite SIT are all monthly mean values, one satellite SIT grid cell are actually scanned once or twice through a month. And the average of one or two values has a poor representation of the mean SIT throughout the whole month. Theoretically, more valid measurements in one grid cell, more accurate the mean SIT is. In general, uncertainties from both the spatial interpolation and temporal representation can affect the comparisons. However, considering the typical sea ice motion in the Weddell Sea, monthly average ULS sea ice thickness could be referred as a spatial average value, represent one hundred kilometers around the fixed ULS positions. In general, Envisat and ICESat can overpass ULS positions several times a month and are comparable to that of ULS SIT." *(please see P9 line 259-271 in the revised manuscript)*

(4) Regarding the limitations of the empirical equation 4, we added several sentences to clarify: "This empirical equation is derived from drillings in the Weddell Sea based on the assumption that the snow depth values from drillings and ULS are comparable. But it still bears the uncertainties from the production of slush and snow ice caused by flooding (Harms et al., 2001)." *(please see P7 line 188-190 in the revised manuscript)*

**Point 9:** Lines 141-148: "humidity" The amount of humidity in the atmosphere is certainly a impacting the quality of the Envisat RA2 measurements but is to my knowledge taken into account by the on-board microwave radiometer being sensitive to the atmospheric water vapor load. While I would not exclude that the humidity of the atmosphere could also have an impact on snow wetness or the available of free (liquid) water within the snow I an inclined to say that its effect is much smaller than the direct influences of both the air temperature / surface temperature and the net

surface radiation flux. These parameters directly determine snow melt - be at the surface or internally, i.e. within the snow. Above a surface or snow temperature of -5degC the liquid water fraction in the snow increases. I would therefore suggest to look for air or surface temperature rather than humidity and also look at the net surface radiation. The relative humidity is a derived, secondary quantity which is possibly also more prone to errors than the temperature itself.

*Response:* We looked into the correlations between the ENV-ICE SIT difference and each meteorological element including relative humidity, 2 meter temperature, surface net solar radiation and surface net thermal radiation. The results have been shown in the response for Point 29. In general, we find it hard to derive direct relations between the difference of ENV-ICE SIT and meteorological parameters.

In addition, we also employed the 2 meter temperature parameter derived from ERA-5 reanalysis provide by Copernicus Climate Change Service (C3S) (2017) to generate the accumulative freezing-degree-days (FDD). We added the information in this section: "In this study, we use 2 meter temperature data derived from ERA-5 reanalysis provide by Copernicus Climate Change Service (C3S) (2017) to generate the accumulative freezing-degree-days (FDD). FDD is calculated by daily degrees below freezing summed over the total number of days the temperature was below freezing. Here the freezing point is set to -1.8 degrees Celsius for ocean water. According to Lebedev (1938), we construct a simple model to produce sea ice thickness and examine the growth from autumn (MJ) to spring (ON) every year:

Thickness (cm) = 1.33 * FDD (℃)$^{0.58}$

Then we can compare it with the thermodynamic growth represented by Envisat and ICESat SIT. " *(please see P7 line 200-206 in the revised manuscript)* And the results are shown in the response for Point 22.

**Point 10:** Line 151-153: - "For each period ... If ICESat data has overlapping time ..." –> Perhaps better: "We use Envisat data of every month which overlaps for at least ten days with a particular ICESat measurement period. For some ICESat measurement period this requires to average over Envisat sea-ice thickness data of two months (see Table 1)." - I note that period ON2005 starts October 21 and hence has 11 days in October with data while Envisat data are only taken from November. Please fix.

*Response:* Thanks for your comments, but here we chose to follow the suggestions from another referee and do a time-weighted average of the monthly Envisat data to match the ICESat period. For example, consider the ON04 period from Oct 3 to Nov 8 in 2004, which is 37 days long – 29 days in October and 8 days in November. So we calculate the corresponding Envisat SIT as: $SIT_{ON04}$ = (29/37)*($SIT_{Feb}$) + (8/37)*($SIT_{Mar}$). *(please see P8 line 211-213 in the revised manuscript)*

**Point 11:** Figure 1: I suggest to replace "sea ice freeboard" with "freeboard" in the caption of this

figure to underline that you are targeting to illustrate both, total freeboard and sea-ice freeboard.

*Response:* We amended it as your suggestion.

[Figure]

Figure 2. An illustration of measuring freeboard using ICESat and Envisat.

**Point 12:** Figure 2: - Did you create this map on your own or is this taken from the publication / data website? If the latter you either need to get the permission or at least state that you modified if from ... - What are the numbers in the white boxes referring to? - Please provide a table with the respective operation times of the ULS which overlapped with your combined ICESat - Envisat sea-ice thickness data set. If I recall correctly not all of these were in operation all the time. - Please write what the background information is ... it is bathymetry.

*Response:*

(1) We produced this figure consulting Fig.2 in Behrendt et al. (2013).

(2) The numbers in the white boxes refer to the serial numbers of each ULS mooring location.

(3) We added a table for the data used in the comparison with ULS:

Table 3. Respective operation times of the ULS, Envisat and ICESat sea ice thickness data set during the comparison between ULS and satellite SIT.

|  | ULS | Envisat | ICESat |
| --- | --- | --- | --- |
| Site 207 | Apr 2005 to Mar 2008 | Apr 2005 to Mar 2008 | MJ05 to FM08 |
| Site 229 | Jan 2003 to Nov 2005 | Jan 2003 to Nov 2005 | FM04 to ON05 |
| Site 231 | Mar 2005 to Feb 2008 | Mar 2005 to Feb 2008 | FM05 to ON07 |

(4) The background is the land topography and ocean bathymetry from ETOPO1 Global Relief Model data (doi:10.7289/V5C8276M).

(5) Although we reproduced the figure in the following, we chose to remove this figure and marked the three sites used in the comparison in Fig. 4, shown in the response for Point 13.

[Figure]

Figure 3. Map of the AWI ULS mooring locations. The background is the land topography and ocean bathymetry from ETOPO1 Global Relief Model data (doi:10.7289/V5C8276M). The circles and the corresponding numbers in the white boxes refer to the sites of the ULS. The gray line refers to the coastline of the Antarctica.

Modified from Fig.2 in Behrendt et al. (2013).

**Point 13:** Figure 3: - The lines denoting the borders between the different regions is hard to see and should be made thicker or plotted using a different color. - Specify what the white grid cells stand for.

*Response:* This figure was modified as your comments:

[Figure]

Figure 4. Map of the different sectors referred to in the study. The background is the average of the September sea ice thickness from Envisat during 2003-2011 with 50 km grid size. Each sector and the two ice shelf polynyas are indicated in the figure. The circles and the corresponding numbers refer to the sites of the ULS. The white grid cells stand for area with sea ice concentration less than 70% or missing data.

**Point 14:** Lines 159-166: - How were ULS data co-located with the satellite data products? Also, I note that Envisat is on EASE grid but ICESat is on polar-stereographic grid. Which one did you choose? - What do you mean by "absolute" in Line 159? - "in situ" –> ULS is not an in situ measurements. - What does "-6" "-5" appended behind the ULS station numbers mean? - I don't think "deformed ice" and "thin ice" are particularly well chosen because also thin ice can be deformed. It might be better to refer to old or perennial ice in case of the location close to the peninsula - albeit local polynyas might produce an applicable amount of seasonal ice as well - and predominantly seasonal ice in case of the Eastern Weddell Sea.

*Response:*

(1) In this study, the comparisons were carried by linearly interpolating the two satellite SIT data onto each ULS location. *(please see P8 line 223-224 in the revised manuscript)* For the inter-comparison we chose the polar-stereographic grid from ICESat and interpolated Envisat SIT onto this grid.

(2) We used "absolute" to imply that compared with the results of inter-comparison between Envisat and ICESat, the comparison with observation from ULS allows us to give a certain statement on which is more accurate. We decided to delete "absolute" because "accuracy" is enough to express

the meaning.

(3) We substituted "in situ observations" by "field observations".

(4) The number appended behind the ULS station numbers refer to different ULS operation periods at each location. For example, "207-6" represents the sixth leg of measurements at the station #207. However, we combined the successive records for 229 and 231 as your comments below, so we changed the expression here as: "there are only three sites having enough valid data for the evaluation: 207, 229 and 231". *(please see P8 line 227-228 in the revised manuscript)*

(5) We have amended this: "Site 207 is near the coast of the Antarctic Peninsula, mostly characterized by perennial ice, while the others belong to the Eastern Weddell Sea, predominantly characterized by seasonal ice." *(please see P8 line 229-230 in the revised manuscript)*

**Point 15:** Lines 167-181 / Figure 4: - It appears to me that the number of ICESat values used in this comparison are so small that it is almost impossible to provide an adequate statistics. In order to make the comparison look at bit more reasonable I suggest to combine the two successive records for 229 and 231, i.e. plot time series for the combined data set 229-5 and 229-6 as well as 231-6 and 231-7. That way you would not be forced to draw conclusions from just two data pairs. In addition, it appears to me that you computed RMSD values based on this completely different statistics and I suggest that you add a comparison between ULS data and Envisat data at the same sampling in time provided by ICESat. In that case you would need to specify whether you'd average over May and June in the Envisat product to obtain a value representing the ICESat May/June measurement periods. - Please adequately note in the text that you take uncertainty information from the data sets, i.e. that both data sets, the ICESat and the Envisat one provide uncertainty information based on the retrieval. In this context you need to take into account that the uncertainty of the Envisat is possibly reduced by a factor of 1/sqrt(n) with n being the number of grid cells used to compute the 100km mean gridded version of the Envisat sea-ice thickness. - I suggest to rewrite the full paragraph once you have this third set of more sparsely sampled Envisat data. Please when re-writing it provide systematically the information about the range of differences and the RMSD for all data sets, i.e. ICESat - ULS, Envisat - ULS, Envisat_sparse - ULS for 207, 229 5+6, 231 6+7 - ideally in a table. I would not over-interprete grid cells with "open water" in the satellite products because values might simply be missing due to other factors. In your discussion of these results you should remind the reader that ULS is a measurement device obtaining very local data of sea ice which passes / develops over it - in your case - over the course of a month. My question is: how accurately did you take the exact measurement period of ICESat into account here? Did you average over the, e.g. 35 days of a period or did you average over the calendar months? In addition: you compare one sea-ice thickness value from a 100 km x 100 km large grid cell with a value representative of a footprint of several 10 meters (I assume). This has an influence on both the sea-ice thickness as well as your estimate of the open water fraction obtained to carry out the comparison using the grid-cell mean sea-ice thickness.

*Response:*

(1) We combined the successive records and reproduced this figure:

[Figure]

Figure 5. Time series of sea ice thickness and their errors for the Weddell Sea ULS, Envisat and ICESat. The numbers on the top of each panel represent the location of each site for the comparisons. The site locations can be searched in Fig. 2. ICESat SIT values are placed between the two months that each period covers. The mean differences (MD) and their standard deviations (SD) are shown in the figures.

(2) We calculated the mean differences (D), standard deviations (SD), root mean square deviations (RMSD) values at the same scale ICESat SIT for each site and the total samples. The Envisat and ULS SIT are all time-weighted corresponding to ICESat operating periods.

Table 4. Statistical results of the comparison between two satellite SIT data with ULS data.

|  | Envisat-ULS | | | ICESat-ULS | | |
| --- | --- | --- | --- | --- | --- | --- |
|  | D (m) | SD (m) | RMSD (m) | D (m) | SD (m) | RMSD (m) |
| 207 | 1.01 | 1.00 | 0.91 | 0.60 | 0.74 | 0.66 |
| 229 | 0.49 | 0.10 | 0.07 | -0.04 | 0.09 | 0.07 |
| 231 | 0.59 | 0.12 | 0.10 | 0.27 | 0.36 | 0.34 |
| Total | 0.78 | 0.72 | 0.69 | 0.32 | 0.52 | 0.50 |

(3) The Envisat error bars come from the SIT uncertainty $\sigma_{sit}$ contained in the Envisat SIT product and is computed as the error propagation of all input uncertainties with the assumption that the sea water density is negligible (Paul et al. 2017):

$$\sigma_{sit} = \sqrt{(\frac{\rho_w}{\rho_w - \rho_i}\sigma_{frb})^2 + (\frac{frb \cdot \rho_w + sd \cdot \rho_i}{\rho_w - \rho_i}\sigma_\rho^i)^2 + (\frac{\rho_s}{\rho_w - \rho_i}\sigma_{sd})^2 + (\frac{sd}{\rho_w - \rho_i}\sigma_\rho^s)^2}$$

where *frb* represents Envisat sea ice freeboard, *sd* represents snow depth, $\rho_w$, $\rho_s$, $\rho_i$ refer to the density of the sea water, snow cover and sea ice, $\sigma_{frb}$, $\sigma_\rho^i$, $\sigma_{sd}$, $\sigma_\rho^s$ represent the uncertainties of sea ice freeboard, ice density, snow depth and snow density, respectively.

ICESat SIT uncertainties $\sigma_I$ are calculated based on the uncertainties of sea ice and snow, also neglecting the uncertainty of water density (Li et al., 2018):

$$\sigma_I = \sqrt{(\frac{R * F * \rho_w}{(R + 1) * (\rho_w - \rho_i^*)})^2 \sigma_{\rho_i}^2 + (\frac{F * \rho_w}{(R + 1) * (\rho_w - \rho_i^*)})^2 \sigma_{\rho_s}^2}$$

where $F$ represents ICESat total freeboard, $R$ represents the sea ice thickness to snow depth ratio, $\rho_w$, $\rho_s$, $\rho_i^*$ refer to the density of the sea water, snow cover and the modified density of sea ice, $\sigma_{\rho_i}$, $\sigma_{\rho_s}$ represent the uncertainties of sea ice density and snow density, respectively.

Therefore, we added the information as: "We draw the error bars from uncertainty information provided by both SIT products. The Envisat SIT uncertainty is computed as the error propagation of all input uncertainties with the assumption that the sea water density is negligible (see Section 2.9.8 in Paul et al. 2017). ICESat SIT uncertainties are calculated based on the uncertainties of sea ice and snow, also neglecting the uncertainty of water density (see Eq. (6) in Li et al., 2018)." *(please see P8 line 237-240 in the revised manuscript)*

(4) We understand that the uncertainty of a remote sensing product might be possibly reduced with more valid measurements taken into account in one grid cell, i.e. by enlarging its grid cells. However, we think that the 100 km grid version of Envisat SIT might not significantly reduce the uncertainties. The thickness uncertainty is based on a random (noise) component, which is already quite low on 25 km scale, let alone 50 km. The bulk uncertainty comes from erroneous snow depth, ice density and freeboard biases and they are not reduced by coarser gridding.

(5) We agreed and changed the "open water" with "grid cells where sea ice concentration below 60%, or missing data caused by failure of retrieval or instrument". *(please see P9 line 247-248 in the revised manuscript)*

(6) We calculated the monthly average ULS SIT over the calendar months the same as Envisat SIT. During the comparison with ICESat SIT, we also did time-weighted average on monthly ULS SIT. Therefore, we added that "We calculate the RMSD values for Envisat-ULS and ICESat-ULS at the same time scale as valid ICESat SIT. It is noted that ULS measurements obtain very local data of sea ice which passes or develops over it over the course of a month. The Envisat and ULS SIT are all time-weighted processed." *(please see P9 line 250-252 in the revised manuscript)*

(7) The discussion of uncertainties caused by different footprints have been added in Section 3.1 as the response for Point 8. *(please see P9 line 259-271 in the revised manuscript)*

**Point 16:** Figures 5 to 7: - Please consider extending the latitudinal range further to the north so that you do not truncate the ice-covered area. - Please consider to show all maps with the same size.

Currently, the maps for autumn are largest while those for summer are smallest. Having three maps in a row (as in Figure 7) seems to be a viable solution; hence, if you transpose rows to columns and show the years in rows rather than columns you can realize my suggestion. - I note that you show the two sea-ice thickness data sets at their native grid resolution while the difference map (of course) is in the grid resolution of the ICESat product. Please make a note in the caption of Figure 5 to clarify that this is done on purpose. - I assume that white denotes missing data? This is fine (as long as stated once in the caption) for the sea-ice thickness maps but interferes with grid cells having a valid sea-ice thickness value in both data sets but a difference close to zero; in the difference maps the color white is not used uniquely - which I suggest to change. - I note a vertical line of grid cells without valid values just west of the date line, i.e. 180°E. Please consider removing this / filling it with valid difference values. - Cosmetics: The color bar would be sufficient with half its horizontal width. Am I correct in assuming that the quantity displayed is sea-ice thickness or sea-ice thickness difference? Consider you want to uses these figures in a presentation - it cannot go without the caption the way you solved it now. If you would state along with the color bar which physical quantity is shown the figures would be self-explainable. You could combine this information with the information about the product which currently is provided left of the maps.

*Response:* Thank you for your comments. The figures are modified as follows:

[Figure]

Figure 6. Comparisons of Envisat versus ICESat sea ice thickness for each ICESat operating period in spring (October & November). The first and second columns show the sea ice thickness distribution of Envisat and ICESat respectively, and the last column shows the difference map (Envisat minus ICESat) of sea ice thickness. Each row represents a year from 2004 to 2007. The sea ice thickness maps are at their native grid resolution while the difference map is interpolated onto the polar-stereographic grid of the ICESat product. The white cells denote sea ice concentration less than threshold or missing data.

[Figure]

Figure 7. Same as Fig. 5 but for summer (February & March).

[Figure]

Figure 8. Same as Fig. 5 but for autumn (May & June).

**Point 17:** Table 2: - Please consider adding the standard deviation of the mean for the columns denoted "ENV (m)" and "ICE (m)". Ideally, you do this after the interpolation of the Envisat data to the ICESat resolution.

*Response:*

Table 5. Statistical results of the comparisons between Envisat sea ice thickness and ICESat sea ice thickness for each ICESat operating period. The correlation coefficients (CC) in italic type have not passed the 95% significance test.

|  |  | ENV(SD) (m) | ICE(SD) (m) | Difference (SD) (m) | RMSD (m) | CC | N |
|---|---|---|---|---|---|---|---|
|  | Seasonal average | 2.30(0.73) | 1.79(0.71) | 0.51(0.70) | 0.72 | 0.47 | 224 |
| Summer (FM) | 2004 | 2.80(0.68) | 1.91(0.80) | 0.89(0.82) | 0.78 | 0.33 | 115 |
|  | 2005 | 3.05(0.71) | 2.31(0.83) | 0.74(0.76) | 0.75 | 0.45 | 97 |

| | | 2.57(0.68) | 2.10(0.76) | 0.47(0.96) | 0.93 | 0.02 | 82 |
|---|---|---|---|---|---|---|---|
| | 2006 | 2.57(0.68) | 2.10(0.76) | 0.47(0.96) | 0.93 | 0.02 | 82 |
| | 2007 | 2.30(0.77) | 1.69(0.83) | 0.61(0.84) | 0.84 | 0.39 | 182 |
| | 2008 | 2.74(0.72) | 1.82(0.65) | 0.92(0.88) | 0.85 | 0.04 | 105 |
| Autumn (MJ) | Seasonal average | 1.89(0.67) | 1.36(0.57) | 0.53(0.49) | 0.49 | 0.68 | 838 |
| | 2004 | 1.94(0.72) | 1.33(0.62) | 0.61(0.51) | 0.51 | 0.71 | 718 |
| | 2005 | 1.98(0.80) | 1.43(0.68) | 0.55(0.60) | 0.60 | 0.68 | 705 |
| | 2006 | 1.92(0.64) | 1.32(0.59) | 0.60(0.57) | 0.56 | 0.57 | 610 |
| Spring (ON) | Seasonal average | 1.57(0.50) | 1.58(0.57) | -0.01(0.44) | 0.43 | 0.64 | 1205 |
| | 2004 | 1.64(0.59) | 1.64(0.68) | 0.00(0.54) | 0.54 | 0.63 | 885 |
| | 2005 | 1.59(0.60) | 1.54(0.67) | 0.05(0.59) | 0.59 | 0.56 | 740 |
| | 2006 | 1.44(0.53) | 1.63(0.67) | -0.19(0.54) | 0.54 | 0.61 | 705 |
| | 2007 | 1.71(0.58) | 1.57(0.61) | 0.14(0.56) | 0.56 | 0.55 | 927 |

**Point 18:** Lines 186-202: - Line 186: Delete "And" - Line 187: Who is "They"? - Line 189-191: I don't find the spring maps clear enough to make the statement given about the Ronne Ice Shelf polynya. The other issue, that Envisat kind of fails to see the young in in the southern Ross Sea is something which is noted in the above-mentioned (by you) ESA-SICCI-2 project PVIR report; you might want to cite this here. - Line 191: I don't think that this recirculation of thicker ice from the Amundsen Sea into the Ross Sea is an anonamous phenomenon. You might consider removing this statement. - Line 193/194: What are "posivite" or "negative deviation" between Envisat and ICESat SIT? Perhaps it would be better to speak about "Envisat SIT exceeds ICESat SIT" ... and "Envisat SIT is below ICESat SIT" ... or similar. - Line 194-195: Isn't this statement a bit hypothetic? In the Arctic, for example, maximum SIT values are observed in April or even May with little change between April and May. In addition, since you compare grid-cell mean SIT values you might have a substantial influence of a potentially lower sea-ice concentration. It would be interesting to look at the pure SIT values, wouldn't it? - Line 195/196: Where is this statement substantiated from?

*Response:*

(1) We deleted "And".

(2) We changed "They" to "Envisat and ICESat SIT are able to ..." *(please see P10 line 278 in the revised manuscript)*

(3) The spring ICESat SIT maps show a small area of thin ice at the Ronne Ice Shelf from 2005 to

2007. However, this situation is indeed not clear in 2004. Therefore, we modified the sentence that: "The same is found for the Ronne Ice Shelf polynya in the Weddell Sea expect in 2004." *(please see P10 line 282 in the revised manuscript)*

(4) We added the citation (Kern et al., 2018) here.

(5) We were meant to give an explanation on this phenomenon, but we will remove this statement as suggested because this is not critical.

(6) We amended it as: "Envisat SIT exceeds ICESat SIT in 2004, 2005 and 2007, while below it in 2006." *(please see P10 line 284-286 in the revised manuscript)*

(7) We examined the monthly mean pure sea ice thickness from 2004 to 2007, shown in Fig. 9. The results show that all the maximum SIT values appear in October during the four years. Therefore, we can attribute the reason that SIT in 2005 and 2006 are smaller than 2004 and 2007 to this. However, now we changed the way to deal with the Envisat corresponding time periods. Based on the updated calculation, Envisat SIT is only below ICESat SIT in 2006. Therefore, we considered removing this statement. To give an explanation to the inter-annual variations, we examined the 2 meter temperature anomaly corresponding to the comparison time period from 2004 to 2007 with respect to the four years average, shown in Fig. 10. There are positive biases in Bell&Amund Sea in 2005 and Weddell Sea in 2006. Both Envisat and ICESat react to these biases. *(please see P10 line 286-287 in the revised manuscript)*

[Figure]

Figure 9. Time series of the monthly mean sea ice thickness of Envisat from 2004 to 2007.

[Figure]

Figure 10. The 2 meter temperature anomaly maps corresponding to ICESat periods in spring with respect to the four years average.

(8) This is a hypothetical explanation for the inter-annual variations of the difference maps based on last statement. But we do not have evidence to prove it. Meanwhile, since the results of the inter-annual variations have changed, we decide to remove this statement.

**Point 19:** Lines 203-213: - Line 203: I note that FM-periods are at the end of summer and mark the begin of the freezing season; in 2007 this period is even shifted by one month towards winter (March-April). It might therefore be better to refer to this period as autumn or summer-autumn transition and denote the May/June period as winter. Or simply change the beginning to "At the end of summer melt (FM), ..." - Lines 208/209: "may result": Don't you have sea-ice concentration data provided along with the Envisat SIT data? You can simply check this out and make a more clear statement here. - Lines 209/210: "Envisat also detects the Ross sea ice ..." –> I guess you wanted to refer to the Ross Ice shelf polynya and the sea-ice thickness? - Line 211: See my previous comment about this formulation. - "is bad" –> please change to something without valueing, e.g. "We find larger RMSD values ... by XY and smaller correlation values ... by XY than for spring.

*Response:*

(1) We divided the seasons following Kurtz and Markus (2012), and we agreed to change the beginning to "At the end of summer melt (FM), ...". *(please see P10 line 296 in the revised manuscript)*

(2) We checked the sea ice concentration for ICESat and confirm that the lack of data is due to the concentration threshold. Therefore, we deleted "may" here.

(3) We changed it to "... detects sea ice thickness in the Ross Sea ..." *(please see P10 line 303 in the revised manuscript)*

(4) We amended it to "Generally, Envisat SIT exceeds ICESat SIT of 0.51 m in summer, the largest difference among the three seasons." *(please see P10 line 304-305 in the revised manuscript)*

(5) We amended it to "We find largest RMSD values among the three seasons of 0.72 m and smallest correlation values of 0.47 than for spring." *(please see P10 line 305-306 in the revised manuscript)*

**Point 20:** Lines 214-224: - Line 215: Which "errors" are you referring to here? I guess you still refer to the differences between Envisat and ICESat SIT, right? So there is no error involved. - Line 218: Which "deviations"? To which panel of Figure 7 are you referring? I only find differences. - Line 219: "The ice growth ..." –> I suggest to delete this sentence as it is not relevant in this context. - Line 221: "from to thinner" –> "from thicker to thinner" - Line 223: "... are high among all periods ..." –> I assume you wanted to write "highest among"? - Line 224: I suggest to add 1-2 sentences pointing out that correlations are substantially better for 2004 and 2005 than 2006 while, at the same time, the coverage with valid Envisat SIT data (Fig. 7, top row) is considerably smaller than for ICESat (Fig. 7, middle row). This needs to be discussed: two months for 2004 and 2005, including May, only 1 month (June) for 2006; since MJ periods are located in the seasonal upswing of sea-ice coverage and thickness it can make a big difference to include or exclude one month of data. I also suggest to point out the inter-annual variability in data pairs with valid data for each season. There a considerable variations in the number of valid data pairs within one season - mostly because of data gaps in the ICESat SIT data set.

*Response:*

(1) We changed it to "the differences between Envisat and ICESat SIT". *(please see P11 line 310 in the revised manuscript)*

(2) We changed "deviations" to "differences".

(3) We added the missing "thicker".

(4) We changed it to "highest".

(5) Agreed. Although we changed the way to produce the corresponding Envisat SIT and used both months data for each period, the total coverage is an intersection of both months and the problem still exists. Therefore, we added several sentences to discuss this: "It is noted that correlations between Envisat and ICESat SIT are substantially better for 2004 and 2005 than 2006 while, at the same time, the coverage with valid Envisat SIT data (Fig. 7, top column) is considerably smaller than for ICESat (Fig. 7, middle column). In this study, we use time-weighted Envisat SIT for the comparison, indicating that only valid data for both months are used. Since MJ periods are during the seasonal upswing of sea-ice coverage and thickness, it can make a big difference between two months coverage." *(please see P11 line 320-324 in the revised manuscript)*

(6) We agreed and added: "Also, the inter-annual variability in data pairs with valid data for each season is noteworthy. There are considerable variations in the number of valid data pairs within one season, mostly because of data gaps in the ICESat SIT data set." *(please see P11 line 324-326 in the revised manuscript)*

**Point 21:** Figure 8: - I suggest to re-order the panels to [end-of-]summer (left), winter (middle)

and spring (right) to have the correct time line of development. This would ease discussion about whether mean (and modal?) SIT increase from winter to spring - which is what should be the case. - I am wondering about the coarse bin-size of 0.5 meters; did you experiment with smaller bin-sizes of, e.g. 0.25 m or 0.2 m. If not, what is your motivation to use such a large bin size. Could that be the accuracy? - I suggest to use blue and red also as font colors for the annotation within the panels, make the rectangles in a) simple lines, and add the modal value, ideally both as a (dashed) vertical line and as a numner. Note that the modal SIT value is a better measure for thermodynamic sea ice thickness growth than the mean SIT. - Are the normalized to have sum 1?

*Response:*

(1) We modified the figure as suggested:

[Figure]

Figure 11. Probability of the Envisat SIT and the ICESat SIT for all the individual comparison pairs. The blue stairs represent Envisat ice thickness and the red stairs represent ICESat ice thickness. The solid lines indicate the modal ice thickness and the dashed lines indicate the mean ice thickness of both data sets.

It is noted that since we changed the way to calculate the corresponded Envisat SIT, there is a little difference with the figure in the previous manuscript.

(2) We use 0.5 m for bin-size because this is enough for us to look into the general distributions. But to make the distributions clearer, we changed it to 0.2 m.

(3) Yes, the sum of each normalized distribution is 1.

**Point 22:** Lines 225-238: - Line 226: "... agreement ... turns bad ..." –> please re-phrase and omit "bad". - Line 231: "Both of the two data sets cannot detect the thin ice smaller than 0.5 m ..." –> Looking at Fig. 8 a) and b) reveals that there are very small but non-zero fractions of SIT values y 0.5 m. I suggest to rewrite: "We find that thin ice with thickness values < 0.5 m is practically absent in both investigated products." In order to avoid the conclusion that ICESat or Envisat are not capable to resolve such thin ice you might want to add a statement into that direction. - Lines 232-235: This passage should be re-written. It is not clear what you are referring to here. Of course the

mean (!) SIT is larger in summer than winter or spring because any young ice or thin ice is missing and the only grid cells left with sea-ice coverage contain - at that time of the year which is actually end-of-melt - thick ice; this is comparable to the situation in the Arctic during the sea ice minimum in September. One should not refer to "growth" or "grow" here. What is more interesting is that the mean SIT derived with ICESat increases from MJ to ON periods while the Envisat one does not. As commented in the context of Figure 8 I also encourage to include modal SIT values into the discussion. I recommend to delete the "The rest can be ..." sentence. - Line 236: "weaker" –> "smaller"; in addition you mention warm and wet weather condition and flooding sometimes. Does this refer to summer only or to the entire year? You might want to specify the key issue for penetration depth here which is wet snow as a consequence of, for example, above-freezing air temperatures - Line 237/238: "Since ICESat ..." –> This sentence should be deleted because it is wrong. ICESat is much better suited to detect thin ice than Envisat thanks to its small footprint size.

*Response:*

(1) We changed the sentence to "the distribution in summer is quite unmatched". *(please see P11 line 330 in the revised manuscript)*

(2) Agreed. We changed the sentence to "We find that thin ice with thickness values < 0.5 m is practically absent in both investigated products." *(please see P12 line 336-337 in the revised manuscript)*

(3) We recognized this negligence and focused on the different variations of the two data sets from autumn to spring. We added the analysis: "We notice that Envisat and ICESat mean SIT have different variations from autumn to spring. ICESat SIT increases while Envisat SIT does not. For Envisat SIT, the modal (mean) values change from 1.70 m (1.95 m) to 1.30 m (1.60 m) and the distribution indicates that more thin ice is produced. But for ICESat SIT, the modal SIT stays constant of 1.1 m, while the mean values vary from 1.36 m to 1.60 m, and the distribution indicates that more thick ice is found. Since May-June period is the beginning of the upswing freezing season and October-November period is near the end of it, we assume that thermodynamic growth contributes more to the SIT variations than dynamic growth. Therefore, we further compare this development with the FDD model results in 2004, 2005 and 2006, shown in Table 6. As introduced in Sect. 2.4, this model is only involved in the temperature, representing the thermodynamic growth and neglecting any dynamic contributions to sea ice thickness. Figure 8 shows the sea ice thickness growth maps from autumn to spring derived from FDD model, Envisat and ICESat in 2004, 2005 and 2006. Table 6 shows the statistical results of the growth maps. We calculate the period-average SIT from the model corresponding to ICESat operating periods. The results show that Envisat SIT has an opposite development from ICESat and model. And the development of ICESat SIT is smaller than and not constant as the model results. Although ICESat SIT has an overall increase, it also shows a decrease in Western Weddell Sea similar to Envisat SIT. In addition, both of them show an increase along the coast of Amundsen Sea in 2006. Combined with the typical sea ice motion in the Weddell Sea and Ross Sea shown in the first panel, we consider that ICESat shows more realistic sea ice change, e.g. a decrease SIT in the Ross Sea polynya and increase around it, or the increase

[Figure]

in the East Weddell Sea." *(please see P12 line 339-355 in the revised manuscript)*

Figure 12. The sea ice thickness growth maps from autumn to spring derived from FDD model, Envisat and ICESat in 2004, 2005 and 2006. The arrows refer to the typical sea ice motions in the West Weddell Sea and Ross Sea.

Table 6. Statistical results of the sea ice thickness development from autumn (MJ) to spring (ON) in 2004, 2005 and 2006.

|  | Envisat SIT (m) | ICESat SIT (m) | FDD (m) |
|---|---|---|---|
| MJ04 | 1.94 | 1.33 | 0.55 |
| ON04 | 1.64 | 1.64 | 1.11 |
| ON04-MJ04 | -0.30 | 0.31 | 0.56 |
| MJ05 | 1.98 | 1.43 | 0.57 |
| ON05 | 1.59 | 1.54 | 1.13 |
| ON05-MJ05 | -0.39 | 0.11 | 0.56 |
| MJ06 | 1.92 | 1.32 | 0.58 |

| | | | |
|---|---|---|---|
| ON06 | 1.44 | 1.63 | 1.14 |
| ON06-MJ06 | -0.48 | 0.31 | 0.56 |

(4) We deleted the "The rest can be ..." sentence.

(5) We changed "weaker" to "smaller". Warm and wet weather condition mostly occur in summer, which we suppose causes the large positive difference between Envisat and ICESat SIT. Therefore, we modified the sentence to "Envisat's RA-2 altimeter will be affected by smaller penetration into snow depth due to the warm and wet weather condition when air temperatures are above freezing point, as well as flooding sometimes, measuring a larger sea ice freeboard and thus a larger sea ice thickness." *(please see P12 line 359-361 in the revised manuscript)*

(6) We realized the mistake and deleted the sentence.

**Point 23:** Lines 239-251: - I suggest to delete the very first sentence. It is fine to simply also show results broken down by region as is commonly done in many publications about sea-ice concentration and thickness products. - I find this paragraph not particularly well structured. Perhaps you can solve this by avoiding to jump between the different statistical parameters but stick with one and describe it in a comparative manner across the regions / seasons - ideally highlighting the most interesting issues. Clarification what you mean by "deviations" would also help - see respective previous comments. - Lines 248-251: I suggest to delete these sentences as they appear to be too hypothetical. Also, you start the discussion in the next section anyways.

*Response:*

(1) We agreed and deleted it.

(2) We modified the description: "We make further comparison for each region and each season as shown in Fig. 9. Due to the limited measurements in the Indian Ocean and western Pacific Ocean, we combine them into the whole Eastern Antarctic. For all the five panels, the regression lines have large positive intercepts in all three seasons. This indicates that Envisat SIT tend to exceed ICESat SIT for thin ice. From Fig.9a, we can see that in western Weddell Sea where covered perennial sea ice, sea ice thickness in summer and autumn override sea ice thickness in spring. This reveals that ICESat SIT are nearly constant in three seasons in western Weddell Sea while Envisat SIT are noticeably larger in summer and autumn than spring, which can also be seen from the statistical results shown in Table 6. From the end of melting season to the end of freezing season, Envisat SIT changes from 3.01 m to 3.16 m to 2.22 m, while ICESat SIT changes from 2.03 m to 2.28 m to 2.22 m. Considering the regional average differences between Envisat and ICESat SIT, the largest is in western Weddell Sea of 0.62 m and the smallest is in Ross Sea of -0.10 m, and they are smallest in spring for most regions, just as the seasonal results shown in Table 5. Regarding the spatial correlation, the largest coefficients is in Bellingshausen and Amundsen Sea of 0.59, albeit there is

no correlation in summer, and the smallest coefficients is in western Weddell Sea of 0.38." *(please see P13 line 376-387 in the revised manuscript)*

(3) Agreed. We deleted it.

**Point 24:** Figure 9: - I suggest to add the identity line in each scatterplot.

*Response:*

[Figure]

Figure 13. Scatterplots of the individual data pairs between Envisat SIT and ICESat SIT for each region and each season. Since the comparison pairs are too few in Indian Ocean and western Pacific Ocean, we combine these two regions into Eastern Antarctic. The respective correlation coefficients are indicated in the panels.

**Point 25:** Table 3: - I suggest to re-order seasons such that one can clearly see the change in SIT from autumn (or winter as I'd call it) to spring. - Please add a unit for RMSD. - I possible please provide standard deviations for the means of ENV and ICE (like I suggested already in the context of Table 2).

*Response:*

Table 7. Statistical results of the comparisons between Envisat sea ice thickness and ICESat sea ice thickness for each region divided as Fig. 9. The correlation coefficients (CC) in italic type have not passed the 95% significance test.

| | | ENV(SD) (m) | ICE(SD) (m) | Difference(SD) (m) | RMSD (m) | CC | N |
|---|---|---|---|---|---|---|---|
| W. Weddell | Regional average | 2.79(0.87) | 2.17(0.72) | 0.62(0.89) | 0.89 | 0.38 | 835 |
| | summer (FM) | 3.01(0.65) | 2.03(0.66) | 0.98(0.72) | 0.72 | 0.35 | 301 |
| | autumn (MJ) | 3.16(0.89) | 2.28(0.74) | 0.88(0.73) | 0.73 | 0.60 | 252 |
| | spring (ON) | 2.22(0.76) | 2.22(0.76) | 0.00(0.83) | 0.83 | 0.40 | 282 |
| E. Weddell | Regional average | 1.64(0.58) | 1.36(0.55) | 0.28(0.57) | 0.56 | 0.50 | 1924 |
| | summer (FM) | 2.57(0.70) | 1.84(0.69) | 0.73(0.73) | 0.71 | 0.45 | 152 |
| | autumn (MJ) | 1.85(0.53) | 1.08(0.42) | 0.77(0.41) | 0.41 | 0.65 | 617 |
| | spring (ON) | 1.47(0.46) | 1.47(0.51) | 0.00(0.46) | 0.46 | 0.55 | 1155 |
| Eastern Antarctic | Regional average | 1.48(0.60) | 1.58(0.72) | -0.10(0.63) | 0.63 | 0.49 | 969 |
| | summer (FM) | 2.48(0.82) | 2.35(1.01) | 0.13(0.97) | 0.95 | 0.39 | 44 |
| | autumn (MJ) | 1.63(0.54) | 1.50(0.64) | 0.13(0.58) | 0.57 | 0.41 | 325 |
| | spring (ON) | 1.34(0.52) | 1.55(0.68) | -0.21(0.60) | 0.60 | 0.50 | 600 |
| Ross Sea | Regional average | 1.72(0.45) | 1.41(0.57) | 0.31(0.53) | 0.53 | 0.43 | 1706 |
| | summer (FM) | 1.81(0.48) | 1.59(0.80) | 0.22(0.67) | 0.67 | 0.56 | 91 |

| | | | | | | | |
|---|---|---|---|---|---|---|---|
| | autumn (MJ) | 1.75(0.39) | 1.19(0.38) | 0.56(0.45) | 0.45 | 0.18 | 657 |
| | spring (ON) | 1.69(0.49) | 1.51(0.57) | 0.18(0.50) | 0.50 | 0.55 | 958 |
| Bell/Amund | Regional average | 2.06(0.67) | 1.88(0.79) | 0.18(0.65) | 0.64 | 0.59 | 546 |
| | summer (FM) | 2.35(0.52) | 2.22(0.84) | 0.13(0.87) | 0.83 | 0.10 | 28 |
| | autumn (MJ) | 2.05(0.61) | 1.64(0.72) | 0.41(0.53) | 0.53 | 0.68 | 217 |
| | spring (ON) | 2.03(0.70) | 1.99(0.79) | 0.04(0.66) | 0.66 | 0.58 | 301 |

**Point 26:** Lines 253-262: Title of this section: "deviation" –> I'd say "deviation" is used if one wants to describe how well a data set agrees to a known standard or calibration data set. This is not the case in your study. You inter-compare different data sets of which none is the standard. Therefore the term "difference" would match considerably better. - I suggest to delete the first sentence of this paragraph. You don't need this kind of introduction here. - Line 259/260: "respond differently to different surface roughness." –> This is true ... sure ... but how about resolving leads / open water required for an adequate representation of the local sea-surface height required to derive the free-board? Isn't this a much more important difference in the obervation capabilities of the two sensors? - Line 261: "... while ICESat uses ..." –> This reads as if the ICESats sea-ice thickness values are not retrieved using the hydrostatic equilibrium assumption ... but for both sensors this assumption is applied; ONE of the differences is in fact the treatment of the snow depth - here you are correct.

*Response:*

1) We changed "deviation" to "difference".

2) We deleted the first sentence of this paragraph.

3) Agreed. We focused on the footprint-dependent surface mixing in the discussion and change the sentence to "they are likely to respond differently to resolve leads or open water required for an adequate representation of the local sea-surface height during the freeboard retrieval". *(please see P13 line 395-396 in the revised manuscript)*

4) We modified the sentence to make it clearer: "Envisat directly uses the hydrostatic equilibrium with an extra AMSR-E snow depth product while ICESat also uses the hydrostatic equilibrium but accompanied with a modified snow–ice density method to get rid of the biased remote sensing snow depth." *(please see P13 line 398-400 in the revised manuscript)*

**Point 27:** Figure 10: I have the same comments as I voiced in the context of Figures 5 to 7 in terms of latitudinal extent, the vertical line of missing data just west of 180E and the fact that the Figure cannot be used on its own because the annotation of the color bar is not complete.

*Response:* We modified the figure as suggested:

[Figure]

Figure 14. Seasonal average differences between Envisat and ICESat sea ice thickness for the three seasons.

**Point 28:** Figure 11: - The maps in this figure would need to be changed according to changes noted for Figure 10. - Relative humidity is typically given in percent, i.e. your scale would go from 70% to 100% and would be entitled "Relative humidity [percent]". - Do you think these values are credible? I can agree with the fringe of low values around the continent, indicating the impact of the catabatic air flow ... but why is the relative humidity higher over sea-ice covered areas than over open water areas? Isn't open water a much more efficient source of evaporation and hence input of water vapor into the lower troposphere? The only reason I could think of why the relative humidity is higher over sea ice than open water is because of the colder temperatures. I note that the difference between open water and sea ice is most pronounced for the cold May/June period compared to the milder October/November period. This is kind of in line with my argumentation. - It does make a difference whether one uses saturation with respect to a water or to an ice surface when computing the vapor pressure and hence the relative humidity. Which of these is used for the relative humidity you plot here?

*Response:*

(1) We modified the figure as suggested:

[Figure]

Figure 15. Seasonal average relative humidity distributions for the three seasons.

(2) We agreed with your argument. Sea ice is a barrier between atmosphere and ocean and decreases the evaporation of the sea water. Regarding the water vapor pressure, it should be larger over the ocean than the sea ice cover. However, relative humidity involves humidity and temperature at the same time during calculation. According to Fig. 16 in the response for Point 29, the 2 meter temperatures are much smaller over sea ice than the ocean, leading to a smaller saturation vapor pressure. Therefore, the relative humidity is higher over sea-ice covered areas than over open water areas. And this is why we would like to investigate the relation between the ENV-ICE SIT difference and relative humidity. However, we also realized that this data has a lot of uncertainties.

(3) Here the saturation vapor pressure is computed against water.

**Point 29:** Lines 264-278: - Line 268: "weaken" –> "decrease"; "into the snow-ice interface" –> "into the snow" - In the context of what is written here and in the paragraph Line 279++, I recommend to read » Kwok and Kacimi, 2019: Three years of sea ice freeboard, snow depth, and ice thickness of the Weddell Sea from Operation Icebridge and CryoSat-2 and the paper by » Paul et al., 2018: Empirical parametrization of Envisat freeboard retrieval of Arctic and Antarctic sea ice based on CS-2: progress in the ESA CCI . These papers are good preparations for a thorough discussion of the issues touched upon. - Lines 271++: I don't want to rule out that the atmospheric humidity might have had an impact on Envisat RA2 measurements that differs from the impact on ICESat measurements. However, I would expect that this would involve the entire troposphere and not the near surface relative humidity. I am sceptical, however, whether the near surface humidity, which is second order parameter, is a good measure to judge whether a snow surface / cover is wet or dry. I would have thought that the air-temperature is the more direct driver. Please check.

*Response:*

(1) We changed "weaken" to "decrease" and changed "into the snow-ice interface" to "into the snow".

(2) Thank you for your guidance. We have read the two papers and modified our discussion of

section 4 in the revised manuscript.

(3) We plotted the seasonal average maps of the four parameters as you suggested including 2 meter temperature, sea surface temperature, surface net solar radiation and surface net thermal radiation, shown in Fig. 16. All of them come from ERA-Interim reanalysis data. We also calculated the linear correlation coefficients between the difference of ENV-ICE SIT and each meteorological parameter in each season and the results are shown below in Table 8. We can see that the best linear correlation coefficient is the relative humidity and the others are all bad, albeit the relation with humidity is also weak. We assume that there might be relations between the difference and temperature and radiation, but they are absolutely not linear relations. In order to dig into this relation, more analyses are needed. However, this is not the kernel in this study. Therefore, we considered removing this part.

[Figure]

Figure 16. Seasonal average maps of 2 meter temperature, sea surface temperature, surface net solar radiation and surface net thermal radiation based on ICESat operating periods.

Table 8. The linear correlation coefficients between the difference of ENV-ICE SIT and each meteorological parameter in each season. The values in italic type have not passed the 95% significance test.

|  | Relative humidity | 2 meter temperature | Sea surface temperature | Surface net solar radiation | Surface net thermal radiation |
|---|---|---|---|---|---|
| ON | 0.28 | -0.12 | -0.14 | -0.24 | *0.01* |
| FM | 0.38 | -0.26 | -0.44 | -0.37 | 0.17 |
| MJ | 0.52 | -0.25 | -0.23 | -0.22 | 0.10 |

**Point 30:** Lines 279-284: - Line 280 "footprint is rather rough" –> I don't understand. How can a footprint be rough? The ice or snow surface can be rough. - That the detection of leads can affect the detection of thinner ice sounds trivial. If a sensor cannot resolve leads and/or resolve leads only indirectly and potentially also at the wrong position, then it is more than likely that thin ice thickness is not retrieved overly well. But what has this to do with "shorter ice"? What is this? What is "wider sea ice". I suggest to completely rewrite this paragraph after having studies the two paper I just recommended to read. Also, and this is what I cannot understand: You have two experts for radar altimetry based sea-ice thickness retrieval among your co-authors - so two people you could ask directly.

*Response:*

(1) We changed "rough" to "large".

(2) We modified this paragraph: "Envisat is a pulse-limited radar altimeter with a large footprint of 2 – 10 km and ICESat is a laser altimeter with a small footprint of about 70 m. This makes them receive different fractions of ambiguous signals caused by partial reflection different surface types in the footprint, e.g. ice and lead surfaces. Less surface-type mixing will occur for smaller footprints, subsequently allowing a better classification of lead and sea surface height retrieval in principal. Many studies have pointed that Envisat has uncertainties from such mixing (Schwegmann et al., 2016; Paul et al., 2018; Tilling et al., 2019). While this is directly applicable to radar altimeters with different footprints, the difference of lead detection skill between laser (ICESat) and radar (Envisat) altimeters is not directly a function of footprint size, since leads amplify radar backscatter and lead signatures, while the laser backscatter is a function of the surface albedo and thus leads return lower laser backscatter power. For different ice surfaces however, the lower footprint of ICESat has the capability to provide more detailed observations in areas with heterogenous ice conditions than the pulse-limited Envisat footprint." *(please see P15 line 437-447 in the revised manuscript)*

**Point 31:** Lines 286-302: - I suggest to repeat one more time a) Envisat SIT is based on an AMSR-E snow-depth climatology and b) ICESat SIT does not use snow depth in the retrieval but uses climatological ratios between snow depth and sea-ice thickness in form of the factor R to compute the modified density. This is a lot of "meat" for discussion. –> pitfalls of using a climatology instead of actual snow depth values (Bunzel and Notz, 2018, Retrievals of Arctic sea-ice volume and its

trends significantly affected by interannual snow variability); –> reduction of the relative uncertainties by using a climatology rather than actual values; –> limitations of the R-factor usage in the ICESat SIT data set (also based on climatology, and in addition based on ASPeCt observations with their known difficulties); –> discussion how the new approach by Li et al. (2018, mentioned above) would compare here; discussion of when which snow physical properties would cause which bias for which of the periods considered (i.e. winter compared to spring compared to summer). - Line 299/300: Kwok and Maksym (2014, see above) looked into this and figured out what typical snow depths appear to look like and Kern and Ozsoy-Cicek (2016, see above) figured out that indeed there is a substantial under-estimation. I recommend to refer to numbers from the respective publications to underline your assumptions and hypotheses. - Line 200/301: The sentence "Therefore, the sea ice ... negative." needs to be rewritten. - Line 302: "snow depth uncertainty over the deformed sea ice" –> Please be more specific. You know which sign this uncertainty would have (a positive bias!); hence, cases with actually a lot of deformed ice would have too little snow retrieved while cases with just a bit deformation would have fairly good retrievals. But: The ENV data set uses a climatology. Therefore, the largest effect might not come from deformed / level ice's impact on the snow-depth retrieval but the largest effect might be due an actual snow depth which differs completely from that represented by the climatology - no matter whether the sea ice is particularly deformed or not.

*Response:*

(1) We added the discussion of climatology uncertainties: "In this study, the snow depth climatology is employed during Envisat SIT retrieval process, which neglects the inter-annual snow variability. According to Bunzel et al. (2018), We find that the impact of using a snow climatology is small when the snow depth is thin. If the snow has reached a certain depth, thickness derived from it is unreliable." *(please see P16 line 481-484 in the revised manuscript)*

(2) We added the discussion of climatology advantages: "However, the usage of snow climatology allows to reduce the relative uncertainties than the actual values. The latter ones are affected by many factors as discussed above and have large uncertainties." *(please see P16 line 484-486 in the revised manuscript)*

(3) We added the discussion of the limitations of ICESat SIT: "the modified density used in the Worby retrieval algorithm does not consider the small-scale or regional variability of the snow depth, instead, only a seasonal constant density derived from the ASPeCt observations is given. Therefore, the largest uncertainty of ICESat comes from the potentially underestimations of the sea ice and snow observations for the computation of density (Kern et al., 2016)." *(please see P17 line 491-494 in the revised manuscript)*

(4) We added the discussion of the difference with ICESat SIT from Li et al. (2018): "This bias has been modified in Li et al. (2018), who derived first guess values of snow depth and sea ice thickness directly from ICESat data with empirical approaches, instead of the observation climatology used by Kern et al. (2016). This product has been demonstrated to be more reasonable than ICESat SIT used in this study because it takes the actual situation into account better. However, it is noted that

the empirical approaches used by Li et al. (2018) were developed from a suite of historic in-situ observations of freeboard, snow depth and sea-ice thickness which in a way have the character of a climatology as well." *(please see P17 line 494-499 in the revised manuscript)*

(5) In spring, large negative differences are observed in the western Weddell Sea, the Bellingshausen Sea, the western Pacific Ocean and the Indian Ocean, mainly along the coast, as shown in Fig. 10. Perennial ice with significant deformation exists near the Antarctic coast and in the western Weddell Sea and the Bellingshausen Sea, which could result in underestimations of snow depth (Kwok and Maksym, 2014; Kern and Ozsoy-Cicek, 2016). Considering Eq. (1), if sea ice freeboard F remains constant, then the Envisat sea ice thickness is biased low, and the difference that Envisat minus ICESat SIT is negative. Therefore, we attribute the negative differences shown in Fig. 4 to snow depth uncertainty. *(please see P16 line 474-480 in the revised manuscript)*

(6) We added the two citations about this statement. *(please see P16 line 477 in the revised manuscript)*

(7) We rewrote the sentence and clarified the effects: "Considering Eq. (1), if sea ice freeboard F remains constant, then the Envisat sea ice thickness is biased low, and the difference that Envisat minus ICESat SIT is negative. Therefore, we attribute the negative differences shown in Fig. 5 to snow depth uncertainty." *(please see P16 line 477-480 in the revised manuscript)*

(8) Agreed. We added the discussion of snow climatology effects in this part as the response for (1). *(please see P16 line 480-484 in the revised manuscript)*

**Point 32:** Lines 318-325: This paragraph would read better in Section 4 I suggest.

*Response:* Agreed. We moved this paragraph to Section 4. *(please see P16 line 488-494 in the revised manuscript)*

**Typos / editoral issues**

Line 11: "global" –> "hemispheric" because you consider the Antarctic not the entire Earth.

*Response:* Agreed.

Line 16: Suggest to add "these" between "with" and "field" to underline that the results of your comparison are only based on the ULS data.

*Response:* Agreed.

Line 42: "errors" –> I'd prefer to see "limitations"

*Response:* Agreed.

Line 43: "AEM data" –> Please tell the reader which parameter is derived from the AEM data.

*Response:* We added that "airborne electromagnetic (AEM) data which measure total freeboard (sea ice thickness plus snow depth) were collected ..."

Line 44: "only exists in the Weddell Sea" –> this is not correct. Please change to: "has mostly been obtained in the Weddell Sea"

*Response:* Agreed.

Line 45: - Why "time series"? These observations are not carried out at a specific point. - "freeboard observation" –> Yes, but there is an additional snow radar which allows additional snow depth estimates. Also, you might want to include the information whether the freeboard measured is the total or the sea-ice freeboard.

*Response:*

(1) We changed "time series" to "along-track data".

(2) We changed "freeboard observations" to "total freeboard and snow depth estimations".

Line 46: "limited to several trajectories in the Weddell Sea" –> This is not entirely true since there are also trajectories in the Bellingshausen Sea; more importantly, however, is that substantial work has been done with these data, e.g. Kwok and Maksym, 2014: Snow depth of the Weddell and Bellingshausen sea ice covers from IceBridge surveys in 2010 and 2011: An examination; and Kwok and Kacimi, 2019: Three years of sea ice freeboard, snow depth, and ice thickness of the Weddell Sea from Operation Icebridge and CryoSat-2; this work the the importance of this data for our understanding of sea-ice thickness retrieval of Antarctic sea ice using altimeter data should not be undervalued.

*Response:* We realized the importance of those works and have added this: "which have been investigated in some valuable studies previously (e.g. Kwok and Maksym, 2014; Kwok and Kacimi, 2019)."

Line 47: ULS data provide primary the sea-ice draft; sea-ice thickness is a derived variable. I suggest to change this to draft.

*Response:* Agreed. We changed "thickness" to "draft".

Line 48: "the basin ..." –> "a basin ..."

*Response:* Agreed.

Line 50-52: - Bernstein et al. (2015) –> I suggest to add Li et al. 2018: Spatio-temporal variability of Antarctic sea-ice thickness and volume obtained from ICESat data using an innovative algorithm

- I suggest to remove passive microwave and SAR here to remain concise and then stress that altimetry has proven to currently be the best source for Antarctic wide sea-ice thickness retrieval over the full thickness range.

*Response:*

(1) We cited Li et al. (2018) here.

(2) We removed the part of SAR and stressed the advantage of altimetry.

Line 55: "three"? –> I guess it is two decades because ERS1/2 is in progress but not yet ready to be released.

*Response:* We wrote "three decades" here to refer to the goal of this project, but of course the SICCI-1&2 only releases two decades of sea ice thickness data set. We have change it.

Line 58: "one of the latest ..." –> Perhaps latest is not the predominant issue here but the fact that it is finally a combination of CryoSat-2 AND Envisat.

*Response:* We added this information: "The SICCI product covers the entire Antarctic sea ice for the complete annual cycle from 2002 to 2017, and it is finally a combination data set of CryoSat-2 and Envisat."

Line 71: "estimates ... depth)" –> "allowed to estimate the total freeboard (sea ice freeboard + snow depth) via determination of the surface elevation"

*Response:* Agreed.

Line 74: I suggest to write "compared" instead of "developed".

*Response:* Agreed.

Line 76: "period" –> "periods"

*Response:* Agreed.

Line 78-81 / 88 - Since you defined ULS already there is not need to again write it in full length. - The same applies to airborne electromagnetic EM sounding which should be "AEM sounding" in Lines 79. - I am sure the PVIR has an author team so you can refer to it as "author et al., 2018". - I am relatively sure that the evaluation this report refers to also included CryoSat-2 data. Please check and amend accordingly.

*Response:* We changed these to abbreviations and modified the citation as: "The Envisat and CryoSat-2 sea ice thickness data in the Antarctic has already been evaluated with the drilling data, AEM, ULS and some ship-based data (Kern et al., 2018)."

Line 96/97: The URL given only points to the Envisat sea-ice thickness data set; hence you need to remove the CryoSat-2 part of this sentence.

*Response:* We realized the negligence but chose to delete the URL since it has been given in the "Data availability" section.

Line 97: "gridded" –> "grid"

*Response:* Agreed.

Line 98: "consistent" –> why consistent? What do you mean by this?

*Response:* We used a wrong word and it should be "successive". We aimed to point out that there is no gap for the monthly average data set and it also provides data in the melting season, which is different from other remote sensing data in the Arctic.

Line 108: "girds" –> "grids"

*Response:* Agreed.

Line 112: You have defined ICESat already, so no need to still use the full name. Line 113: This is total freeboard, right? See GC1.

*Response:* We amended this sentence: "ICESat, operated as part of NASA's Earth Observing System, provides a set of Antarctic total freeboard data from 2003 to 2009."

Line 116: "five main" –> "several"

*Response:* Agreed.

Line 119: "suggested by Worby" –> see comment I had under "specific comments" already.

*Response:* We realized this but chose to delete "suggested by ..." since Kern et al. (2016) has been mentioned at the beginning of this sentence.

Line 126: "ASPeCt observations" –> cite the respective Worby et al. (2008) paper here. Line 128: This is total freeboard, right? See GC1.

*Response:*

(1) We added the citation: "which is a seasonally dependent factor and calculated from ASPeCt observations (Worby et al., 2008a)".

(2) Yes, we amended this: "where F represents ICESat total freeboard".

**References:**

Beaven, S. G., Lockhart, G. L., Gogineni, S. P., Hossetnmostafa, A. R., Jezek, K., Gow, A. J., Perovich, D. K., Fung, A. K., and Tjuatja, S.: Laboratory measurements of radar backscatter from bare and snow-covered saline ice sheets, Int. J. Remote Sens., 16, 851–876, https://doi.org/10.1080/01431169508954448, 1995.

Behrendt, A.: The Sea Ice Thickness in the Atlantic Sector of the Southern Ocean, Ph.D. thesis, University of Bremen, Germany, 239 pp., 2013.

Behrendt, A., Dierking, W., Fahrbach, E., and Witte, H.: Sea ice draft in the Weddell Sea, measured by upward looking sonars, Earth Syst. Sci. Data, 5, 209–226, https://doi.org/10.5194/essd-5-209-

2013, 2013.

Bunzel, F., Notz, D., and Pedersen, L. T.: Retrievals of Arctic Sea-Ice Volume and Its Trend Significantly Affected by Interannual Snow Variability, Geophys. Res. Lett., 45(21), 11,751-11,759. https://doi.org/10.1029/2018GL078867, 2018.

Cavalieri, D. J., Markus T., and Comiso J. C.: AMSR-E/Aqua Daily L3 12.5 km Brightness Temperature, Sea Ice Concentration, & Snow Depth Polar Grids, Version 3, Boulder, Colorado USA. NASA National Snow and Ice Data Center Distributed Active Archive Center. http://dx.doi.org/10.5067/AMSR-E/AE_SI12.003, 2014.

Connor, L. N., Laxon, S. W., Ridout, A. L., Krabill, W. B., and McAdoo, D. C.: Comparison of Envisat radar and airborne laser altimeter measurements over Arctic sea ice, Remote Sens. Environ., 113, 563–570, https://doi.org/10.1016/j.rse.2008.10.015, 2009.

Harms, S., Fahrbach, E., and Strass, V. H.: Sea ice transports in the Weddell Sea, J. Geophys. Res., 106, 9057–9073, https://doi.org/10.1029/1999JC000027, 2001.

Kwok, R., and Maksym, T.: Snow depth of the Weddell and Bellingshausen sea ice covers from IceBridge surveys in 2010 and 2011: An examination, J. Geophys. Res., 119, 4141-4167, https://doi.org/10.1002/2014JC009943, 2014.

Kern, S., and Ozsoy-Cicek, B.: Satellite Remote Sensing of Snow Depth on Antarctic Sea Ice: An Inter-Comparison of Two Empirical Approaches. Remote Sens., 8, 450, https://doi.org/10.3390/rs8060450, 2016.

Kern, S., Ozsoy-Çiçek, B., and Worby, A. P.: Antarctic sea-ice thickness retrieval from ICESat: Inter-comparison of different approaches, Remote Sens., 8, 538, https://doi.org/10.3390/rs8070538, 2016.

Kern, S., Khvorostovsky K., and Skourup, H.: D4.1 Product Validation & Intercomparison Report (PVIR-SIT), available at: http://icdc.cen.uni-hamburg.de/fileadmin/user_upload/ESA_Sea-Ice-ECV_Phase2/SICCI_P2_PVIR-SIT_D4.1_Issue_1.1.pdf, 2018.

Kurtz, N. T., and Markus, T.: Satellite observations of Antarctic sea ice thickness and volume, J. Geophys. Res., 117, https://doi.org/10.1029/2012JC008141, 2012.

Landy, J. C., Petty, A. A., Tsamados, M., and Stroeve, J. C.: Sea Ice Roughness Overlooked as a Key Source of Uncertainty in CryoSat-2 Ice Freeboard Retrievals, J. Geophys. Res., 125, e2019JC015820, https://doi.org/10.1029/2019JC015820, 2020.

Laxon, S., Peacock, N., and Smith, D.: High interannual variability of sea ice thickness in the Arctic region, Nature, 425, 947-950, https://doi.org/10.1038/nature02050, 2003.

Lebedev, V. V.: The dependence between growth of ice in Arctic rivers and seas and negative air temperature (in Russian). Probl. Arkt. 5-6, 9-25, 1938.

Li, H., Xie, H., Kern, S., Wan, W., Ozsoy, B., Ackley, S., and Hong, Y.: Spatio-temporal variability of Antarctic sea-ice thickness and volume obtained from ICESat data using an innovative algorithm, Remote Sens. Environ., 219, 44-61, https://doi.org/https://doi.org/10.1016/j.rse.2018.09.031, 2018.

Paul, S., Hendricks, S., and Rinne, E.: Sea Ice Thickness Algorithm Theoretical Basis Document (ATBD), v1.0, ESA Climate Change Initiative on Sea Ice (SICCI), https://icdc.cen.uni-hamburg.de/fileadmin/user_upload/ESA_Sea-Ice-ECV_Phase2/SICCI_P2_ATBD_D2.1__SIT__Issue_1.0.pdf, 2017.

Paul, S., Hendricks, S., Ricker, R., Kern, S., and Rinne, E.: Empirical parametrization of Envisat freeboard retrieval of Arctic and Antarctic sea ice based on CryoSat-2: progress in the ESA Climate Change Initiative, The Cryosphere, 12, 2437–2460, https://doi.org/10.5194/tc-12-2437-2018, 2018.

Ricker, R., Hendricks, S., Helm, V., Skourup, H., and Davidson, M.: Sensitivity of CryoSat-2 Arctic sea-ice freeboard and thickness on radar-waveform interpretation, The Cryosphere, 8, 1607–1622, https://doi.org/10.5194/tc-8-1607-2014, 2014.

Schwegmann, S., Rinne, E., Ricker, R., Hendricks, S., and Helm, V.: About the consistency between Envisat and CryoSat-2 radar freeboard retrieval over Antarctic sea ice, The Cryosphere, 10, 1415–

1425, https://doi.org/10.5194/tc-10-1415-2016, 2016.

Tilling, R., Ridout, A., and Shepherd, A.: Assessing the Impact of Lead and Floe Sampling on Arctic Sea Ice Thickness Estimates from Envisat and CryoSat-2, J. Geophys. Res., 124, 7473–7485, https://doi.org/10.1029/2019JC015232, 2019.

Worby, A. P., Geiger, C. A., Paget, M. J., Van Woert, M. L., Ackley, S. F., and DeLiberty, T. L.: Thickness distribution of Antarctic sea ice, J. Geophys. Res., 113, https://doi.org/10.1029/2007JC004254, 2008a.

Zelli, C., and Aerospazio, A.: ENVISAT RA-2 advanced radar altimeter: Instrument design and pre-launch performance assessment review, Acta Astronaut., 44, 323–333, https://doi.org/https://doi.org/10.1016/S0094-5765(99)00063-6, 1999.

---

## Author Comment (AC2) · 4 Jun 2020

**General Comments:**

In general, the data comparisons in this paper are done adequately, but the analysis of differences is speculative and weak. I think major revisions will be needed.

Dear Reviewer:

We would like to thank you for the helpful comments to improve this manuscript. We made major revisions on this study as suggested. Specifically, we added more description about the importance of Antarctic sea ice thickness in the Introduction part. We added the discussion of possible uncertainties induced by comparing large-footprint satellite remote sensing data (Envisat and ICESat) with point measurements (ULS observations). We also corrected the mistake that Point 3 referred to by considering the sea ice thickness growth during the freezing season, rather than the melting season. In Section 4, we added some quantitative analyses to avoid the speculative and weak statements.

The specific responses and revisions are shown below. They are in blue font for clarity.

Corresponding Authors:

Qinghua Yang (yangqh25@mail.sysu.edu.cn) and Qian Shi (shiq9@mail.sysu.edu.cn)

**Main Comments:**

**Point 1:** The first paragraph of the Introduction discusses the extent of Antarctic sea ice, but it fails to mention the fact that almost all of that sea ice is seasonal – it completely melts away every year, except in a small portion of the western Weddell Sea. Therefore, it's not clear to me why "sea ice thickness is an equally critical component as sea ice extent" (line 33). I understand the importance of sea-ice extent as a barrier between the ocean and the atmosphere that affects albedo and the exchange of heat and moisture, but I think the authors need to explain better why Antarctic sea-ice thickness is so critical, given that almost all the ice melts away every year.

*Response:* We have added the description about the importance of sea ice thickness in the first paragraph of the Introduction: "Sea ice thickness combined with sea ice extent is necessary to quantify the sea ice volume and sea ice mass (e.g. Kurtz and Markus, 2012; Massonnet et al., 2013).

Changes in sea ice volume can influence the fresh water input into the Southern Ocean. Moreover, sea ice thickness is also necessary for assessing sea ice mass balance, the surface energy budget, and predicting changes in the polar climate system." *(please see P2 line 41-44 in the revised manuscript)*

**Point 2:** In the comparisons of ULS data with Envisat and ICESat (Section 3.1) there is no mention of the fact that ULS measurements are made at a single point, whereas Envisat and ICESat measurements are made over large footprints or areas. How might that affect the comparisons?

*Response:* We discussed the uncertainties caused by such comparison at the end of Section 3.1: "However, it is noted that the ULS measurements are recorded at fixed locations with approximately 6–8 m footprint in diameter, while Envisat has a footprint of 2–10 km and the SIT data used in the comparison represents mean values over 50 km grid cells, and ICESat has a footprint of 70 m and the SIT data represents mean values over 100 km grid cells. This large scale difference can increase the selection biases. When the ULS measures a single point like a ridge or an edge of thin ice, satellites will detect a large area including the single point and other sea ice, and their SIT are averaged through the area. In addition, although the ULS SIT and satellite SIT are all monthly mean values, one satellite SIT grid cell are actually scanned once or twice through a month. And the average of one or two values has a poor representation of the mean SIT throughout the whole month. Theoretically, more valid measurements in one grid cell, more accurate the mean SIT is. In general, uncertainties from both the spatial interpolation and temporal representation can affect the comparisons. However, considering the typical sea ice motion in the Weddell Sea, monthly average ULS sea ice thickness could be referred as a spatial average value, represent one hundred kilometers around the fixed ULS positions. In general, Envisat and ICESat can overpass ULS positions several times a month and are comparable to that of ULS SIT." *(please see P9 line 259-271 in the revised manuscript)*

**Point 3:** Lines 232-238. The mean sea-ice thickness in both the Envisat and ICESat data increases from spring to summer. The authors call this an "anomalous thickness growth" as if it can't possibly be true, and they attribute it to "limited comparison pairs" and "uncertainties of both data sets". However, isn't it possible that the mean ice thickness could actually be greater in summer, because the thinnest ice melts away, leaving only thicker ice? Antarctic sea-ice extent is about 18 million sq km in spring and 3 million sq km in summer. Suppose the spring ice extent consists of 15 million sq km of 1.3 m ice and 3 million sq km of 3.2 m ice, for a mean thickness of about 1.6 m (matching the actual spring ICESat mean thickness). And suppose that all the ice loses 1.3 m of thickness in the summer melt. Then only 3 million sq km are left, with a mean thickness of 3.2 - 1.3 = 1.9 m (matching the actual summer ICESat mean thickness). I'm not saying that these are the correct numbers, I'm just pointing out the plausibility of the argument that the mean thickness could be greater in summer than in spring. Of course it requires a more careful analysis.

*Response:* We recognized this fact and strongly agreed with you. We reproduced the probability distribution figure and changed the bin size into 0.2 m, shown in Fig.1. In addition, we chose to focus on the different growth of the two SIT products from autumn to spring following the other referee's comments. We added the analysis:

"We notice that Envisat and ICESat mean SIT have different variations from autumn to spring. ICESat SIT increases while Envisat SIT does not. For Envisat SIT, the modal (mean) values change from 1.70 m (1.95 m) to 1.30 m (1.60 m) and the distribution indicates that more thin ice is produced. But for ICESat SIT, the modal SIT stays constant of 1.1 m, while the mean values vary from 1.36 m to 1.60 m, and the distribution indicates that more thick ice is found." *(please see P12 line 339-343 in the revised manuscript)* We further examined this problem by comparison the variations with that from FDD model and the results are shown in the response for Point 4 (4).

[Figure]

Figure 1. Probability of the Envisat SIT and the ICESat SIT for all the individual comparison pairs. The blue stairs represent Envisat ice thickness and the red stairs represent ICESat ice thickness. The solid lines indicate the modal ice thickness and the dashed lines indicate the mean ice thickness of both data sets.

**Point 4:** Section 4 discusses the reasons for the differences between Envisat and ICESat sea-ice thickness: (i) if the snow is wet, the Envisat radar does not penetrate all the way to the snow/ice interface; (ii) the footprints of Envisat and ICESat are different; (iii) snow depth is treated differently in the retrieval algorithms. These are all legitimate potential reasons for the differences between Envisat and ICESat ice thickness, but as presented here, they are speculative and qualitative arguments, not quantitative. For example, consider equation (1) for the ice thickness from Envisat, in which F is the measured freeboard and S is the assumed snow depth. If the radar backscatter is from wet snow within the snow layer, rather than the snow/ice interface, then the measured freeboard F is partially ice and partially wet snow. This would lead to a modification of equation (1) and a new ice thickness I' instead of I. Does the difference I-I' account for the bias in Envisat relative to ICESat? Another example: regarding snow depth, how much of a change in snow depth in the Envisat retrieval algorithm would be needed to account for the bias in ice thickness relative to ICESat? Is this change in snow depth within the uncertainty of the snow depth measurements?

Another example: Figures 10 and 11 suggest a connection between ice thickness differences and relative humidity. What is the correlation? Have other researchers considered this connection? Another example: a very simple model of ice thickness is based on cumulative freezing-degree-days (FDD), e.g. Lebedev 1938. Using temperature fields from (say) a reanalysis product, how does a simple FDD ice thickness model compare to Envisat and ICESat ice thickness? Would this provide any insight into biases? My overall point is that the analysis in this paper (Section 4) needs to be more quantitative. The authors claim that "without enough observation data and numerical model experiments we cannot quantify the impacts of the uncertainties over the sea ice thickness." (lines 327-328). But my suggestions above for further quantitative analysis do not require any additional ice thickness data or numerical model runs that are not already publicly available. This is not a question of lack of data, it's a question of digging into the comparisons of Section 3 more quantitatively.

*Response:*

(1) We did not consider this I-I' before, because the uncertainties of Envisat caused by wet snow are not involved in SICCI data. However, we conducted some quantitative analyses here by assuming the total freeboard derived from ICESat and the snow depth product used in the retrieval of Envisat SIT are accurate. Then we can calculate the expected difference *D'* between Envisat and ICESat SIT caused solely by the radar altimeter penetration. Considering Eq. (1) in the manuscript for a transformation adapted to ICESat total freeboard *Ft*:

$$I' = \frac{(Ft-S) \cdot \rho_{water} + S\rho_{snow}}{\rho_{water} - \rho_{ice}} \quad ,$$

(6)

where *S* represents snow depth, *I'* represents the new sea ice thickness, $\rho_{water}$, $\rho_{snow}$, $\rho_{ice}$ refer to the density of the sea water, snow cover and sea ice, respectively. Then we can derive a new difference by deducting Eq. (6) from Eq. (1):

$$D' = \frac{\rho_{water} \cdot (F+S-Ft)}{\rho_{water} - \rho_{ice}}$$

(7)

where *F* represents Envisat sea ice freeboard. We set $\rho_{water}$ for 1024 kg m$^{-3}$, $\rho_{ice}$ for 916.7 kg m$^{-3}$ and *S* for the NSIDC AMSR-E snow depth (nsidc.org/data/ae_si12). For the three seasons which are spring, summer and autumn, we can calculate that the values are -0.57 m, 0.47 m and 0.32 m, respectively. Compared to the seasonal average difference shown in Table 5, which are -0.01 m, 0.51 m and 0.53 m, we can conclude that differences in summer are mainly due to radar altimeter inability. Differences in spring cannot be explained by this bias, implying that snow depth uncertainties are the major cause. However, the real difference caused by penetration is a bit more complicated than that. If the range has only partial penetration, then it is also required to adjust the snow propagation speed correction since the radar waves are not traversing the entire snow layer. *(please see P14-15 line 414-428 in the revised manuscript)*

(2) We also conducted some quantitative analyses on the uncertainties caused by the biased snow

depth. Assuming that sea ice freeboard derived from Envisat is accurate, we can calculate the required snow depth $\Delta S$ for compensating the difference between Envisat and ICESat SIT ($D$). According to Eq. (1), we can derive that:

$$\Delta S = \frac{\rho_{water} - \rho_{ice}}{\rho_{snow}} \cdot D$$

(8)

Setting $\rho_{snow}$ for 300 kg m$^{-3}$, we can get $\Delta S$ for spring, summer and autumn as -0.36 cm, 18.24 cm and 18.96 cm, respectively. According to Kern et al. (2015), the average monthly retrieval uncertainties are commonly below 2 cm, albeit the standard deviations of the data are substantially larger than uncertainties. Therefore, we can infer that in spring the cause of differences is mainly the bias of snow depth product, while in summer and autumn the biases mainly come from the differences of observational technique. *(please see P16 line 466-473 in the revised manuscript)*

(3) We plotted the seasonal average maps of the four parameters as the other referee suggested, including 2 meter temperature, sea surface temperature, surface net solar radiation and surface net thermal radiation. All of them come from ERA-Interim reanalysis data. We also calculated the linear correlation coefficients between the difference of ENV-ICE SIT and each meteorological parameter in each season and the results are shown below in Table 8. We can see that the best linear correlation coefficient is the relative humidity and the others are all bad, albeit the relation with humidity is also weak. We assume that there might be relations between the difference and temperature and radiation, but they are absolutely not linear relations. In order to dig into this relation, more analyses are needed. However, this is not the kernel in this study. Therefore, we considered removing this part.

Many studies have pointed out that wet snow can affect the penetration. But they don't combine the relative humidity with the difference before, since the relative humidity values are not that accurate due to the precipitation observations in the Antarctic are difficult and unreliable.

Table 1. The linear correlation coefficients between the difference of ENV-ICE SIT and each meteorological parameter in each season. The values in italic type have not passed the 95% significance test.

|  | Relative humidity | 2 meter temperature | Sea surface temperature | Surface net solar radiation | Surface net thermal radiation |
|---|---|---|---|---|---|
| ON | 0.28 | -0.12 | -0.14 | -0.24 | *0.01* |
| FM | 0.38 | -0.26 | -0.44 | -0.37 | 0.17 |
| MJ | 0.52 | -0.25 | -0.23 | -0.22 | 0.10 |

[Figure]

Figure 2. Seasonal average maps of 2 meter temperature, sea surface temperature, surface net solar radiation and surface net thermal radiation based on ICESat operating periods.

(4) We use 2 meter temperature data derived from ERA-5 reanalysis provide by Copernicus Climate Change Service (C3S) (2017) to generate the accumulative freezing-degree-days (FDD). According to Lebedev (1938), we construct a simple model to produce sea ice thickness and examine the growth from autumn (MJ) to spring (ON) every year. Then we compare it with the thermodynamic growth represented by Envisat and ICESat SIT. The results are added in the manuscript: "Since May-June period is the beginning of the upswing freezing season and October-November period is

near the end of it, we assume that thermodynamic growth contributes more to the SIT variations than dynamic growth. Therefore, we further compare this development with the FDD model results in 2004, 2005 and 2006, shown in Table 6. As introduced in Sect. 2.4, this model is only involved in the temperature, representing the thermodynamic growth and neglecting any dynamic contributions to sea ice thickness. Figure 8 shows the sea ice thickness growth maps from autumn to spring derived from FDD model, Envisat and ICESat in 2004, 2005 and 2006. Table 6 shows the statistical results of the growth maps. We calculate the period-average SIT from the model corresponding to ICESat operating periods. The results show that Envisat SIT has an opposite development from ICESat and model. And the development of ICESat SIT is smaller than and not constant as the model results. Although ICESat SIT has an overall increase, it also shows a decrease in Western Weddell Sea similar to Envisat SIT. In addition, both of them show an increase along the coast of Amundsen Sea in 2006. Combined with the typical sea ice motion in the Weddell Sea and Ross Sea shown in the first panel, we consider that ICESat shows more realistic sea ice change, e.g. a decrease SIT in the Ross Sea polynya and increase around it, or the increase in the East Weddell Sea." *(please see P12 line 343-355 in the revised manuscript)*

[Figure]

Figure 3. The sea ice thickness growth maps from autumn to spring derived from FDD model, Envisat and ICESat in 2004, 2005 and 2006. The arrows refer to the typical sea ice motions in the West Weddell Sea and Ross Sea.

Table 2. Statistical results of the sea ice thickness development from autumn (MJ) to spring (ON) in 2004, 2005 and 2006.

|  | Envisat SIT (m) | ICESat SIT (m) | FDD (m) |
|---|---|---|---|
| MJ04 | 1.94 | 1.33 | 0.55 |
| ON04 | 1.64 | 1.64 | 1.11 |
| ON04-MJ04 | -0.30 | 0.31 | 0.56 |
| MJ05 | 1.98 | 1.43 | 0.57 |
| ON05 | 1.59 | 1.54 | 1.13 |
| ON05-MJ05 | -0.39 | 0.11 | 0.56 |
| MJ06 | 1.92 | 1.32 | 0.58 |
| ON06 | 1.44 | 1.63 | 1.14 |
| ON06-MJ06 | -0.48 | 0.31 | 0.56 |

**Minor Comments:**

Lines 19-23. It's not clear whether the deviations are Envisat minus ICESat, or ICESat minus Envisat.

*Response:* We added "that Envisat SIT minus ICESat SIT" to explain this ambiguity.

Line 21. "the large correlation coefficient" – Is this a spatial correlation or a temporal correlation? Please add the word "spatial" or "temporal" as appropriate.

*Response:* This is the correlation between two SIT distribution maps and we added "spatial" in the revised manuscript.

Line 103. Following equation (1), say that rho is density.

*Response:* We add this information: "$\rho_{water}$,$\rho_{snow}$,$\rho_{ice}$ refer to the density of the sea water, snow cover and sea ice respectively." *(please see P5 line 136-137 in the revised manuscript)*

Line 104. Give a reference for AMSR-E snow depth climatology.

*Response:* The snow depth climatology is produced by and for ESA CCI by Stefan Kern. This snow-depth climatology is derived from Advanced Microwave Scanning Radiometer-EOS (AMSR-E) and AMSR-2 data for the Antarctic and is based on a revised version of the approach described by Cavalieri et al. (2014) and provided by the Integrated Climate Data Center (ICDC, http://icdc.cen.uni-hamburg.de). *(please see P5 line 138-141 in the revised manuscript)*

Line 108. A threshold of 70% is given here, but on line 131 it says 60%.

*Response:* The usage of different SIC thresholds is because of the different thresholds used in the retrieval of the two data sets. Envisat SIT employs a SIC threshold of 70% during the retrieval while

the ICESat SIT uses 60%. Only areas with sea ice concentrations greater than the threshold are considered a valid area for detection of leads and sea ice. Therefore, we are meant to point out this information. In order to express more clearly, we changed the two sentences to:

"In addition, it is noted that values with sea ice concentration less than 70 % have been removed during Envisat SIT retrieval." *(please see P5 line 145-146 in the revised manuscript)*

"It is noted that grid cells with sea ice concentration less than 60% have been remove during ICESat SIT retrieval." *(please see P6 line 170-171 in the revised manuscript)*

We also tested the difference between using 60% and 70% SIC threshold for ICESat during the comparison with Envisat SIT. According to Table 3, this different threshold does not play an important role in the results of this paper. D(60) refers to Envisat minus ICESat (ENV-ICE) applying 60% SIC threshold for ICESat, while D(70) refers to ENV-ICE when SIC threshold for ICESat is 70%. Since the ice concentration gradients are usually quite steep, there will not be a lot of area with values 60% < SIC < 70%.

Table 3. Statistical results of the comparison between Envisat SIT and ICESat SIT using 60% and 70% SIC threshold at each operating period.

| | ON04 | ON05 | ON06 | ON07 | FM04 | FM05 | FM06 | MA07 | FM08 | MJ04 | MJ05 | MJ06 |
|---|---|---|---|---|---|---|---|---|---|---|---|---|
| D(60) | 0.00 | 0.05 | -0.19 | 0.14 | 0.89 | 0.74 | 0.47 | 0.61 | 0.92 | 0.61 | 0.55 | 0.60 |
| D(70) | 0.00 | 0.06 | -0.21 | 0.15 | 0.79 | 0.66 | 0.43 | 0.61 | 0.89 | 0.60 | 0.55 | 0.61 |

Lines 125-126. "R is the ratio of sea ice thickness over snow depth, which is a seasonally dependent factor and calculated from ASPeCt observations." I understand that R changes seasonally, but does it also change from year to year based on ASPeCt observations?

*Response:* No, R values are constant in each season (6.8 in summer, 6.0 in autumn, and 5.4 in spring) and do not change from year to year.

Line 139, equation (4). Please provide UNITS for the quantities in this equation.

*Response:* We added the units in the equation: $z(m) = 0.028+1.012d(m)$. *(please see P7 line 183 in the revised manuscript)*

Line 146. "the deviations." – please say explicitly "the deviations between Envisat and ICESat sea-ice thickness" or whatever deviations are being referenced here.

*Response:* We refined it to "the differences between Envisat and ICESat sea-ice thickness".

Lines 151-153. "The seasonal classification is based on the ICESat operating periods... If ICESat data has overlapping time over ten days with respective months, we average Envisat data over the two months." OK, but wouldn't it be better to do a time-weighted average of the monthly Envisat data to match the ICESat period? For example, consider the ICESat period from Feb 17 to Mar 20.

Instead of averaging Envisat over all of February and March, consider this: the ICESat period is 32 days long – 12 days in February and 20 days in March. So calculate: (Env. avg.) = (12/32)*(Env. Feb.) + (20/32)*(Env. Mar.) Wouldn't that provide a more accurate Envisat average with which to compare ICESat?

*Response:* Thanks for your comments. We followed your suggestions and did a time-weighted average of the monthly Envisat data to match the ICESat period. But it is noted that the little change has little affect on the results of the comparisons but only add more accuracy.

Line 164. For the ULS data, I understand that "207" refers to location #207 in Figure 2, but I don't understand "207-6" – what is the "6"? Please explain your numbering system.

*Response:* The number appended behind the ULS station numbers refer to different ULS operation periods at each location. For example, "207-6" represents the sixth leg of measurements at the station #207, from 14 March 2005 to 27 March 2008. However, we combined the successive records for 229 and 231, so we changed the expression here as: "there are only three sites having enough valid data for the evaluation: 207, 229 and 231".

Line 168. "Due to the discontinuity..." – what discontinuity?

*Response:* The "discontinuity" here means the ICESat measurements are not continuous due to cloud coverage or lack of valid data. Each measurement campaign lasts for about 35 days and only operates three times a year. Therefore, the number of ICESat measurements for comparison with ULS is limited. To make it clearer, we change "discontinuity" to "operating period gaps".

Lines 173-174. In Figure 4, where do the error bars come from? What do they represent? One standard deviation? 95th percentile?

*Response:* The Envisat error bars come from the SIT uncertainty $\sigma_{sit}$ contained in the Envisat SIT product and is computed as the error propagation of all input uncertainties with the assumption that the sea water density is negligible (Paul et al. 2017):

$$\sigma_{sit} = \sqrt{(\frac{\rho_w}{\rho_w-\rho_i}\sigma_{frb})^2 + (\frac{frb\cdot\rho_w+sd\cdot\rho_i}{\rho_w-\rho_i}\sigma_\rho^i)^2 + (\frac{\rho_s}{\rho_w-\rho_i}\sigma_{sd})^2 + (\frac{sd}{\rho_w-\rho_i}\sigma_\rho^s)^2}$$

where *frb* represents Envisat sea ice freeboard, *sd* represents snow depth, $\rho_w$, $\rho_s$, $\rho_i$ refer to the density of the sea water, snow cover and sea ice, $\sigma_{frb}$, $\sigma_\rho^i$, $\sigma_{sd}$, $\sigma_\rho^s$ represent the uncertainties of sea ice freeboard, ice density, snow depth and snow density, respectively.

ICESat SIT uncertainties $\sigma_I$ are calculated based on the uncertainties of sea ice and snow, also neglecting the uncertainty of water density (Li et al., 2018):

$$\sigma_I = \sqrt{(\frac{R*F*\rho_w}{(R+1)*(\rho_w-\rho_i^*)})^2\sigma_{\rho_i}^2 + (\frac{F*\rho_w}{(R+1)*(\rho_w-\rho_i^*)})^2\sigma_{\rho_s}^2}$$

where $F$ represents ICESat total freeboard, $R$ represents the sea ice thickness to snow depth ratio, $\rho_w$, $\rho_s$, $\rho_i^*$ refer to the density of the sea water, snow cover and the modified density of sea ice,

$\sigma_{\rho_i}$, $\sigma_{\rho_s}$ represent the uncertainties of sea ice density and snow density, respectively.

Therefore, we added the information as: "We draw the error bars from uncertainty information provided by both SIT products. The Envisat SIT uncertainty is computed as the error propagation of all input uncertainties with the assumption that the sea water density is negligible (see Section 2.9.8 in Paul et al. 2017). ICESat SIT uncertainties are calculated based on the uncertainties of sea ice and snow, also neglecting the uncertainty of water density (see Eq. (6) in Li et al., 2018)." *(please see P8 line 237-240 in the revised manuscript)*

Line 190. It would be helpful to indicate on one of the maps the location of the Ross Ice Shelf polynya and the Ronne Ice Shelf polynya.

*Response:* The locations of the two polynyas have been indicated on the figure.

[Figure]

[Figure]

Figure 4. Map of the different sectors referred to in the study. The background is the average of the September sea ice thickness from Envisat during 2003-2011 with 50 km grid size. Each sector and the two ice shelf polynyas are indicated in the figure. The circles and the corresponding numbers refer to the sites of the ULS. The white grid cells stand for open water or sea ice with concentration less than 70% or missing data.

Line 192. I think "clockwise" should be "counter-clockwise". Please check.

*Response:* Thanks for your remind but here we removed this statement as the other referee's comments because this is not critical in the paper.

Lines 216-217. "Comparing the values in the eastern Antarctic, ICESat shows some deformed ice up to 3 m while Envisat shows smaller thickness by about 1.5 m." I don't see this in Figure 7. Please give approximate longitudes or otherwise indicate in Fig 7.

*Response:* Along the coast of the Eastern Antarctic (90–160), there are some points of larger SIT from ICESat, which can be proved by the negative deviations in the difference maps.

[Figure]

Figure 5. Comparisons of Envisat versus ICESat sea ice thickness for each ICESat operating period in autumn (May & June). The first and second columns show the sea ice thickness distribution of Envisat and ICESat respectively, and the last column shows the difference map (Envisat minus ICESat) of sea ice thickness. Each row represents a year from 2004 to 2007. The sea ice thickness maps are at their native grid resolution while the difference map is interpolated onto the polar-stereographic grid of the ICESat product. The white cells denote sea ice concentration less than threshold or missing data.

Lines 241-242. In reference to Figure 9, "In the western Weddell Sea, the regression lines have large positive intercepts in all three seasons." Yes, this is true for all five regions, not just the western Weddell Sea.

*Response:* Agreed, we changed the sentence to "For all the five panels, the regression lines have large positive intercepts in all three seasons." *(please see P13 line 377-378 in the revised manuscript)*

Lines 261-262. "ICESat uses the modified snow-ice density to get rid of the biased snow depth." I don't see how equation (2) would get rid of a biased snow depth. The snow depth S is part of the factor R = I/S.

*Response:* Here we were meant to clarify that this product can get rid of the biased snow depth from remote sensing. Based on the previous study (Kern and Ozsoy-Cicek 2016; Kern and Ozsoy 2019; Price et al. 2019), Antarctic snow depth products from remote sensing have a lot of uncertainties. Although the one-layer-method employs the ASPeCt SIT and snow observations, which are also likely biased, the ratio between them is more reliable.

Lines 295-298. "wet snow caused by melt or flooding could lead to underestimations while refreezing of molten snow could lead to overestimations [of snow depth]. All of the above biases can also cause the differences between Envisat and ICESat." I don't see how underestimates AND overestimates of snow depth can BOTH lead to a positive bias in Envisat ice thickness relative to ICESat. Please clarify.

*Response:* Here "cause the differences" does not refer to the actual difference between Envisat and ICESat SIT. It only means that the biases in snow depth might affect the difference between Envisat and ICESat SIT. For the calculated differences in each season, we will discuss the main reasons respectively.

Lines 300-301. "in underestimations of snow depth... the sea ice thickness deviation presents negative." It's not clear to me whether "deviation" refers to the error in the Envisat ice thickness, or the difference Envisat-ICESat ice thickness. Consider equation (1). If the snow depth S is an underestimate, then the true snow depth S' > S. If the measured freeboard F remains constant, then the true ice thickness I' > I. The error I-I' < 0 (i.e. negative error in the original estimate I). On the other hand, I'-ICESat > I-ICESat (i.e. increased bias of Envisat). Please clarify the use of "deviation" and the effect of snow depth on the calculated ice thickness.

*Response:* The "deviation" refers to the difference of Envisat-ICESat ice thickness. We clarified this part as: "Considering Eq. (1), if sea ice freeboard F remains constant, and the snow depth is underestimated, then the Envisat sea ice thickness is biased low, and the difference that Envisat minus ICESat SIT is negative. Therefore, we attribute the negative differences shown in Fig. 4 to snow depth uncertainty." *(please see P16 line 477-480 in the revised manuscript)*

Figure 2. It would be helpful to outline the zero contour (the coastline) to make it easier to distinguish land from ocean. Also, the caption should say that the background is bathymetry, and give the source of the bathymetry data.

*Response:* Although we reproduced the figure in the following, we chose to remove this figure and marked the three sites used in the comparison in Fig. 2 in the revised manuscript. The figure has been modified as follows:

[Figure]

Figure 6. Map of the AWI ULS mooring locations. The background is the land topography and ocean bathymetry from ETOPO1 Global Relief Model data (doi:10.7289/V5C8276M). The circles and the corresponding numbers in the white boxes refer to the sites of the ULS. The gray line refers to the coastline of the Antarctica.

Modified from Fig.2 in Behrendt et al. (2013).

Figure 3. Please add at the end of the caption: "with 50 km grid size."

*Response:* Accepted.

[Figure]

Figure 7. Map of the different sectors referred to in the study. The background is the average of the September sea ice thickness from Envisat during 2003-2011 with 50 km grid size. Each sector and

the two ice shelf polynyas are indicated in the figure. The circles and the corresponding numbers refer to the sites of the ULS. The white grid cells stand for open water or sea ice with concentration less than 70% or missing data.

Figure 4. See comment above for line 164: I understand that 207 refers to a location in Figure 2, but what does "207-6" mean? Also, see comment above for lines 173-174: in the caption, say what the error bars represent. Also, the dates along the horizontal axes should be in a more readable format such as 2008/03 instead of 200803. Perhaps the journal has a standard format for such dates.

*Response:* The number appended behind the ULS station numbers refer to different ULS operation periods at each location. For example, "207-6" represents measurements from 14 March 2005 to 27 March 2008 at location #207. But here we reproduced the figures combining different periods at one site together.

[Figure]

Figure 8. Time series of sea ice thickness and their errors for the Weddell Sea ULS, Envisat and ICESat. The numbers on the top of each panel represent the location of each site for the comparisons. The site locations can be searched in Fig.2. ICESat SIT values are placed between the two months that each period covers. The mean differences and their standard deviations are shown in the figures.

Figure 6. Consider rotating the whole figure into landscape mode, which would allow the panels to be larger.

*Response:* The figure has been modified as follows:

[Figure]

Figure 9. Comparisons of Envisat versus ICESat sea ice thickness for each ICESat operating period in summer (February & March). The first and second columns show the sea ice thickness distribution of Envisat and ICESat respectively, and the last column shows the difference map (Envisat minus ICESat) of sea ice thickness. Each row represents a year from 2004 to 2007. The sea ice thickness maps are at their native grid resolution while the difference map is interpolated onto the polar-stereographic grid of the ICESat product. The white cells denote sea ice concentration less than threshold or missing data.

New table. This is just a suggestion, but I found it helpful to create a table for myself of the different data sources, their spatial and temporal resolutions, and their treatment of snow. For example:

Source | Spatial res | Temporal res | Snow ——————————————————————————-
Envisat | 50 km grid | monthly avg | AMSR-E climatology ICESat | 100 km grid | see Table 1 | ASPeCt observations ULS | single point | monthly avg | built into eq (4)

*Response:* Thanks for your suggestion. We added the new table in the revised manuscript.

Table 4. A summary of the sea ice thickness data used during the comparison, including

different data sources, spatial resolution, temporal resolution and snow product.

| Source | Instrument | Operation time | Footprint | Grid resolution | Temporal resolution | Snow product |
|--------|-----------|----------------|-----------|-----------------|---------------------|--------------|
| Envisat satellite | Radar altimeter | 2002-2011 | 2–10 km | 50 km grid | Monthly average | AMSR-E climatology |
| ICESat satellite | Laser altimeter | 2003-2009 | 70 m | 100 km grid | See Table 1 | ASPeCt observations |
| Weddell Sea ULS | Upward Looking Sonars | 1990-2010 | 6–8 m | Single point | Monthly average | built into Eq. (4) |

**Typographical Corrections**

*Response:* All the suggested typographical problems have been corrected.

Line 280. What is "shorter" ice? Does this mean "less extensive" in area? Line 281. Does "wider" mean "more extensive"?

*Response:* Yes, "shorter" means "less extensive" and "wider" means "more extensive". In the revised manuscript we have modified the paragraph and deleted those words.

Line 296. What is "molten snow"? Is it "wet snow"?

*Response:* Yes, "molten snow" is the "wet snow" and we amended this.

**References:**

Behrendt, A., Dierking, W., Fahrbach, E., and Witte, H.: Sea ice draft in the Weddell Sea, measured by upward looking sonars, Earth Syst. Sci. Data, 5, 209–226, https://doi.org/10.5194/essd-5-209-2013, 2013b.

Cavalieri, D. J., Markus T., and Comiso J. C.: AMSR-E/Aqua Daily L3 12.5 km Brightness Temperature, Sea Ice Concentration, & Snow Depth Polar Grids, Version 3, Boulder, Colorado USA. NASA National Snow and Ice Data Center Distributed Active Archive Center. http://dx.doi.org/10.5067/AMSR-E/AE_SI12.003, 2014.

Copernicus Climate Change Service (C3S) (2017): ERA5: Fifth generation of ECMWF atmospheric

reanalyses of the global climate. Copernicus Climate Change Service Climate Data Store (CDS), date of access. https://cds.climate.copernicus.eu/cdsapp#!/home

Kern, S., and Ozsoy-Cicek, B.: Satellite Remote Sensing of Snow Depth on Antarctic Sea Ice: An Inter-Comparison of Two Empirical Approaches. Remote Sens., 8, 450, https://doi.org/10.3390/rs8060450, 2016.

Kern, S., and Ozsoy, B.: An attempt to improve snow depth retrieval using satellite microwave radiometry for rough antarctic sea ice. Remote Sens., 11, 2323, https://doi.org/10.3390/rs11192323, 2019.

Kern, S., Frost, T., and Heygster, G.: D1.3 Product User Guide (PUG) for Antarctic AMSR-E snow depth product SD v1.1, available at : https://icdc.cen.uni-hamburg.de/fileadmin/user_upload/ESA_Sea-Ice-ECV/SICCI_ANT_SIT_Option_PUG_D1.3_Issue_2.1_final.pdf, 2015.

Kurtz, N. T., and Markus, T.: Satellite observations of Antarctic sea ice thickness and volume, J. Geophys. Res., 117, https://doi.org/10.1029/2012JC008141, 2012.

Lebedev, V. V.: The dependence between growth of ice in Arctic rivers and seas and negative air temperature (in Russian). Probl. Arkt. 5-6, 9-25, 1938.

Li, H., Xie, H., Kern, S., Wan, W., Ozsoy, B., Ackley, S., and Hong, Y.: Spatio-temporal variability of Antarctic sea-ice thickness and volume obtained from ICESat data using an innovative algorithm, Remote Sens. Environ., 219, 44-61, https://doi.org/https://doi.org/10.1016/j.rse.2018.09.031, 2018.

Massonnet, F., Mathiot, P., Fichefet, T., Goosse, H., König Beatty, C., Vancoppenolle, M., and Lavergne, T.: A model reconstruction of the Antarctic sea ice thickness and volume changes over 1980–2008 using data assimilation, Ocean Model., 64, 67-75, https://doi.org/https://doi.org/10.1016/j.ocemod.2013.01.003, 2013.

Paul, S., Hendricks, S., and Rinne, E.: Sea Ice Thickness Algorithm Theoretical Basis Document (ATBD), v1.0, ESA Climate Change Initiative on Sea Ice (SICCI), https://icdc.cen.unihamburg.de/fileadmin/user_upload/ESA_Sea-Ice-ECV_Phase2/SICCI_P2_ATBD_D2.1__SIT__Issue_1.0.pdf, 2017.

Price, D., Soltanzadeh, I., Rack, W., and Dale, E.: Snow-driven uncertainty in CryoSat-2-derived Antarctic sea ice thickness – insights from McMurdo Sound, The Cryosphere, 13, 1409–1422, https://doi.org/10.5194/tc-13-1409-2019, 2019.